# Synergistic dual-phase air electrode enables high and durable performance of reversible proton ceramic electrochemical cells

Zuoqing Liu[1], Yuesheng Bai[1], Hainan Sun [2], Daqin Guan [3], Wenhuai Li[1], Wei-Hsiang Huang[4], Chih-Wen Pao [4], Zhiwei Hu [5], Guangming Yang [1]✉, Yinlong Zhu [6]✉, Ran Ran[1], Wei Zhou [1] & Zongping Shao [1,7]✉

Reversible proton ceramic electrochemical cells are promising solid-state ion devices for efficient power generation and energy storage, but necessitate effective air electrodes to accelerate the commercial application. Here, we construct a triple-conducting hybrid electrode through a stoichiometry tuning strategy, composed of a cubic phase $Ba_{0.5}Sr_{0.5}Co_{0.8}Fe_{0.2}O_{3-\delta}$ and a hexagonal phase $Ba_4Sr_4(Co_{0.8}Fe_{0.2})_4O_{16-\delta}$. Unlike the common method of creating self-assembled hybrids by breaking through material tolerance limits, the strategy of adjusting the stoichiometric ratio of the A-site/B-site not only achieves strong interactions between hybrid phases, but also can efficiently modifies the phase contents. When operate as an air electrode for reversible proton ceramic electrochemical cell, the hybrid electrode with unique dual-phase synergy shows excellent electrochemical performance with a current density of 3.73 A cm$^{-2}$ @ 1.3 V in electrolysis mode and a peak power density of 1.99 W cm$^{-2}$ in fuel cell mode at 650 °C.

The rapid advancement of modern technology has been driven by the exploitation and extensive use of fossil fuels, but this has also resulted in an unprecedented living dilemma for humanity due to their limited resources and the greenhouse effects of their oxidation product. As a carbon-free energy material, hydrogen is widely accepted to play an essential part in the future sustainable energy system owing to its high energy density and non-pollution nature, and the wide availability of hydrogen element in earth[1]. Reversible solid oxide cell (R-SOC), a device allowing the mutual conversion of chemical and electrical energy, can meet the demand for efficient utilization and production of hydrogen energy[2,3]. Unfortunately, the conventional R-SOCs based on oxygen-conducting electrolytes normally operate at 700 to 900 °C. Such high temperature makes the cells suffer from numerous

challenges, such as difficult hermetic sealing, high operating costs, material incompatibility, and poor durability[4]. Due to the significantly lower energy barrier for proton migration compared to that of oxygen ions, it is believed that reversible proton ceramic electrochemical cells (R-PCECs) can be operated potentially at the medium temperature range (400–700 °C) without an obvious increase in cell ohmic resistance, while the reduction in operation temperature can effectively lower operation costs[5,6]. In addition, the different locations of water formation in fuel cell (FC) operation and hydrogen generation in electrolysis cell (EC) operation for R-PCECs from R-SOCs make them higher energy utilization efficiency and easier hydrogen purification[7]. Therefore, during the past decade, there has been an obviously increasing interest in R-PCECs for power generation and storage[5,8].

[1]State Key Laboratory of Materials-Oriented Chemical Engineering, College of Chemical Engineering, Nanjing Tech University, 211816 Nanjing, People's Republic of China. [2]Department of Materials Science and Engineering, Korea Advanced Institute of Science and Technology (KAIST), Daejeon 34141, Republic of Korea. [3]Department of Building and Real Estate, Research Institute for Sustainable Urban Development (RISUD) and Research Institute for Smart Energy (RISE), The Hong Kong Polytechnic University, Kowloon, China. [4]National Synchrotron Radiation Research Center, 101 Hsin-Ann Road, Hsinchu 30076, Taiwan. [5]Max-Planck-Institute for Chemical Physics of Solids, Nöthnitzer Str. 40, 01187 Dresden, Germany. [6]Institute for Frontier Science, Nanjing University of Aeronautics and Astronautics, 210016 Nanjing, People's Republic of China. [7]WA School of Mines: Minerals, Energy and Chemical Engineering (WASM-MECE), Curtin University, Perth, WA 6845, Australia. ✉e-mail: ygm89525@njtech.edu.cn; zhuyl1989@nuaa.edu.cn; shaozp@njtech.edu.cn

Although favorable progress in a relatively short period of time for the R-PCECs[9–13], their commercialization is still limited, mainly due to the insufficient electrochemical performance at the medium temperature range, as caused by the scarcity of high-efficiency air electrodes with high activity in both FC and EC modes. Designing alternative air electrodes with high oxygen activation and hydration capabilities is an attractive approach to improve the kinetic rate of oxygen reduction and oxygen evolution reactions (ORR and OER) in R-PCECs[14,15]. However, the single-phase air electrode has a unitary structural feature and limited $e^-/O^{2-}/H^+$ transport channels, resulting in single-phase electrodes that generally exhibit only acceptably single catalytic activity in fuel cells or electrolysis cells[16]. Therefore, air electrodes in R-PCECs are extensively researched using dual-phase or multiphase hybrid electrodes, due to the hybrid electrodes can provide more abundant active sites and good ionic conductivity[17]. Additionally, the hybrid electrodes can satisfy a variety of catalytic requirements owing to the synergistic effects of multiphase. The self-assembly method for creating hybrid electrodes, compared to the straightforward physical mixing method, is a promising synthesis strategy because it allows for a homogeneous distribution of multiphase and close phases contact, lowering the energy barriers for electron and ion conduction (Fig. 1a)[18]. For example, Song et al. synthesized a tetragonal and Ruddlesden-Popper (RP) nanocomposite electrode $Sr_{0.9}Ce_{0.1}Fe_{0.8}Ni_{0.2}O_{3-\delta}$ (SCFN) surface-enriched with $CeO_2$ and NiO nanoparticles[19]. NiO and $CeO_2$ nanoparticles can effectively accelerate oxygen adsorption and dissociation, and obtain fast oxygen and vapor surface exchange kinetics and good $e^-/O^{2-}/H^+$ conductivity under the multiphase synergy. A more effective nanocomposite electrode called $Ba_{0.95}(Co_{0.4}Fe_{0.4}Zr_{0.1}Y_{0.1})_{0.95}Ni_{0.05}O_{3-\delta}$ (BCFZYN) was also created by Liang et al. using composition and cationic tailoring manipulation[20]. The air electrode consists predominantly of a cubic perovskite phase, with a minor presence of the NiO phase, and the two phases enhance the bulk proton conduction and oxygen surface exchange process, respectively. As mentioned above, improper tolerance factors of partial elements in oxides induce phase separation, thus producing two or multiphase hybrids with distinct differences in elemental composition. Similarly, hybrid electrodes created through improper tolerance factors include the following: $BaCo_{0.7}(Ce_{0.8}Y_{0.2})_{0.3}O_{3-\delta}$ (BCCY), $Ba_{0.5}Sr_{0.5}Co_{0.6}Fe_{0.2}Zr_{0.1}Y_{0.1}O_{3-\delta}$ (BSCFZY) and $Nd_{0.1}Ca_{0.1}Ba_{1.8}Co_9O_{14}$ (NCBCO)[18,21,22]. However, due to the significant differences in the elemental composition of the different phases, the thermal strain generated between the various phases is difficult to counteract, which results in a decline in the catalytic activity of electrode and a decrease in the stability of the cell (Fig. 1b). Additionally, considering the inevitable elemental migration at higher temperatures, further disclosure is required to determine the relationship between the activity of the hybrid electrode and the composition/content of each phase under practical operating conditions.

Constructing a hybrid with strong heterogeneous interaction and controllable phase structure and composition is an effective strategy to develop stable and efficient catalysts[23]. Xu et al. synthesized a hybrid catalyst with controllable phase content by A-site Sr tailoring-induced phase separation for RP-structured $LaSr_3Co_{1.5}Fe_{1.5}O_{10}$[24]. The hybrid $LaSr_{3-y}Co_{1.5}Fe_{1.5}O_{10}$ (LaSr3-y, $0.1 \le y \le 0.5$) consists of single perovskite (SP) phase $La_{0.33}Sr_{0.67}Co_{0.5}Fe_{0.5}O_3$ and RP phase $LaSr_3Co_{1.5}Fe_{1.5}O_{10}$, in which the LaSr2.7 hybrid exhibits the best OER activity at room temperature. Additionally, it has been demonstrated that the robust coupling contact between the RP and SP phases can effectively promote oxygen ion migration and activate the OER activity of lattice oxygen. However, LaSr3-y hybrids exhibit unacceptable oxygen activation and hydration capacity in the medium temperature range because of the insufficient oxygen vacancies present in them, which severely restricts the applicability of these hybrids as air electrodes for R-PCECs.

It is widely accepted that the cubic perovskite $Ba_{0.5}Sr_{0.5}Co_{0.8}Fe_{0.2}O_{3-\delta}$ (C-BSCF), which contains alkaline earth and transition metals at the A-site and B-site, respectively, has strong ORR activity and oxygen ion conductivity. In 2004, C-BSCF made its pioneering debut as a cathode for solid oxide fuel cells (SOFCs), achieving a remarkable peak power density of 1010 mW cm$^{-2}$ at 600 °C[25]. Furthermore, Zhu et al. produced complex oxide $Ba_4Sr_4(Co_{0.8}Fe_{0.2})_4O_{16-\delta}$ (H-BSCF) with a distinctive hexagonal structure based on C-BSCF by tuning the stoichiometric ratio[26]. H-BSCF was applied as a low-cost catalyst for OER, exhibiting high catalytic activity and excellent stability, which outperforms most of the metal oxides applied for OER. The one-sided high catalytic activity and restricted $e^-/O^{2-}/H^+$ triple conductivity of C-BSCF and H-BSCF make it difficult to be widely utilized as air electrodes for R-PCECs, despite the fact that each material has distinct advantages. Actually, the lack of $e^-/O^{2-}/H^+$ conductivity will restrict the electrocatalytic reaction of the air electrode in PCECs to the triple-phase boundary.

Herein, we report a new family of BSCF-based hybrid electrodes with unique structures and distinct compositions by simply controlling the elemental content ratio of A-site to B-site in $(Ba_{0.5}Sr_{0.5})_xCo_{0.8}Fe_{0.2}O_{2+x-\delta}$ (BSCF-$x$, where x varies from 1 to 2 in increments of 0.1) (Fig. 1c). Electrochemical tests and characterization demonstrate that the hybrid $Ba_{1.5}Sr_{1.5}Co_{1.6}Fe_{0.4}O_{7-\delta}$ (C/H-BSCF, BSCF-1.5), which consists roughly 57.26 wt.% of the cubic phase C-BSCF and 42.74 wt.% of the hexagonal phase H-BSCF, has the optimal electrocatalytic activity (Fig. 1c). The hybrid electrode C/H-BSCF exhibits excellent ORR and OER activity and achieves unprecedented performance in both FC and EC modes in R-PCECs, which is found to be a result from the synergistic effect between cubic C-BSCF with excellent oxygen activation and conductivity and hexagonal H-BSCF with strong hydration reaction and abundant oxygen vacancies. Additionally, the consistent elemental composition strengthens the interaction between the two phases and enhances the air electrode's structural and chemical stability (Fig. 1d). Such C/H-BSCF oxide may present a fresh chance for quickening the adoption of R-PCECs technology.

## Results

The influence of different A-site alkali metal contents on the structure of C-BSCF was explored to provide insight into the search for superior air electrodes. The synthesized C-BSCF, C/H-BSCF, and H-BSCF were subjected to analysis using X-ray diffraction (XRD). The results show that C/H-BSCF exhibits a superposition of cubic C-BSCF and hexagonal H-BSCF diffraction peaks, which suggests that it is a self-assembled dual-phase oxide made up of cubic and hexagonal phases (Supplementary Fig. 1). The cubic and hexagonal structures of C-BSCF and H-BSCF powders, respectively, were confirmed through Rietveld refinement data (Fig. 2a, b). C-BSCF was determined to have a space group of $Pm$-$3m$ and lattice parameters of $a = b = c = 3.9865(3)$ Å ($R_{exp} = 5.19\%$, $R_{wp} = 5.61\%$, $GOF = 1.08$)[27]. H-BSCF was found to have a space group of $P6_3mc$ and lattice parameters of $a = b = 11.7115(7)$ Å, $c = 6.9075(4)$ Å ($R_{exp} = 5.53\%$, $R_{wp} = 8.81\%$, $GOF = 1.59$)[26]. In order to improve the accuracy of the XRD refinement results for the C/H-BSCF hybrid, transmission electron microscopy and energy dispersive X-ray (TEM-EDX) techniques were employed to determine the stoichiometries of the two phases (Supplementary Fig. 2). These determined values were then utilized as input parameters for the XRD refinement process. The crystal structural composition of the C/H-BSCF sample surface was revealed to be 57.26 wt.% cubic and 42.74 wt.% hexagonal through XRD refinement, where the cubic and hexagonal phases are consistent with the space group of C-BSCF and H-BSCF, respectively (Fig. 2c). The refined results were confirmed to be accurate based on reliability factors of $R_{exp} = 5.44\%$, $R_{wp} = 5.42\%$, and $GOF = 1.00$[28]. To investigate the non-accidental features of C/H-BSCF oxides for self-assembling dual phases, a series of BSCF-$x$ (where x varies from 1 to 2 in increments of 0.1) oxides were synthesized. XRD patterns show that, except for C-BSCF and H-BSCF, which are single-phase oxides, all other oxides exhibit a hybrid of cubic and hexagonal phases (Supplementary

Fig. 3)[29]. Furthermore, the XRD refinement of the oxides of BSCF-1.3 and BSCF-1.7 confirms their dual-phase composition (Supplementary Fig. 4). By combining these results with the Rietveld refinement data of C/H-BSCF, it is clear that the hexagonal phase content gradually increases and the cubic phase content decreases with the *x*-value approaches 2 (Supplementary Table 1). Thus, this strategy allows for

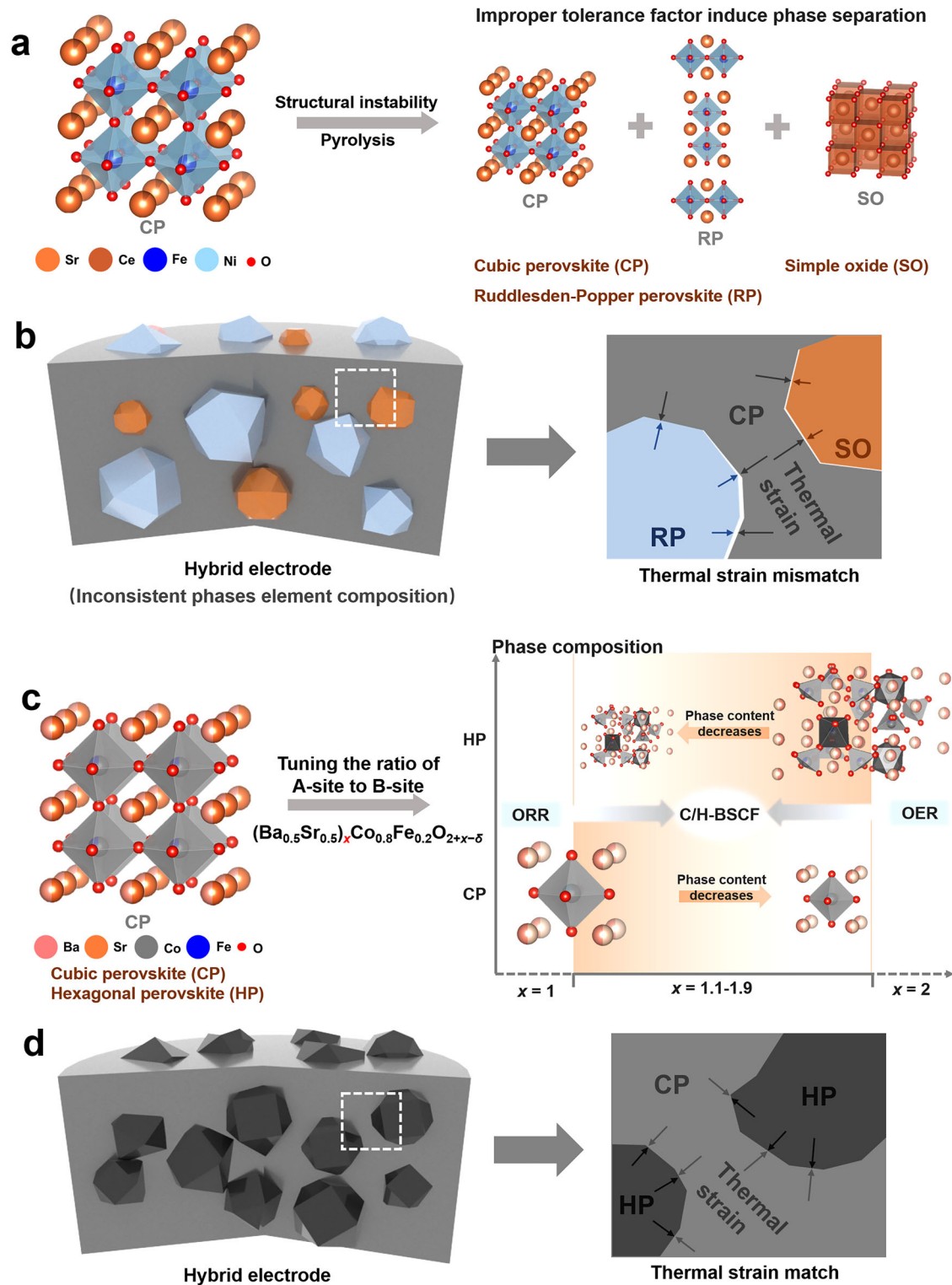

**Fig. 1 | Schematic diagram of the synthesis strategies and thermal effects of hybrid electrodes. a** The hybrid electrodes formed by structural instability and pyrolysis induce phase separation due to an improper tolerance factor of the partial cation in the bulk phase thus forming a Ruddlesden-Popper perovskite or simple oxide. **b** Mismatched thermal stress between multiple phases of hybrid electrodes synthesized by improper tolerance factor. **c** The phase content-controlled hybrid electrode composed of cubic and hexagonal perovskites induced by tuning the ratio of A-site to B-site of cubic perovskite. **d** Thermal stress match for hybrid electrodes with consistent phase element composition.

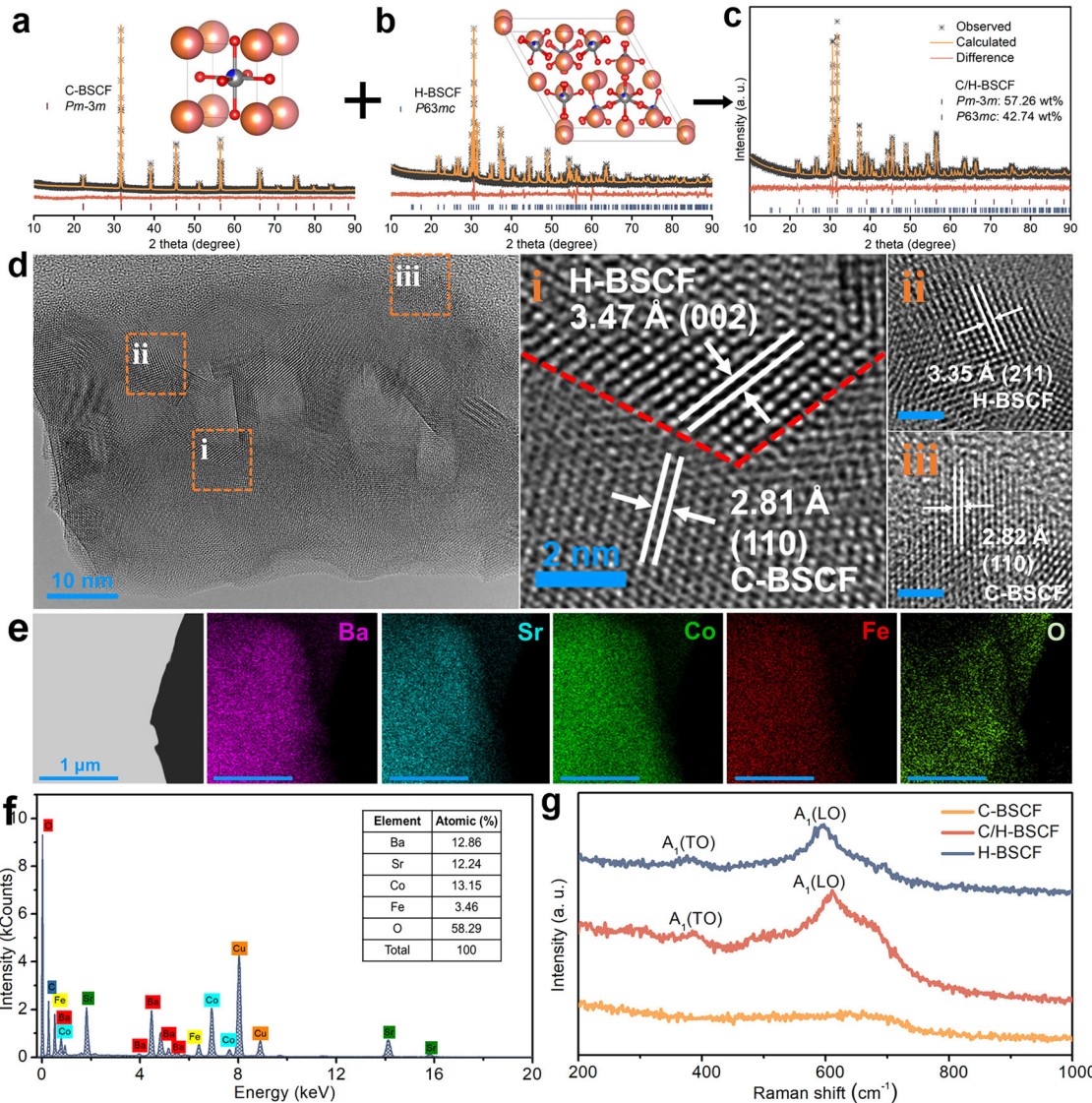

**Fig. 2 | Phase composition and crystal structure analysis of hybrid C/H-BSCF.** Refined XRD profiles of the prepared **a** C-BSCF, **b** H-BSCF, and **c** C/H-BSCF. The inset shows the structural schematics of C-BSCF and H-BSCF, respectively. Color code: red (O), pink (Ba), orange (Sr), gray (Co), blue (Fe). **d** HR-TEM image of C/H-BSCF particle, (i), (ii), and (iii) show the zoomed-in view of the corresponding positions in Fig. d, respectively (scale bar is 2 nm). **e** TEM-EDX element mapping and **f** results for C/H-BSCF. **g** Raman spectra of C-BSCF, H-BSCF, and C/H-BSCF samples.

the creation of dual-phase hybrids, as well as the intentional modulation of the content composition of both phases.

To confirm more precisely the existence of both cubic and hexagonal phases in the C/H-BSCF oxide, a high-resolution transmission electron microscopy (HR-TEM) image was collected. In Fig. 2d, a close contact between the (110) crystal plane of the cubic phase and the (002) crystal plane of the hexagonal phase is clearly seen in the frame area (i). It is evident that the (211) hexagonal crystal plane and (110) cubic crystal plane can be observed at positions (ii) and (iii), respectively. Moreover, the crystal plane spacing is consistent with the XRD Rietveld refinement data of the hybrid C/H-BSCF, further confirming the presence of these planes. In the C/H-BSCF hybrid, the strain and interfacial effects yielded by the different atomic arrangements at the interface of the C-BSCF (110) and H-BSCF (002) phases have positively contributed to the enhancement of the catalytic activity[30–32]. TEM-EDX elemental mapping images were used to determine the bulk properties of C/H-BSCF, which revealed a uniform element distribution at the micron scale, as well as the precise determination of the atomic percentage of each element, which remained in close agreement as

expected (Fig. 2e, f). However, the local EDX spectroscopy line-scan profiles of the C/H-BSCF particle in Fig. 2e display a considerable fluctuation in the elemental content of Ba and Sr, indicating the coexistence of two phases (C-BSCF and H-BSCF) in C/H-BSCF particle (Supplementary Fig. 5)[18]. Raman spectroscopy was employed to determine the crystal structure and vibrational characteristics of perovskite oxides[33,34]. Although the ideal cubic structure of perovskite is not expected to exhibit Raman active bands, C-BSCF and C/H-BSCF oxides exhibit broad bands at 675 cm⁻¹, which could be ascribed to Jahn-Teller distortion of the cubic perovskite at room temperature (Fig. 2g)[33,35]. Furthermore, the peaks observed at 380 cm⁻¹ and 574 cm⁻¹ in C/H-BSCF and H-BSCF are indicative of the $A_I$(TO) and $A_I$(LO) optical phonon modes of the *P63mc* space group[36]. Therefore, Raman spectra also verified the dual-phase composition of C/H-BSCF oxide at room temperature. Additionally, C/H-BSCF maintains a stable phase structure and composition under various temperatures and longtime calcination conditions (Supplementary Fig. 6).

The hybrid oxide C/H-BSCF represents an excellent electrocatalytic activity as an air electrode. The electrochemical impedance spectra

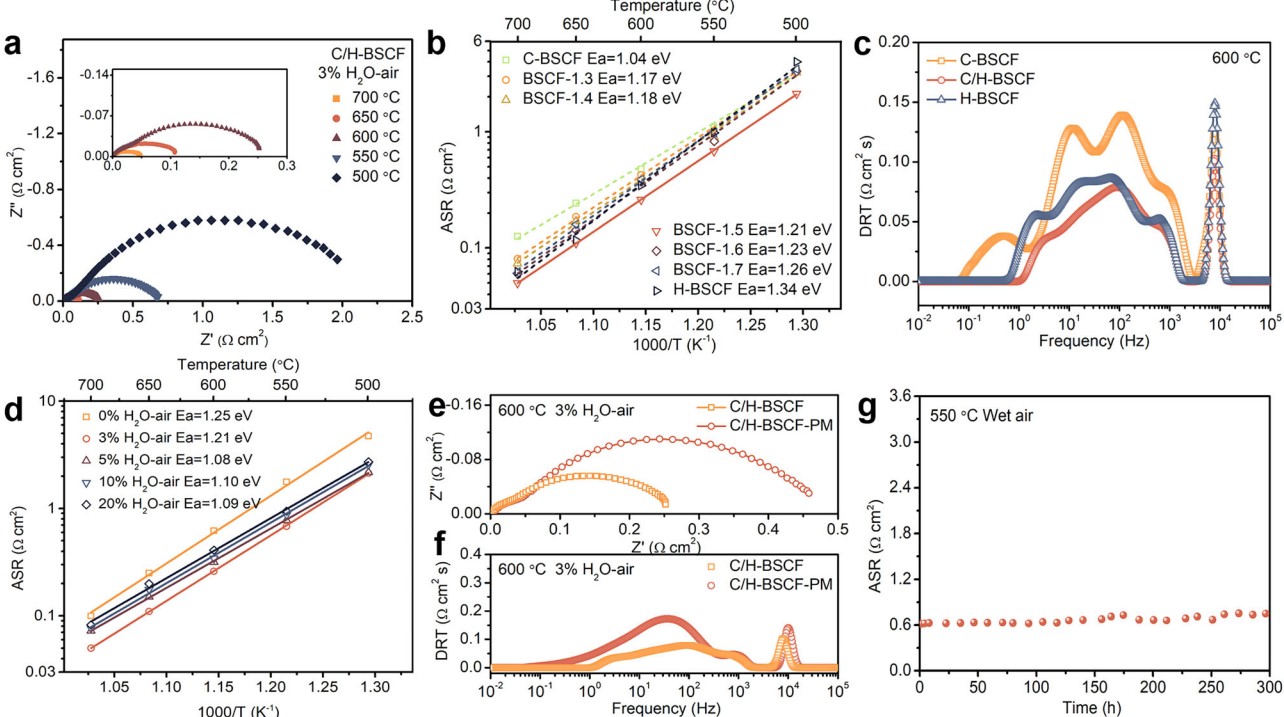

**Fig. 3 | Oxygen activation performance of air electrodes. a** EIS plots of BZCYYb-supported symmetric cell with C/H-BSCF electrode at 500–700 °C under 3% $H_2O$-air. **b** Arrhenius plot of the ASRs for C-BSCF, H-BSCF, and BSCF-$x$ electrodes ($x$ = 1.3, 1.4, 1.5, 1.6, and 1.7). **c** The DRT analysis of EIS was measured at C-BSCF, C/H-BSCF, and H-BSCF electrodes at 600 °C. **d** Arrhenius plot of the ASRs of symmetric cell with C/H-BSCF electrode at different water pressures (0, 3, 5, 10, 20% $H_2O$-air) at 500–700 °C. **e** EIS and **f** DRT analysis of self-assembled and physically mixed C/H-BSCF electrodes measured at 600 °C under 3% $H_2O$-air conditions. **g** Time dependence of ASR of symmetric cell with C/H-BSCF electrode in wet air at 550 °C.

(EIS) of the symmetric cell featuring the C/H-BSCF/BaZr$_{0.1}$Ce$_{0.7}$Y$_{0.1}$Yb$_{0.1}$O$_{3-\delta}$ (BZCYYb)/C/H-BSCF structure, were depicted in Fig. 3a and measured under 3% $H_2O$-air at 700, 650, 600, 550, and 500 °C, revealing the area specific resistance (ASR) of the C/H-BSCF hybrid electrode to be 0.05, 0.11, 0.26, 0.68, and 2.13 Ω cm², respectively. Comparatively, under identical operating conditions, the ASR of the single-phase C-BSCF and H-BSCF electrodes are 0.48 and 0.34 Ω cm² at 600 °C, respectively (Supplementary Fig. 7a, b). The results of these tests suggest a significant improvement in ASR of 45.8% and 23.5% for the dual-phase C/H-BSCF electrode relative to the single-phase oxides C-BSCF and H-BSCF (Supplementary Table 2), underscoring the potential advantages of the dual-phase coordination and enhancement for C/H-BSCF as an air electrode. Furthermore, the ORR activity of H-BSCF was found to be much lower than that of C-BSCF through EIS of C-BSCF and H-BSCF electrodes on symmetric cells using Ce$_{0.8}$Sm$_{0.2}$O$_{1.9}$ (SDC) electrolyte (Supplementary Fig. 8), while the good electrochemical performance of H-BSCF on proton-conducting symmetric cells may be attributed to the excellent hydration ability. Compared to the typical self-assembled air electrode, the capacity to modulate the composition of the two phases to optimize the catalytic activity of the electrode has higher research value and applicability. EIS of the various electrodes with varying cubic and hexagonal phase contents are obtained as well (Supplementary Fig. 7), revealing that the C/H-BSCF hybrid has the lowest ASR in the operating temperature range. Notably, as the hexagonal phase content increases, the ASR decreases and then increases, while the activation energy gradually increases (Fig. 3b). This pattern may reflect interactions between the cubic and hexagonal phases in oxygen activation and hydration reactions under varying temperatures.

The distribution of relaxation times (DRT) is a widely used approach for EIS deconvolution in the intensive analysis of electrochemical reaction processes on electrode surfaces[37]. Figure 3c depicts the deconvolution peaks of the EIS for C-BSCF, C/H-BSCF, and H-BSCF electrodes at 600 °C by DRT treatment. According to the frequency distribution, the deconvolution peaks are categorized into three distinct regions: high frequency (HF, >2 × 10³ Hz), intermediate frequency (IF, 2 × 10³–10 Hz), and low frequency (LF, <10 Hz). These regions are related to the processes of charge transfer, ion migration, and surface exchange, as well as surface mass transfer, respectively[20,38]. As depicted in Fig. 3c, compared to the LF and HF regions, the resistance in the IF region primarily governs the total resistance, indicating that the surface exchange and bulk diffusion rates of the air electrode, specifically the surface adsorption and dissociation of oxygen and water, as well as the bulk diffusion rates of oxygen ions and protons, significantly limit the electrocatalytic process. Interestingly, the peak area of the hybrid electrode C/H-BSCF in the IF region is not between, but lower than that of the single-phase C-BSCF and H-BSCF electrodes. This not only demonstrates the excellent surface exchange and bulk diffusion rates of the C/H-BSCF electrode but also highlights the advantages of the synergistic effect from the dual phase in the ORR and hydration reaction on the air electrode surface. Meanwhile, the temperature dependence of ASR for C-BSCF, C/H-BSCF, and H-BSCF electrodes in the HF, IF, and LF regions is illustrated in Supplementary Fig. 9. The activation energy of the C/H-BSCF electrode in the IF region is 1.24 eV, which is in the middle of the activation energies of C-BSCF and H-BSCF. This is mainly due to the synergistic promotion of ionic conduction in the cubic and hexagonal phases in the C/H-BSCF hybrid electrode[39].

To understand and compare the surface reactions of C-BSCF, C/H-BSCF, and H-BSCF air electrodes under practical conditions, the EIS of symmetric cells with different air electrodes at different partial pressures of oxygen ($P_{O2}$) were tested and analyzed in conjunction with DRT for different processes (Supplementary Fig. 10). When the air electrode was exposed to pure oxygen at 600 °C, as expected, the

symmetric cell demonstrated a relatively low polarization resistance ($R_p$). As the $P_{O2}$ decreased, the $R_p$ showed a significant increase. The C/H-BSCF hybrid electrode exhibits the lowest $R_p$ at different oxygen partial pressures at 600 °C. However, in the HF region, the C-BSCF, C/H-BSCF, and H-BSCF electrodes show similar $R_p$ and $n$ values of about 0.2, indicating that the charge transfer processes of different electrodes do not significantly discrepancy[9,10]. The lower $R_p$ of C/H-BSCF electrode is mainly attributed to the decrease in resistance of the IF and LF regions compared to C-BSCF and H-BSCF electrodes, which further suggests that the fast ORR rate of the C/H-BSCF hybrid electrode is mainly attributed to the facilitated diffusion rate of the oxygen ions and the surface mass transfer by the cubic and hexagonal phases, respectively. Moreover, the symmetric cell with C/H-BSCF electrode is evaluated for EIS at different water pressures in order to further demonstrate that the C/H-BSCF hybrid electrode has a strong hydration capacity and proton conductivity (Fig. 3d). The ASR of C/H-BSCF dramatically lowered under 3% $H_2O$-air compared to dry air conditions, for example, falling of 58% from 0.62 to 0.26 Ω cm$^2$ at 600 °C. This significant reduction in ASR indicates that the C/H-BSCF electrode has a fast hydration reaction rate and excellent proton diffusion capability. However, beyond 3% $H_2O$-air, it is worth noting that the ASR at 700 °C increased more prominently compared to 500 °C. This can be attributed to the increased water vapor pressure leading to competition between oxygen and water molecules for adsorption on the electrode surface, thus lowering the ORR rate of the electrode. Additionally, the activation energy gradually decreased as the water pressure increased, suggesting that the C/H-BSCF hybrid electrode has the potential to be utilized as an air electrode in proton-conducting electrochemical cells[40]. In addition to this, the physically mixed C/H-BSCF electrode (C/H-BSCF-PM) is fabricated via the mechanical mixing of cubic C-BSCF and hexagonal H-BSCF, according to the two-phase ratio of the hybrid C/H-BSCF. The EIS of this electrode is measured on a symmetric cell with ASRs of 0.08, 0.18, 0.47, 1.20, and 3.56 Ω cm$^2$ at 3% $H_2O$-air at 700, 650, 600, 550, and 500 °C, respectively (Supplementary Fig. 11). The self-assembled C/H-BSCF electrode exhibited superior catalytic activity compared to the C/H-BSCF-PM electrode. The primary reason for this is attributed to the fact that mechanical mixing results in a poorer distribution and interface of the two phases, which can negatively impact ion transport and surface diffusion, thereby constraining the oxygen activation rate of the electrode (Fig. 3e and Supplementary Fig. 12)[17,21]. This is confirmed by the DRT analysis of the EIS of both electrodes (Fig. 3f and Supplementary Fig. 13). As demonstrated in Fig. 3g, the symmetric cell with the C/H-BSCF electrode exhibited good stability over extended periods of testing, with a decay rate of merely $4.50 \times 10^{-4}$ Ω cm$^2$ h$^{-1}$ for the catalytic activity.

Previously, we utilized EIS to examine the oxygen activation performance of electrodes with cubic phase, hexagonal phase, and multiple dual-phase hybridizations. To delve deeper into the underlying reasons for the discrepancies in catalytic activity between C-BSCF, C/H-BSCF, and H-BSCF and to reveal the dual-phase hybrid effect on the promotion of ORR and OER at the air electrode, it was characterized and tested by X-ray absorption near edge structure (XANES), X-ray photoelectron spectroscopy (XPS), thermogravimetric analysis (TGA), $O_2$/$H_2O$ temperature-programmed desorption ($O_2$/$H_2O$-TPD), and hydrogen-permeable membranes. The absorption edge (at 0.7–0.8 normalized intensity) in the XANES spectra at the $K$-edge of $3d$ transition elements are remarkably sensitive to their valence state[41,42]. The XANES spectra for the Co $K$-edge and Fe $K$-edge of the C-BSCF, C/H-BSCF, and H-BSCF catalysts were shown in Fig. 4a, b, in which one can see that the H-BSCF located at the higher energy indicating higher Co and Fe valences in BSCF than those in the hybrid oxide C/H-BSCF. This suggests that the hexagonal phase possesses a higher B-site average valence, which may be responsible for the remarkable thermal reduction of H-BSCF at high temperatures that creates oxygen vacancies and promotes hydration, as further verified later[20,26,43]. At

room temperature, cubic C-BSCF has the lowest Co and Fe valences, which implies the presence of more oxygen vacancies and facilitates the acceleration of the ORR rate at the air electrode surface[44,45]. Furthermore, XPS analysis was also used to reveal the oxidation states of Co and Fe in different samples. The average valence states of Co and Fe from high to low are H-BSCF, C/H-BSCF and C-BSCF oxides, respectively, which are compatible with the results of XANES analysis (Supplementary Fig. 14). Figure 4c presents the XPS spectra of the oxygen species on the catalyst surface and the corresponding deconvolution peaks. The oxygen activation capacity of catalysts can be characterized by the ratio of lattice oxygen ($O_{lat}$) to adsorbed oxygen ($O_{ads}$). According to the subpeak areas analysis, the values of $O_{lat}$/$O_{ads}$ for C-BSCF, C/H-BSCF, and H-BSCF are 18.81%, 5.03%, and 4.66%, respectively. The hexagonal H-BSCF shows lower $O_{lat}$/$O_{ads}$ values, which is mainly attributed to the abundant BaO nanoparticles on the catalyst surface that enhance oxygen adsorption and speed up the oxygen activation process (Supplementary Fig. 15)[46–48].

High-temperature induced loss of lattice oxygen is one of the crucial characteristics that reflects the formation of oxygen vacancies and the oxygen ion's ability to migrate in the bulk phase of the air electrode. Furthermore, oxygen vacancies, which can serve as active sites in various electrochemical reactions such as ORR, OER, and hydration, have a noteworthy impact on the overall kinetic rate of the electrode[40,49,50]. Figure 4d clearly depicts the TGA curves of the C-BSCF, C/H-BSCF, and H-BSCF from room temperature to 1000 °C. The results indicate a mass loss of 2.2%, 6.7%, and 10% for the three samples, respectively. This is due to the thermal reduction that lowered the Co and Fe valence states, releasing more lattice oxygen. The dual-phase hybrid C/H-BSCF has an impressive oxygen loss, outperforming many advanced air electrodes, such as $BaCo_{0.4}Fe_{0.4}Zr_{0.1}Y_{0.1}O_{3-\delta}$ (BCFZY, ~1%), BCFZYN (~1.1%), PBSCF (~1.2%), BCCY (~1.2%), and $Ba_2Co_{1.5}Mo_{0.25}Nb_{0.25}O_{6-\delta}$ (BC1.5MN, ~0.7%)[9,18,20,51,52]. $O_2$-TPD tests were used to investigate the characteristics of the oxygen desorption process with increasing temperature for different oxides. The initial desorption temperature and amount of oxygen desorption are critical indications of oxygen activation and oxygen vacancy content of the air electrodes under operating conditions[22]. The hybrid C/H-BSCF has a lower initial desorption temperature and higher resolved oxygen content compared to the cubic C-BSCF (Fig. 4e). In addition, the amounts of oxygen desorption identified by $O_2$-TPD for each material are consistent with the TGA results. The $D_{chem}$ and $k_{chem}$ values of C-BSCF, C/H-BSCF, C/H-BSCF-PM, and H-BSCF in the temperature range of 500 to 700 °C, determined using the conductivity relaxation method (Supplementary Fig. 16), are depicted in Fig. 4f and Supplementary Fig. 19[53]. All samples were calcined to dense bars and the phase structure and composition remained stable (Supplementary Figs. 17 and 18). As anticipated, at each temperature, both $D_{chem}$ and $k_{chem}$ of C/H-BSCF exhibit significantly higher values compared to C-BSCF, C/H-BSCF-PM, and H-BSCF. For instance, at 700 °C, the $D_{chem}$ and $k_{chem}$ values of the C/H-BSCF oxides were measured to be $4.88 \times 10^{-4}$ cm$^2$ s$^{-1}$ and $4.69 \times 10^{-3}$ cm s$^{-1}$, respectively, whereas corresponding values for C-BSCF were $2.27 \times 10^{-4}$ cm$^2$ s$^{-1}$ and $2.51 \times 10^{-3}$ cm s$^{-1}$. On the other hand, H-BSCF exhibited the lowest $D_{chem}$ and $k_{chem}$ values, which were $4.25 \times 10^{-5}$ cm$^2$ s$^{-1}$ and $4.15 \times 10^{-4}$ cm s$^{-1}$, respectively. Even though the $D_{chem}$ and $k_{chem}$ values of C-BSCF were higher than those of H-BSCF at 700 °C, the oxygen surface exchange and bulk phase diffusion rates of H-BSCF were noticeably superior to C-BSCF when the temperature dropped to 550 and 500 °C. Additionally, the $D_{chem}$ and $k_{chem}$ values of the C/H-BSCF-PM composite oxide were also explored under the same conditions as depicted in Fig. 4f and Supplementary Fig. 19. Throughout the tested temperature range, the $D_{chem}$ and $k_{chem}$ values of C/H-BSCF-PM were lower than those of self-assembled synthesized C/H-BSCF (Supplementary Table 3). This discrepancy can be attributed to the inferior phase distribution of the

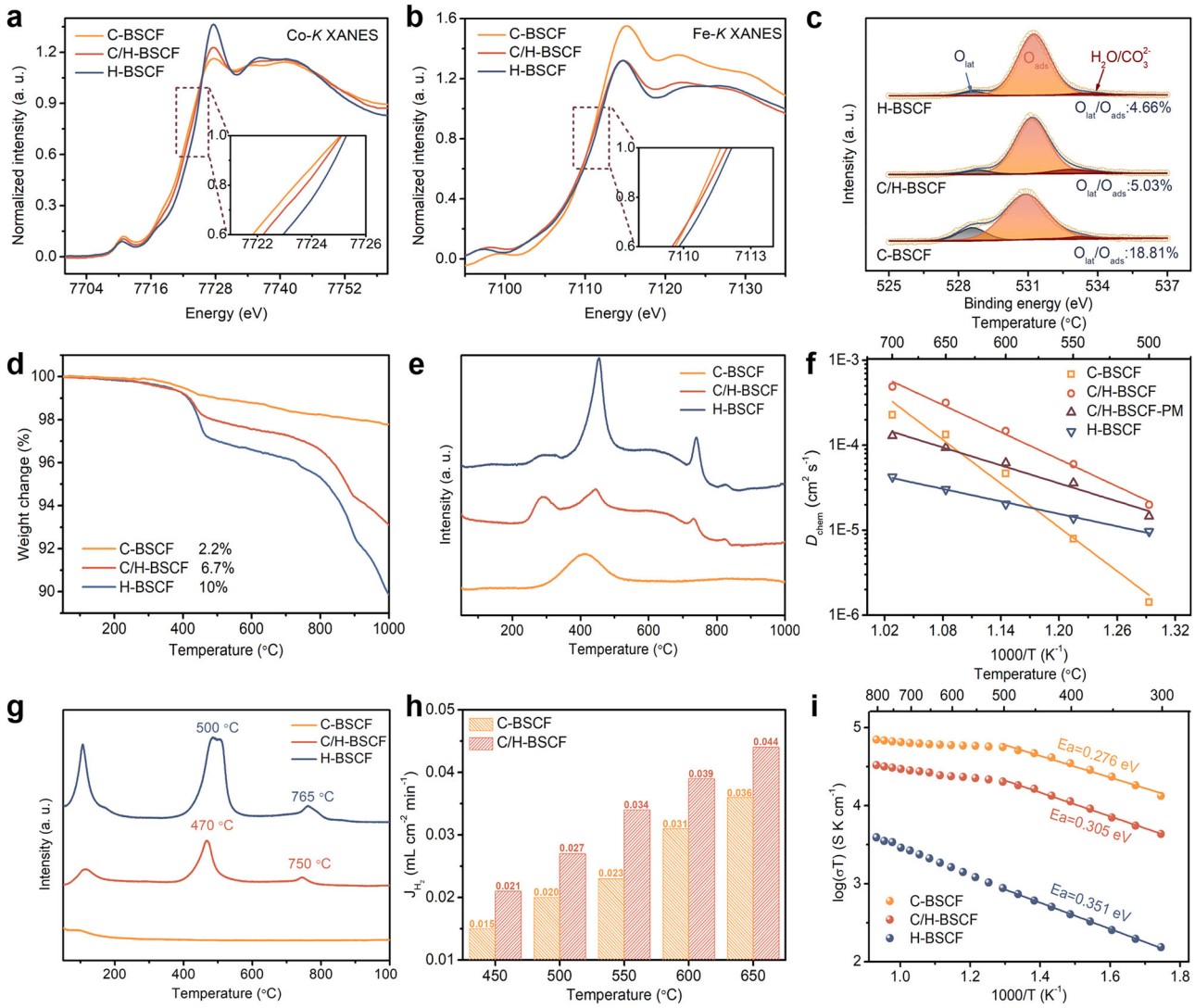

**Fig. 4 | Enhanced oxygen activation, ion conductivity, and hydration reaction.** **a** Co and **b** Fe $K$-edge XANES spectra for C-BSCF, C/H-BSCF, and H-BSCF. The inset shows localized enlargement. **c** XPS spectra of C-BSCF, C/H-BSCF, and H-BSCF, showing the O 1$s$ structures. **d** TGA curves and **e** $O_2$-TPD profiles from room temperature to 1000 °C. **f** $D_{chem}$ and $k_{chem}$ fitted by the electron conductivity relaxation curves of the electrodes in the temperature range of 500–700 °C. **g** $H_2O$-TPD profiles from room temperature to 1000 °C. **h** The $H_2$ permeation fluxes of C-BSCF and C/H-BSCF hydrogen-permeable membranes at 450–650 °C. **i** Electrical conductivities of C-BSCF, C/H-BSCF, and H-BSCF between 300–800 °C in air.

physically mixed samples in C/H-BSCF-PM, which hindered the synergistic effect of the two phases and subsequently hindered ion migration and surface exchange.

The hydration reaction, as a prerequisite for the introduction of protons into the air electrode, is a determinant for both proton conduction and catalytic reaction kinetics[54]. Protons, existing as hydroxide ion defects within the perovskite oxide, undergo bulk phase migration through the Grotthuss mechanism[55,56]. In Fig. 4g, $H_2O$-TPD was utilized to explore the hydration capacity of C-BSCF, C/H-BSCF, and H-BSCF samples. Besides the desorption peak of adsorbed water appearing before 200 °C, C/H-BSCF and H-BSCF also show remarkable $H_2O$ desorption peaks compared to C-BSCF, which is attributed to the contribution of significant presence of oxygen vacancies in the hexagonal phase, significantly enhancing the hydration capacity. The dense hydrogen-permeable membranes were used to test the proton diffusion ability of the hybrid C/H-BSCF and cubic C-BSCF[18,57], as illustrated in Fig. 4h. To ensure reliable hydrogen permeation results, both sides of the membrane were coated with a 500 nm-thick Pd film using magnetron sputtering (Supplementary Fig. 20)[58]. The dual-phase C/H-BSCF hydrogen permeation membrane has a larger $H_2$ permeation flux

at 450–650 °C, indicating that the hybrid C/H-BSCF has superior proton diffusion performance. For example, the hydrogen permeation flux of C/H-BSCF was 0.039 mL cm⁻² min⁻¹ at 600 °C, whereas the flux of C-BSCF exhibited only 0.031 mL cm⁻² min⁻¹. Meanwhile, the corresponding proton conductivities of C/H-BSCF and C-BSCF were also calculated as shown in Supplementary Fig. 21. The quick electrochemical reactions of the electrode, such as the OER process, need the electrode to have adequate conductivity to match the fast dynamic response in addition to oxygen ion and proton conduction. The hexagonal H-BSCF (0.3–3.6 S cm⁻¹) had weak conductivity at 300–800 °C, while the hybrid C/H-BSCF (5.9–30.9 S cm⁻¹) is second only to the cubic C-BSCF (23.2–65.3 S cm⁻¹) (Fig. 4i)[53]. In addition, all samples exhibit thermally activated semiconductor behavior in the range of 300–800 °C. The activation energies of C-BSCF, C/H-BSCF, and H-BSCF are 0.276, 0.305, and 0.351 eV in the range of 300–500 °C, respectively. It is noteworthy that a turning point occurs around 500 °C, suggesting different conductive mechanisms in the 300–500 °C and 500–800 °C ranges. This is mainly attributed to the fact that the conductive mechanism is closely related to small polaron hopping behavior, which is affected by the combination of the oxygen

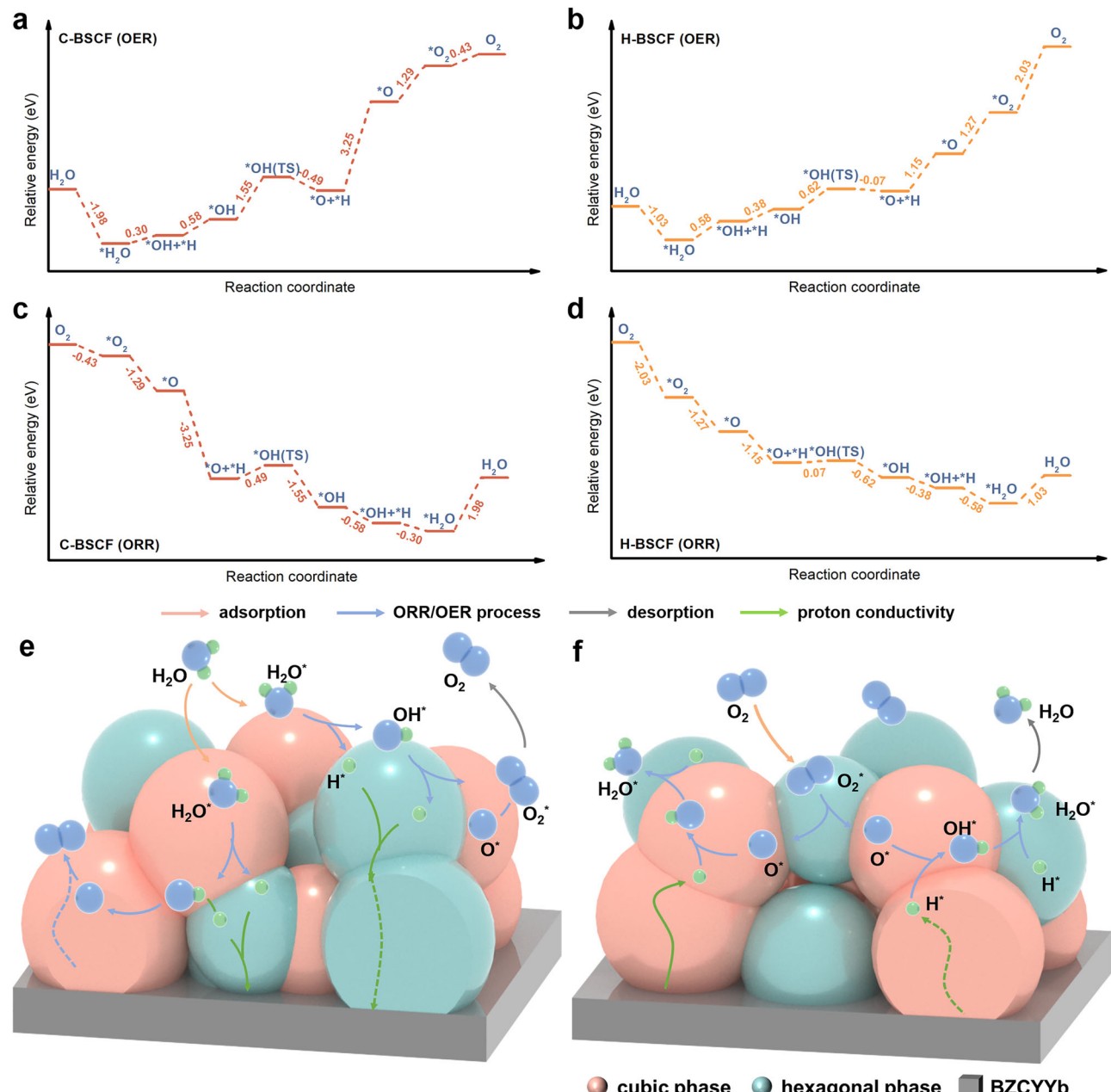

**Fig. 5 | DFT calculations regarding the ORR and OER mechanisms on the air electrodes of R-PCECs.** The energy barrier of the OER process on the **a** cubic C-BSCF (100) and **b** hexagonal H-BSCF (0001) surfaces. The energy barrier of the ORR process on the **c** cubic C-BSCF (100) and **d** hexagonal H-BSCF (0001) surfaces. Catalytic mechanism of the dual-phase synergistic **e** OER and **f** ORR on the surface of the hybrid C/H-BSCF.

release and the carrier migration rate. Overall, the dual-phase hybrid electrode C/H-BSCF is confirmed to have superior triple conductivity ($e^-$/$O^{2-}$/$H^+$) and hydration capacity in the synergistic effect of the hexagonal and cubic perovskite phases, revealing the origin of its high oxygen activation.

We investigated the thermal expansion coefficients (TEC) of various air electrodes in order to confirm that C/H-BSCF, a self-assembled composite electrode with two phases having the same elemental composition, can significantly reduce the thermal stress to improve the match with the electrolyte. Supplementary Fig. 22a shows the linear expansion (dL/L$_0$) of the C-BSCF, C/H-BSCF, C/H-BSCF-PM and H-BSCF samples with temperature. the TEC of C-BSCF and H-BSCF were $23.8 \times 10^{-6}\,K^{-1}$ and $19.2 \times 10^{-6}\,K^{-1}$, respectively, where the TEC of cubic C-BSCF is consistent with literature reports[18], while hexagonal H-BSCF exhibits a lower TEC than C-BSCF. As expected, the TEC of the

C/H-BSCF and C/H-BSCF-PM hybrids, which were $19.7 \times 10^{-6}\,K^{-1}$ and $20.4 \times 10^{-6}\,K^{-1}$, respectively, were intermediate between those of C-BSCF and H-BSCF. The hybrid electrode shows better thermal matching with the electrolyte compared to the majority of conventional Co/Fe-based electrodes (Supplementary Fig. 22b, c)[9,59–66]. It is worth noting that numerous composite electrodes prepared by physical mixing and self-assembly methods exhibit significant differences in terms of their two-phase structures and elemental compositions. These differences hinder the composite electrodes from effectively regulating thermal stresses among the phases at high temperatures. Therefore, the TEC of conventional composite electrodes is directly dependent on the TEC of each phase and the proportion of each phase within the composite[61,65,67]. Interestingly, the TEC of C/H-BSCF is found to be closer to the TEC of H-BSCF. This suggests that a C/H-BSCF hybrid electrode, which possesses the same elemental composition as

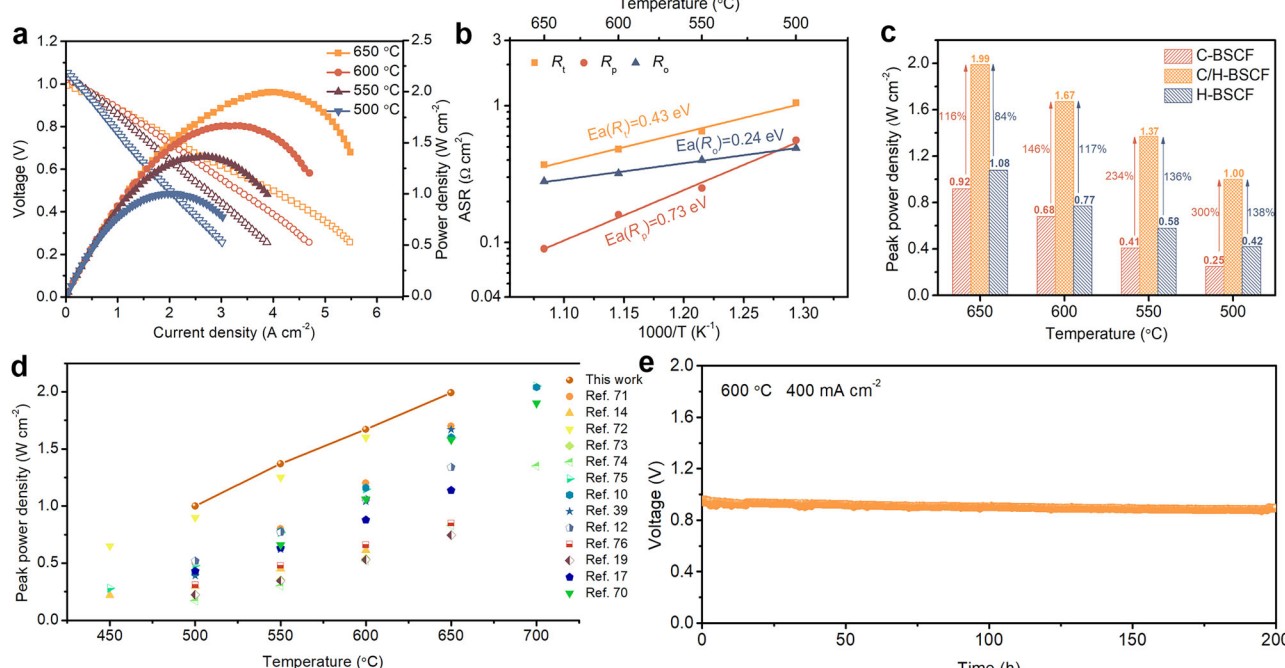

**Fig. 6 | Electrochemical performance and stability of single cells in FC mode.**
**a** I–V and I–P curves of an anode-supported single cell with C/H-BSCF air electrode in FC mode. **b** Arrhenius plots of $R_t$, $R_p$, and $R_o$ for R-PCECs under open-circuit voltage conditions. **c** Comparison of PPDs at different temperatures for single cells with C-BSCF, C/H-BSCF, and H-BSCF electrodes. **d** Comparison of PPDs obtained with C/H-BSCF electrode and advanced electrodes in the literature at different temperatures. **e** Stability test of a single cell with C/H-BSCF air electrode in FC mode.

the two phases, can effectively modulate thermal stress at high temperatures, consequently reducing the overall TEC of the hybrid electrode. To better understand this behavior, we conducted Rietveld refinement on the XRD analysis of the C/H-BSCF hybrid at 700 °C. The results reveal that at 700 °C, the weight ratios of the cubic and hexagonal phases of C/H-BSCF were found to be 48.82 wt.% and 51.18 wt.%, respectively, as shown in Supplementary Fig. 22d. The C/H-BSCF hybrid electrode achieves favorable thermal matching at high temperatures by autonomously transitioning from the high-TEC cubic phase to the lower-TEC hexagonal phase.

The energy barriers of ORR and OER kinetic processes on cubic C-BSCF and hexagonal H-BSCF surfaces were studied using density functional theory (DFT). First, the energies of C-BSCF and H-BSCF with different A- and B-site arrangement configurations were calculated to choose the most reasonable models with the lowest total energy for subsequent calculations of the ORR and OER pathways (Supplementary Figs. 23 and 24; and Supplementary Tables 4 and 5). In addition, we selected stable low-index surfaces of the Co, Fe-terminated C-BSCF (100), and H-BSCF (0001) for DFT calculations, respectively[26,68]. Before using Co, Fe-terminated slab models, we calculated the surface energies of the different surface terminations to obtain the most stable surface atomic configurations (Supplementary Fig. 25 and Supplementary Table 6). To more accurately model the OER process on the surfaces of C-BSCF and H-BSCF, the Fe-top site on the electrode surface with lower $H_2O$ adsorption energy was chosen as the active site for both models (Supplementary Fig. 26 and Supplementary Table 7). Figure 5a, b show the energy profiles of a reasonable reaction pathway in order to fully investigate the OER process on C-BSCF and H-BSCF, respectively. The OER process on the air electrode surface is divided into several steps: $H_2O$ adsorption (*$H_2O$, * indicates the species' adsorbed state), oxygen evolution (*$H_2O$ → *OH + *H → *O + *H → *O → *$O_2$), and *$O_2$ desorption, whereas the ORR process is the inverse of the OER process[69,70]. Notably, the maximum energy barrier to be crossed by C-BSCF in the OER pathway is 3.25 eV, and the reaction process is *O + *H → *O, indicating that proton conduction is the main factor

restricting the OER rate of the C-BSCF electrode. In contrast, H-BSCF only requires crossing 1.15 eV in this process, which further validates the superior proton migration capability of hexagonal H-BSCF. Similarly, the favorable proton migration of H-BSCF is also demonstrated in the *OH + *H → *OH process. For H-BSCF, O desorption becomes the major hindrance for the OER process, needing to overcome an energy barrier of 2.03 eV, whereas C-BSCF exhibits more advantages in terms of oxygen desorption and water adsorption compared to H-BSCF. Hence, it is evident that the combination of cubic and hexagonal phases in C/H-BSCF holds more advantages in the OER process than single-phase C-BSCF and H-BSCF. Based on the OER energy profiles of cubic C-BSCF and hexagonal H-BSCF, it is speculated that for the OER process on the surface of hybrid electrode C/H-BSCF, $H_2O$ will be preferentially adsorbed on the surface of cubic phase and undergo initial decomposition (*$H_2O$ → *OH + *H), followed by further decomposition of *OH on the surface of hexagonal phase and final completion of oxygen desorption (Fig. 5e). Furthermore, as the inverse process of OER, the energy profiles of the ORR processes of the cubic C-BSCF and hexagonal H-BSCF electrodes are illustrated in Fig. 5c, d, respectively. C-BSCF and H-BSCF have oxygen adsorption energies of −0.43 and −2.03 eV, indicating that the oxygen adsorption processes are both spontaneous. However, the largest energy barrier to overcome for the ORR process in the case of non-spontaneous water desorption is 1.98 and 1.03 eV, respectively, for the C-BSCF and H-BSCF electrodes. The energy barrier crossed by the cubic C-BSCF electrode from *$O_2$ to *$H_2O$ is −6.48 eV, but the energy consumed by the H-BSCF electrode for the same reaction process is −3.93 eV, indicating that the cubic C-BSCF electrode has a significant advantage in the reduction of oxygen. As a consequence, based on the ORR process of cubic C-BSCF and hexagonal H-BSCF, a schematic representation of the ORR process occurring on the surface of the hybrid air electrode C/H-BSCF is depicted in Fig. 5f. Hence, the hybrid air electrode C/H-BSCF exhibits exceptional ORR and OER activities, which can be attributed to the synergistic effect arising from the combination of the cubic and hexagonal phases, making these activities highly predictable.

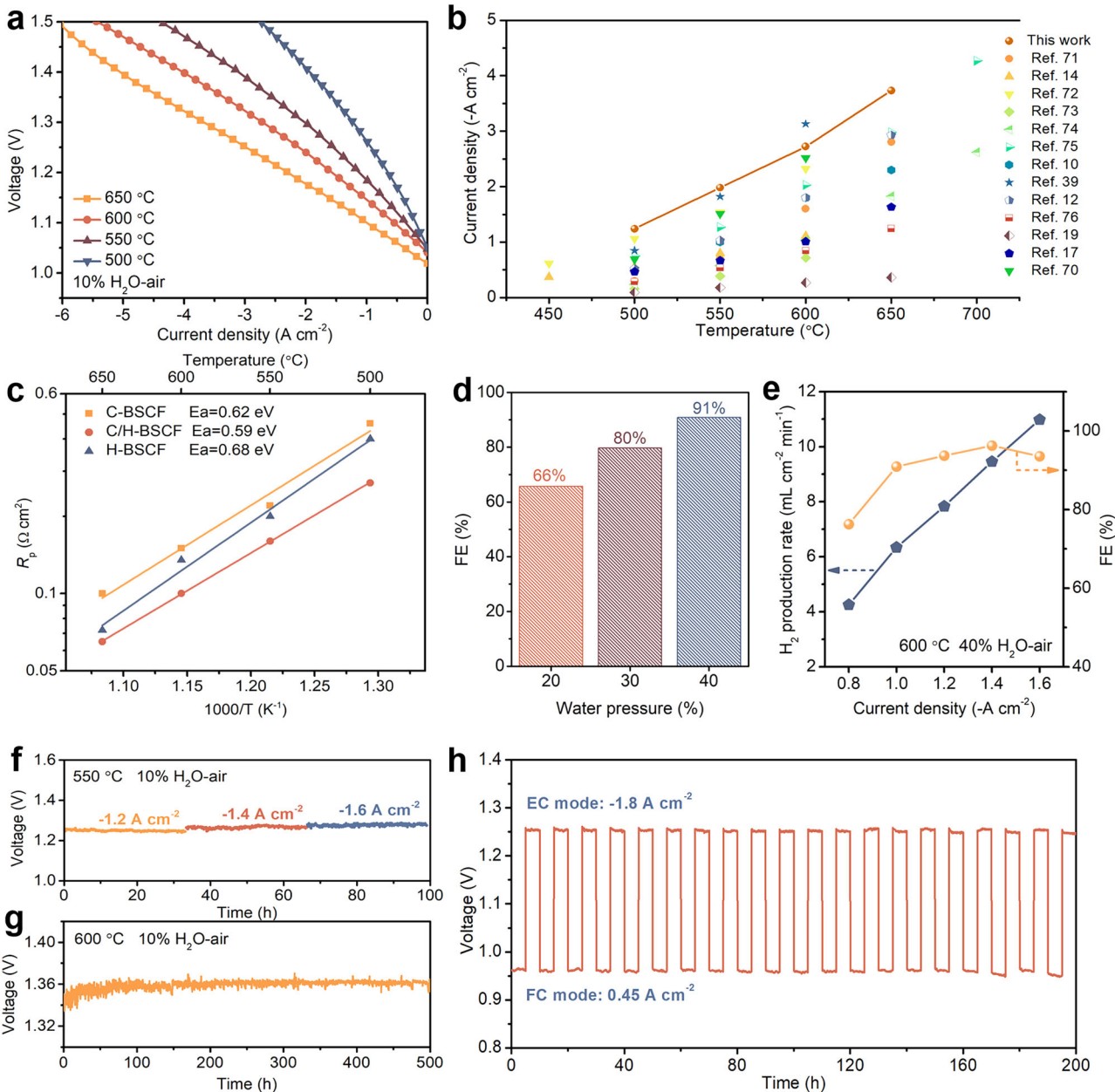

**Fig. 7 | Electrolysis performance, Faraday efficiency, and tolerance of single cells in EC mode. a** $I$–$V$ curves of a single cell with C/H-BSCF air electrode in EC mode at 500–650 °C. **b** Comparison of current density of C/H-BSCF electrode and state-of-the-art electrodes at different temperatures. **c** The $R_p$ was measured for single cells with C-BSCF, C/H-BSCF, and H-BSCF electrodes at 1.3 V in EC mode. **d** FE of the single cell with C/H-BSCF electrodes measured at 600 °C and −1 A cm$^{-2}$ for different water pressures (20%, 30% and 40% $H_2$O-air). **e** Hydrogen production rate and FE of cells at different current densities. **f** The tolerance of cell with C/H-BSCF electrode at different current densities and **g** long-term stability tests in EC mode. **h** The R-PCEC with C/H-BSCF air electrode performed cyclic tests in FC and EC modes.

An anode-supported single cell with the structure Ni-BZCYYb/ BZCYYb/C/H-BSCF was prepared to evaluate the performance of the C/ H-BSCF electrode on R-PCECs with dry hydrogen (80 mL min$^{-1}$) as the proton source for the anode and flowing wet air (10% $H_2$O-air, 100 mL min$^{-1}$) as the oxygen source for the air electrode. Supplementary Fig. 27 showcases the cross-section of a single cell, which incorporates a thin electrolyte layer created using a spin-coating process. This single cell reveals the superior electrochemical activity of the hybrid electrode C/H-BSCF. Notably, at 650, 600, 550, and 500 °C, the peak power densities (PPDs) are 1.99, 1.67, 1.37, and 1.00 W cm$^{-2}$, respectively (Fig. 6a). The excellent performance of the cell with C/H-BSCF electrode is mainly attributed to the smaller electrochemical impedance (Supplementary Fig. 28). The total resistance ($R_t$), $R_p$, and

ohmic resistance ($R_o$) dependency with temperature corresponding to the EIS of this cell at open-circuit voltage (OCV) were summarized in Fig. 6b. At 650 °C, $R_p$ is only 0.09 Ω cm$^2$, and the major component of its $R_t$ is the bigger $R_o$ (0.28 Ω cm$^2$). With a decrease in temperature to 500 °C, $R_p$ and $R_o$ increase to 0.56 and 0.49 Ω cm$^2$, respectively. It should be noted that the resistance of the cell in actual operation should be even lower than that obtained under OCV conditions.

The performances and EIS of cubic C-BSCF and hexagonal H-BSCF as air electrodes for R-PCECs in FC mode were also measured in Supplementary Fig. 29. The single cells equipped with the C-BSCF and H-BSCF electrodes achieve PPDs of 0.92 and 1.08 W cm$^{-2}$, respectively, at 650 °C, whereas the hybrid electrode C/H-BSCF outperforms them by 116% and 84%, respectively (Fig. 6c). Surprisingly, the performance

of the C/H·BSCF electrode greatly enhances at lower temperatures compared to the C·BSCF and H·BSCF electrodes, with improvements of 300% and 138% over the former two electrodes in FC mode at 500 °C, respectively. In order to clearly analyze the reason for the significant increase in performance at lower temperatures, the temperature dependence of $R_p$ of different electrodes is probed and the results showed that the hybrid C/H·BSCF has the lowest activation energy (Supplementary Fig. 30). Furthermore, despite the catalytic activity of the physically mixed C/H·BSCF-PM oxide is slightly lower than that of the hybrid C/H·BSCF, C/H·BSCF-PM on R-PCECs still exhibits commendable performance in FC mode with PPDs of 1.47, 0.94, 0.68, and 0.45 W cm$^{-2}$ at 650, 600, 550, and 500 °C, respectively (Supplementary Fig. 31). We compare the performance of the C/H·BSCF electrode with the cutting-edge electrodes reported in the literature in the same FC mode, and it is apparent that the C/H·BSCF air electrode is still in the forefront (Fig. 6d)[12,14,17,19,70–76]. Figure 6e depicts the stability assessment conducted on a single cell featuring a C/H·BSCF electrode in FC mode. The results indicated that the system operated steadily at 400 mA cm$^{-2}$ for 200 h at 600 °C, showcasing the electrode's excellent catalytic stability.

The superb OER activity and tolerance of the hybrid C/H·BSCF are further verified by electrolysis, Faraday efficiency, stability, and dual-mode cycling tests on single cells. Figure 7a depicts the I–V curves of the cell with C/H·BSCF electrode in EC mode, with current densities of −3.73, −2.72, −1.98, and −1.24 A cm$^{-2}$ at 650, 600, 550, and 500 °C under 10% H$_2$O-air, respectively, at 1.3 V applied voltage. C/H·BSCF exhibits encouraging performance as an air electrode in EC mode, surpassing numerous previously reported air electrodes, e.g., PrNi$_{0.5}$Co$_{0.5}$O$_{3-\delta}$ (PNC), Y$_{0.8}$Er$_{0.2}$BaCo$_{3.2}$Ga$_{0.8}$O$_{7+\delta}$ (YEBCG), Gd$_{0.3}$Ca$_{2.7}$Co$_{3.82}$Cu$_{0.18}$O$_{9-\delta}$ (GCCCO), La$_{0.6}$Sr$_{0.4}$Co$_{0.2}$Fe$_{0.8}$O$_{3-}$BaCoO$_{3-\delta}$ (LSCF-BC), (La$_{0.6}$Sr$_{0.4}$)$_{0.95}$Co$_{0.2}$Fe$_{0.8}$O$_{3-\delta}$-Pr$_{1-x}$Ba$_x$CoO$_{3-\delta}$, Ba(Co$_{0.4}$Fe$_{0.4}$Zr$_{0.1}$Y$_{0.1}$)$_{0.95}$Mg$_{0.05}$O$_{3-\delta}$ (BCFZYM), SCFN, PrBa$_{0.8}$Ca$_{0.2}$Co$_2$O$_{5+\delta}$-BaCoO$_{3-\delta}$ (PBCC-BCO) (Fig. 7b). Besides, it is also superior to Ba$_{0.9}$Co$_{0.7}$Fe$_{0.2}$Nb$_{0.1}$O$_{3-\delta}$ (BCFN) developed by Pei et al. (−2.8 A cm$^{-2}$ at 650 °C), PrNi$_{0.7}$Co$_{0.3}$O$_{3-\delta}$ reported by Bian et al. (−2.32 A cm$^{-2}$ at 600 °C), and the well-known triple-conducting electrodes (PrBa$_{0.8}$Ca$_{0.2}$)$_{0.95}$Co$_2$O$_{6-\delta}$ (−0.72 A cm$^{-2}$ at 600 °C) and PBSCF (−2.93 A cm$^{-2}$ at 650 °C)[10,12,14,17,19,39,70–76].

Under identical operating conditions, I–V curves were obtained for R-PCECs with C·BSCF and H·BSCF electrodes, as shown in Supplementary Fig. 32a, b. The current densities were measured at −1.13 and −1.80 A cm$^{-2}$ (1.3 V) at 650 °C, respectively. As expected, the dual-phase hybrid C/H·BSCF has superior electrolysis performances compared to the single-phase C·BSCF and H·BSCF electrodes (Supplementary Fig. 32c). Furthermore, the OER activity was more exact evaluated by testing the EIS of R-PCECs with different electrodes at 1.3 V (Supplementary Fig. 33). Meanwhile, the temperature dependence of $R_p$ measured on single cells with C·BSCF, C/H·BSCF, and H·BSCF electrodes were summarized in Fig. 7c. C/H·BSCF not only exhibits the lowest resistance as well as the lowest activation energy, which coincides to the ASR results in FC mode. In addition, cells with C/H·BSCF-PM electrodes also measured high performance of −2.15, −1.23, −0.79, and −0.45 A cm$^{-2}$ at 650, 600, 550, and 500 °C under 10% H$_2$O-air, respectively, further confirming the result that the dual-phase hybrid oxide has better OER catalytic activity than either phase acting alone (Supplementary Fig. 34).

Faraday efficiency (FE), defined as the ratio of actual product to theoretical output, is usually regarded as a significant indicator of electrochemical device energy utilization[13,71]. For R-PCECs, FE is utilized to reflect electron utilization during the electrocatalytic reaction and is an important parameter to evaluate the value of the whole system of the cell for practical applications. Herewith, an investigation was conducted on how different water pressures impact the FE

of R-PCEC with C/H·BSCF hybrid electrode while maintaining a current density of −1 A cm$^{-2}$ at 600 °C. As shown in Fig. 7d, the FE gradually increases with the increasing water pressure and reaches up to 91% under 40% H$_2$O-air atmosphere. Furthermore, As the current density increases, the FE gradually increases to a maximum value of 96% at −1.4 A cm$^{-2}$. However, when the current density further increased to −2.2 A cm$^{-2}$, the FE dropped to 85%. This drop can be attributed to the positive correlation between the electronic conductivity of the electrolyte and oxygen partial pressure, leading to increased electronic losses[13,75]. Interestingly, the final FE remained within a reasonable range of approximately 85% (Fig. 7e and Supplementary Fig. 35). However, acceptable FE indicates that C/H·BSCF oxide as an air electrode has a good potential for commercialization. Additionally, Supplementary Fig. 36a displays the XRD patterns of C·BSCF, C/H·BSCF, and H·BSCF powders following treatment at 600 °C under 40% H$_2$O-air. It was discovered that C·BSCF still maintains a cubic phase structure, but that C/H·BSCF and H·BSCF exhibit varying degrees of secondary phase BaO$_x$ diffraction peaks (Supplementary Fig. 36b). This suggests that the high-water pressure induces the segregation of Ba in the hexagonal phase. It has been demonstrated that BaO$_x$ can effectively enhance the ORR and OER activities of Ba-containing perovskite oxides. Hence, appropriate segregation of Ba in the H·BSCF phase under high water pressure offers greater potential for enhancing electrochemical activity[46]. Figure 7f, g shows the results of tolerance and long-term stability tests on R-PCECs with C/H·BSCF electrodes in EC mode. The cell was tested for 100 h at three current densities of −1.2, −1.4, and −1.6 A cm$^{-2}$ at 550 °C and 10% H$_2$O-air, all of which maintained notable stability (Fig. 7f). Meanwhile, the cell operated stably at a constant current density of −2 A cm$^{-2}$ at 600 °C for up to 500 h (Fig. 7g), this result greatly outperforms the stability of previous R-PCECs and demonstrates the remarkable stability of the C/H·BSCF hybrid electrode in EC mode. The feasibility of long-term cyclic running of R-PCECs between FC and EC modes is necessary for the operation of large-scale energy conversion. As shown in Fig. 7h, the R-PCEC with C/H·BSCF air electrode underwent 20 cycles for 200 h at 600 °C. The current densities used were 0.45 A cm$^{-2}$ for FC mode and −1.8 A cm$^{-2}$ for EC mode. The results revealed excellent reversibility and tolerance of R-PCECs using a C/H·BSCF hybrid electrode.

## Discussion

In summary, we synthesized a dual-phase hybrid oxide with controllable phase content using A-site modulation of the classical perovskite oxide C·BSCF. The hybrid air electrode C/H·BSCF, composed of 57.26 wt.% cubic C·BSCF and 42.74 wt.% hexagonal H·BSCF, has the greatest oxygen activation performance among the designed series of hybrid electrodes, with an ASR of just 0.05 Ω cm$^2$ at 700 °C for the symmetric cell. The single cell achieves the top level of performance in FC and EC modes employing the hybrid C/H·BSCF as an air electrode, which is inextricably linked to its extraordinary ORR and OER activity. Furthermore, experimental and characterization analyses show that the strong catalytic activity of the C/H·BSCF electrode is due to the superior oxygen activation and hydration capabilities afforded by the numerous oxygen vacancies and unique structural composition. Meanwhile, the DFT calculations reveal the energy barriers for the cubic and hexagonal perovskites during ORR and OER reactions, demonstrating the distinct advantages of each in the reaction kinetics process, and revealing the rationality of the hybrid electrode C/H·BSCF to burst out excellent catalytic activity because of the synergistic effect of the two phases. These results demonstrate the potential of the highly active and durable hybrid electrode C/H·BSCF constructed using the A-site modulation strategy for energy storage and conversion applications.

## Methods

### Materials synthesis and cell fabrication

The C/H-BSCF electrode powder was produced via the sol-gel process by dissolving $Ba(NO_3)_2$, $Sr(NO_3)_2$, $Co(NO_3)_2 \cdot 6H_2O$, and $Fe(NO_3)_3 \cdot 9H_2O$ in deionized water, along with the use of EDTA and CA as complexing agents, and regulating the pH of the solution to approximately 7 with ammonia. When the solution becomes gel-like and then baked at 200 °C for 10 h to yield the precursor. This precursor was further calcined at 1000 °C for 5 h to create the final electrode powder. The same process was utilized for preparing other electrode and electrolyte powders. Half-cells and electrolyte sheets were prepared by spin-coating and die-casting methods, respectively, and then calcined at 1450 °C and 1400 °C for 5 ho, respectively[11]. The diameters of the calcined half-cells and electrolyte sheets were approximately 11.8 and 12.4 mm, respectively. The active area of the air electrode of the anode-supported single cell is 0.28 cm². The thickness of the symmetric cell after the grinding process is 0.4 mm, and the active area of the single side is about 1.08 cm². 1 g of electrode powder was ball-milled with 10 mL of isopropyl alcohol, 2 mL of ethylene glycol and 1 mL of glycerol at 400 rpm for 30 min to make a well-mixed air electrode slurry. The electrolyte side of the half-cell and the two sides of the electrolyte sheet received the slurry application. After that, sintering was done for 2 h at 1000 °C to create single cells and symmetric cells. The hydrogen permeability and electrical conductivity tests use dense thin films and strips composed of electrode powder, respectively. 0.6 g of C/H-BSCF powder was prepared in sheets and strips by die-casting method and subsequently sintered 1200 °C for 10 h, while C-BSCF and H-BSCF were fired at 1130 °C and 1230 °C, respectively.

### Electrochemical testing

Electrochemical tests of R-PCECs were performed in both FC and EC modes in an electrical heating stove equipped with a temperature controller, which was placed in an environmental chamber. To highlight the reversibility of R-PCECs, the gas composition of the oxygen and fuel electrodes was kept constant in both test modes. At the air electrode, a high-pressure constant flow pump and atomization device were used to mix a certain flow rate of water vapor into the air such that the water pressure in the moist air was 10%. Additionally, make sure the air electrode has a total gas flow rate of 100 mL min⁻¹. Whereas the fuel electrode was supplied with 80 mL min⁻¹ of dry hydrogen gas. In addition, for symmetric cell, Faraday efficiency, and tolerance tests, the pump flow rate and air flow rate can be regulated to obtain wet air with different water pressures in order to meet different test requirements. The fuel cell test workstation (DH7002, Jiangsu Donghua Analytical Instrument Co., Ltd.) was utilized to capture the $I-V$ and power density curves. Cycling tests of R-PCECs in fuel cell and electrolysis modes were performed at 600 °C. An electrochemical workstation (Solartron 1287 + 1260 A) was used to measure the EIS of the cells under a variety of conditions. All sealed connections and current collection in the experiments were made using silver glue (DAD-87, Shanghai, China). The $D_{chem}$ and $k_{chem}$ values were gained by fitting the ECR curves measured on the electrode materials using ECRTOOLS. To induce a relaxation of conductivity, the $P_{O2}$ was altered from 0.21 atm to 0.1 atm, leading to the recording of the electrode transitioning from the original equilibrium to a new equilibrium state.

### Hydrogen permeation

The hydrogen permeation test was performed in a bidirectional permeation reaction cell. The magnetron sputtering technique is used to sputter Pd films on both sides of the hydrogen-permeable film to prevent the reduction of the hydrogen-permeable film and accelerate the hydrogen cleavage, and the thickness is about 500 nm, and then calcined at 800 °C in an argon atmosphere for 2 h. Here both the heating and cooling rates were maintained at 2 °C min⁻¹. Sealed on a ceramic tube with silver glue, the sweep gas on both sides of the hydrogen permeation film was 10% $H_2$-90% $N_2$ and argon. The tail gas of the argon purge side was passed into the gas chromatograph (GC-9860) for detection.

### Basic characterizations

The Bruker D8 Advance was used to acquire room-temperature powder XRD data, while the crystal structure of the electrode material was investigated by high-temperature XRD using a Rigaku D/max 2500 V high-temperature accessory. Raman spectrometer (Seki Technotron model STR 750 with 514.5 nm argon laser) has been used for deducing the vibrational modes of the powder samples. The SEM (JEOL-S4800) was employed to obtain the surface morphology of both powders and cells. Powder elemental distribution and microstructure were obtained by HR-TEM (JEOL, JEM-2100) and TEM-EDX (FEI Tecnai G2 T20). Elemental valence states were explored by XPS (Thermo ESCALAB 250). The mass changes of the sample during the warming process were detected using TGA (Model STA 449 F3, NETZSCH). $O_2$/$H_2O$-TPD were acquired by a mass spectrometer (MS, Hiden, HPR20), and $H_2O$-TPD data were acquired after the powders were pretreated in moist air at 600 °C for 10 h. The material's specific surface area was determined using the Quanta chrome AutoSorb-iQ3 instrument through nitrogen adsorption-desorption isotherms. Dense bars were used to test TEC data in air atmosphere by means of Netzsch 402 C/3/G. Co $K$-edge and Fe $K$-edge XANES were performed at NSRRC in Taiwan. The photon energies of Co $K$-edge and Fe $K$-edge spectra were calibrated using metallic Co and Fe foils as references, respectively.

### Computational methods

All the calculations are performed in the framework of the density functional theory with the projector augmented plane-wave method, as implemented in the Vienna ab initio simulation package[77]. The generalized gradient approximation proposed by Perdew, Burke, and Ernzerhof is selected for the exchange-correlation potential[78]. The long-range van der Waals interaction is described by the DFT-D3 approach[79]. The cut-off energy for the plane wave is set to 500 eV. The energy criterion is set to $10^{-6}$ eV in the iterative solution of the Kohn-Sham equation. In order to consider the strong electronic correlations in the localized Fe-3d and Co-3d orbitals, we add on-site Hubbard U of 3.70 eV and 3.52 eV respectively[80,81]. A vacuum layer of 15 Å is added perpendicular to the sheet to avoid artificial interaction between periodic images, and dipole correction is taken into consideration. The Brillouin zone integration is performed using a $3 \times 3 \times 1$ k-mesh for C-BSCF and a $2 \times 2 \times 1$ k-mesh for H-BSCF. All the structures are relaxed until the residual forces on the atoms have declined to less than 0.03 eV/Å. After the structure optimizations converged, the stabilities of the adsorbed structures were checked by the frequency calculations, and the thermodynamic correction of adsorbates at operating conditions was also calculated (Supplementary Tables 8 and 9). Before the calculations, the most stable doping sites of Fe were determined by calculating the total energy of different doped structures. And the preferred adsorption sites for $H_2O$ are also taken into consideration by calculating the total energy of corresponding adsorption structures. The transition states were calculated by the Climbing Image-Nudged Elastic Band (CI-NEB) method. 6 intermediate points (including the initial state and final state) were taken into consideration while solving the energy minimum path. After the convergence of the calculations, the transition states were verified by frequency calculations.

## Data availability

All data generated in this study are provided in the manuscript and Supplementary Information file. The source data file is provided in this paper. Source data are provided in this paper.

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

## Acknowledgements

The work is supported by National Key R&D Program of China (2022YFB4004000), National Natural Science Foundation of China (21878158 and 21706129), State Key Laboratory of Clean Energy Utilization (Open Fund Project No. ZJUCEU2021001), Natural Science Foundation of Jiangsu Province (BK20221312), Natural Science Foundation for Young Scholars of Jiangsu Province (BK20220879), National Natural Science Foundation for Young Scholars of China (22209072), and Jiangsu Specially-Appointed Professors, Post-graduate Research & Practice Innovation Program of Jiangsu Province (KYCX23_1465) and Cultivation Program for The Excellent Doctoral Dissertation of Nanjing Tech University (2023-09). We also acknowledge support from the Max Planck-POSTECH-Hsinchu Center for Complex Phase Materials.

## Author contributions

G.Y. and Y.Z. had the original idea of material design. Z.L., G.Y., Y.Z., and Z.S. carried out the conception and specific content design of the project. Z.L. and Y.B. performed electrochemical measurements and material characterizations. D.G. conducted the XRD refinements. H.S. assisted in the TEM characterizations and analyzed. W.H., C.P., Z.H., Y.Z., and W.L. performed XAS characterizations and date analyzed. R.R., W.Z., and Z.L. analyzed and discussed the project's data. Z.L., G.Y., Y.Z., and Z.S. co-wrote the paper.

## Competing interests

The authors declare no competing interests.
