## [Peer Review File · Nature Communications]

Synergistic dual-phase air electrode enables high and durable performance of reversible proton ceramic electrochemical cellsREVIEWER COMMENTS

Reviewer #1 (Remarks to the Author):

This is an interesting work. The following issues should be addressed.

1. The hypothesis proposed in Figure 1 is nice. However, there is a lack of experimental or computational results to support this conclusion. It would be great if the authors can provide some results to support the statement that the hybrid electrode can lead to thermal strain match.
2. The stoichiometries of H-BSCF and C-BSCF were not determined in this work. Without knowing their stoichiometries, the XRD refinement results are not reliable, or probably wrong. These stoichiometries should be the input when XRD refinement was performed. I do have a concern that these XRD refinement results in this work should be recalculated.
3. Figure 3c, the DRT analysis results should be discussed in details.
4. Figure 3d, the ASR decreases with increasing $P(H_2O)$ and it increases with further increasing $P(H_2O)$. Additional discussions should be provided.
5. It is good to have hydrogen permeation results. However, the stability of BSCF or the hybrid electrode is not good. The permeation results might not be able to represent the properties of BSCF or hybrid electrode.
6. Figure 4i. Electronic or electrical?
7. Some details regarding the cells should be provided. For example, what's the anode area, cathode area, what's the symmetric cell area. Some additional information was missed. Please add them as much as possible.

Reviewer #2 (Remarks to the Author):

In the field of reversible proton ceramic electrochemical cells, current investigations pertaining to cathodes are predominantly oriented toward the advancement of double perovskite materials. Nevertheless, the primary objective of this article is to substantiate the manifold advantages engendered by employing the same material while manipulating distinct crystal structures for the cathode and the entire cell, thereby manifesting its ingenuity. Within the domain of materials research, it is widely acknowledged that

structural configuration exerts a profound influence on performance; however, expounding the precise rationale behind how structural modifications instigate alterations in material properties presents a challenge in practical applications. Thus, this article possesses the capacity to engender heightened contemplation among its readership and elicit an augmented interest in the exploration and comprehension of material structures.

Collectively, the paper demonstrates exceptional quality, facilitating multiple methodologies for experimental validation, which in turn augments the credibility of the outcomes.

Nonetheless, certain gaps or ambiguities persist within the text, warranting the author's elucidation and explication.

1. The C/H-BSCF-PM electrode was used for comparison with the C/H-BSCF electrode to demonstrate that mechanical mixing results in particle enlargement and non-uniform distribution of the two phases, which adversely affects ion transport and surface diffusion, consequently inhibiting the electrode's oxygen activation rate. However, this statement contradicts the observation in Fig. 3f. Therefore, a detailed explanation is warranted to resolve this contradictory result.

2. More data on higher current densities should be provided in Fig. 7e. This would not only more convincingly verify that higher oxygen partial pressure leads to elevated electron losses, but also provide a better understanding of the Faradaic efficiency at high current densities. If the Faradaic efficiency remains within a reasonable range as the current density increases, the reported exciting current density of 3.73 A cm^{-2} @1.3V in electrolysis mode holds significance. Please provide the requested information.

3. Multiple comparisons of particle sizes are made in the text, such as mechanical mixing generating larger particles of C/H-BSCF-PM than C/H-BSCF, and line-scan distance may need to be compared with particle size to make the distribution of two phases more apparent, and thus strengthen the persuasiveness from this comparison.

4. The author mentions that R_p exhibits a lower activation energy of 0.73 eV in Fig. 6b but does not provide specific comparative data. It is recommended to either indicate the comparative values mentioned in the text or provide a reasonable explanation for the absence of such data.

Reviewer #3 (Remarks to the Author):

Major revision

In this work, Liu et.al developed a dual-phase air electrode for R-PCECs, consisting with self-assembled H-BSCF and C-BSCF, which shows a good potential as air electrode. They found that BSCF-1.5 has the best performance and attribute it to the synergistic effect. This work is interesting and well-organized. However, it lacks of novelty since there are some similar works reported self-assembled air electrodes, and there are also some questions need to be solved. I suggest a major revision of this work.

Questions:

1. It is proposed that C/H-BSCF benefits from the reduced mismatched thermal stress between multiple phases, but the TEC data of C-BSCF and H-BSCF are lacking.
2. From TEM results, we can see that the interface of C-BSCF and H-BSCF consists of C-BSCF (110) and H-BSCF (002). The atomic arrangement of two phases in this interface is so different that there should be an obvious strain in the surface region. Maybe the improved catalysis activity is aroused by the strain effect and should be discussed.
3. When described XRD refinement results of H-BSCF (line 169 in page 9), it should be $a=b=11.7115(7) \text{ \AA}$ but not $a=b=c=11.7115(7) \text{ \AA}$
4. EIS and DRT results in Figure 3 should be discussed further. Activation energies of different processes (HF, IF and LF) should be calculated respectively. It is encouraged to test EIS at various PO₂ and to perform DRT analysis to probe the rate-determine step of C-BSCF, H-BSCF and C/H-BSCF, which may be different among three electrodes.
5. Activation energy of electrical conductivity should be calculated and discussed.
6. Proton conductivity is essential for air electrodes of R-PCECs, which should be explicitly measured but not discussed roughly by H₂O-TPD(line 344 in page 18) .
7. The migration ability of oxygen is discussed in line 324. However, O₂-TPD is not the direct evidence of it, ECR result is more accurate.
8. About ECR:
 - a) ECR results of H-BSCF should be provided to verify the synergistic effect of C/H-BSCF by comparing it with C-BSCF and H-BSCF. It is encouraged to perform ECR tests of PM-C/H-BSCF.
 - b) The raw data curves and fitting curves in Figure S12 should be provided to verify the accuracy of fitting.
 - c) Sintering temperature of bar samples should be given in the experimental section, as well as SEM pictures of the surface and cross-section of bars.
 - d) When fabricating bar samples for ECR, the densification procedure requires a high sintering temperature, which may raise the change in composition or crystal structure. XRD and refinement results of sintered bars should be provided and discussed.
9. The stability of powders at wet atmosphere (40% H₂O) should be discussed according to XRD and SEM results.
10. About DFT calculations
 - a) All of the structure files should be provided.
 - b) When constructing structures of C-BSCF and H-BSCF, there will be many different configurations with different atomic arrangements due to doping. Did authors use the configuration with the lowest energy? The relative energy of different

configurations should be provided and discussed.

- c) More details about DFT calculations should be provided.
- d) How many layers of slab models?
- e) How many layers were fixed during the calculation?
- f) Thermodynamic correction of adsorbates at operating condition should be performed, frequencies, ZPE and $T\Delta S$ of every structure should be provided.
- g) Does dipole correction is taken into consideration for slab models?
- h) Does the Hubbard correction is taken into consideration?
- i) When calculating reaction pathways, is there any oxygen vacancy on the surface?
- j) For calculations of surface, there are many different adsorption sites (line, Co-top, Fe-top, VO-top, bridge, and so on), which should be considered sufficiently to ensure the reaction occurs on the sites with the most negative value of adsorption energy.
- k) The synergistic effect should be calculated by constructing a heterostructure, it is encouraged to perform calculations at the C/H-BSCF interface.

Point-to-Point Responses to Reviewers' Comments and Suggestions

Firstly, we would like to express our gratitude for the time and effort the reviewers dedicated to reviewing our manuscript (NCOMMS-23-23007). We also highly appreciate the thoughtful comments and suggestions provided by the reviewers, which have substantially enhanced the quality and clarity of our manuscript. We have carefully revised the manuscript to thoroughly address all the comments and concerns raised. The modifications have been marked in **yellow** in the revised manuscript. Our point-to-point responses are presented below.

Reviewer #1 (Remarks to the Author):

This is an interesting work. The following issues should be addressed.

Response:

We thank the reviewer for careful reading of our manuscript and the acknowledgement of our work. Following the suggestions from the reviewer, we have made detailed revisions to improve our manuscript.

Comment 1:

The hypothesis proposed in Figure 1 is nice. However, there is a lack of experimental or computational results to support this conclusion. It would be great if the authors can provide some results to support the statement that the hybrid electrode can lead to thermal strain match.

Response to C1:

We appreciate the reviewer's detailed comment, which have been immensely helpful in further refine our work. In response to the reviewer's suggestions, we have incorporated specific tests to validate the conclusions drawn in Figure 1. These additions are now included in the revised manuscript.

Firstly, we investigated the thermal expansion coefficients (TEC) of various air electrodes, a critical factor in assessing thermal compatibility between air electrodes and electrolytes. **Figure R1a** shows the linear expansion (dL/L_0) curves of C-BSCF, C/H-BSCF, C/H-BSCF-PM and H-BSCF samples with respect to

temperature. The TEC values of C-BSCF and H-BSCF were $23.8 \times 10^{-6} \text{ K}^{-1}$ and $19.2 \times 10^{-6} \text{ K}^{-1}$, respectively. Notably, the TEC of the cubic C-BSCF aligns with literature reports [Song *et al.*, *Joule* 3 2842 (2019); Harvey *et al.*, *Phys. Chem. Chem. Phys.* 11 3090 (2009)], whereas the hexagonal H-BSCF exhibits a lower TEC than C-BSCF. As expected, the TEC of the C/H-BSCF and C/H-BSCF-PM were $19.7 \times 10^{-6} \text{ K}^{-1}$ and $20.4 \times 10^{-6} \text{ K}^{-1}$, respectively, were found to be intermediate between C-BSCF and H-BSCF. This suggests that the hybrid electrode demonstrates superior thermal matching with the electrolyte compared to the majority of conventional Co/Fe-based electrodes, as illustrated by its lower TEC relative to multiple advanced Co/Fe-based perovskite air electrodes (**Fig. R1b**) [Swierczek *et al.*, *J. Electrochem. Soc.* 57 1993 (2013); Sakai *et al.*, *J. Mater. Chem. A* 9 3584 (2021); Nikonov *et al.*, *J. Alloys Compd.* 865 158898 (2021); Cascos *et al.*, *ACS Appl. Energy Mater.* 4 500 (2021); He *et al.*, *Adv. Funct. Mater.* 32 2206756 (2022); Ishihara *et al.*, *J. Electrochem. Soc.* 103 1425 (2021); Liang *et al.*, *Chem. Eng. J.* 420 127717 (2020); Kim *et al.*, *Chem. Mater.* 22 883 (2010); Wang *et al.*, *Int. J. Hydrogen Energy* 47 38327 (2022); Lim *et al.*, *J. Mater. Chem. A* 4 6479 (2016); Wang *et al.*, *Int. J. Hydrogen Energy* 47 9395 (2022)]. As shown in **Fig. R1c**, we further compare the TEC of C/H-BSCF with various composite electrodes such as PBC-SSC, LSCSb-20SDC, BSCF4-SDC, BCFY2, BC1.5MN, and NCBCO composite electrodes, which exhibit TEC values of $25.5 \times 10^{-6} \text{ K}^{-1}$, $21.3 \times 10^{-6} \text{ K}^{-1}$, $21.5 \times 10^{-6} \text{ K}^{-1}$, $20.3 \times 10^{-6} \text{ K}^{-1}$, $21.1 \times 10^{-6} \text{ K}^{-1}$ and $22.6 \times 10^{-6} \text{ K}^{-1}$, respectively [Ishihara *et al.*, *J. Electrochem. Soc.* 103 1425 (2021); Zhang *et al.*, *Solid State Sci.* 91 126 (2019); Li *et al.*, *J. Alloys Compd.* 448 116 (2008); Sun *et al.*, *J. Alloys Compd.* 885 160901 (2021); Tsvetkov *et al.*, *Energies* 12 417 (2019); Dyck *et al.*, *Solid State Ionics* 171 17 (2004); Zou *et al.*, *J. Mater. Chem. A* 10 5381 (2022); He *et al.*, *Adv. Funct. Mater.* 32 2206756 (2022); Wei *et al.*, *Int. J. Hydrogen Energy* 46 23868 (2021); Jo *et al.*, *Chem. Eng. J.* 451 138954 (2023)].

It is worth noting that numerous composite electrodes, prepared by physical mixing and self-assembly methods, exhibit substantial differences in their two-phase structures and elemental compositions. These variations hinder effective regulation of thermal stresses among phases at high temperatures, directly impacting the TEC of conventional composite electrodes. The TEC of such composite electrodes depends on the TEC of each phase and its proportion within the composite. For example, the TEC of a composite electrode composed of $\text{CePrO}_{2-\delta}/\text{La}_{0.4}\text{Pr}_{0.1}\text{Ba}_{0.5}\text{Co}_{0.975}\text{O}_{3-\delta}$, as reported by Aristizabal *et al.* [Aristizabal *et*

al., *Mater. Lett.* 350 134966 (2023)], shows a clear linear relationship with the percentage of $\text{CePrO}_{2-\delta}$. Similar conclusions were drawn by He *et al.*, investigating the BC1.5MN composite electrode, and Zhu *et al.*, studying the $\text{SrCo}_{0.8}\text{Fe}_{0.2}\text{O}_{3-\delta}/\text{La}_{0.45}\text{Ce}_{0.55}\text{O}_{2-\delta}$ composite cathode [He *et al.*, *Adv. Funct. Mater.* 32 2206756 (2022); Zhu *et al.*, *Int. J. Hydrogen Energy* 36 12549 (2011)]. Interestingly, the TEC of C/H-BSCF aligns closely with that of H-BSCF. This suggests that a C/H-BSCF hybrid electrode, possessing the same elemental composition as the two phases, effectively modulates thermal stress at high temperatures, consequently reducing the overall TEC of the hybrid electrode. To better understand this behavior, we conducted Rietveld refinement on the XRD analysis of the C/H-BSCF hybrid at 700 °C. The results reveal that the weight ratios of the cubic and hexagonal phases of C/H-BSCF were found to be 48.82 wt.% and 51.18 wt.%, respectively, as shown in **Fig. R1d**. The C/H-BSCF hybrid electrode achieves favorable thermal matching at high temperatures by autonomously transitioning from the high-TEC cubic phase to the lower-TEC hexagonal phase. This behavior can be observed by comparing the mass ratios of the two phases at room temperature.

Fig. R1 (also **Supplementary Fig. 22**) **a** Linear expansion (dL/L_0) curves of C-BSCF, C/H-BSCF, C/H-BSCF-PM and H-BSCF from 40 to 1000 °C. TEC comparison of C/H-BSCF electrode with reported advanced **(b)** single-phase, and **(c)** composite Co/Fe-based air electrodes [$\text{NdBa}_{0.5}\text{Sr}_{0.5}\text{Co}_{1.5}\text{Fe}_{0.5}\text{O}_{5+\delta}$ (NBSCF), $\text{Y}_{0.96}\text{CoO}_{3-\delta}$ (YC); $\text{Pr}_{0.6}\text{Sr}_{0.4}\text{Fe}_{0.8}\text{Co}_{0.2}\text{O}_{3-\delta}$ (PSCF-4020), $\text{SrCo}_{0.95}\text{Ta}_{0.05}\text{O}_{3-\delta}$ (SCT), $\text{BaCoNbO}_{3-\delta}$ (BCN), $\text{Sm}_{0.5}\text{Sr}_{0.5}\text{CoO}_3$ (SSC), $\text{SmBaCo}_2\text{O}_{5+\delta}$ (SBC), $\text{SmBa}_{0.5}\text{Sr}_{0.5}\text{Co}_2\text{O}_{5+\delta}$ (SBSC), $\text{PrBaCo}_2\text{O}_{5+\delta}$ (PBC), $\text{PrBaCo}_{1.5}\text{Fe}_{0.5}\text{O}_{5+\delta}$ (PBCF), $\text{BaCo}_{0.4}\text{Zr}_{0.1}\text{Fe}_{0.4}\text{Y}_{0.1}\text{O}_{3-\delta}$ (BCZFY), $\text{PrBa}_{0.9}\text{Ca}_{0.1}\text{Co}_2\text{O}_{5+\delta}$ (PBCC10), $\text{BaZr}_{0.1}\text{Fe}_{0.75}\text{Ni}_{0.15}\text{O}_{3-\delta}$ (BZFN15), $\text{La}_{0.4}\text{Sr}_{0.6}\text{Co}_{0.9}\text{Sb}_{0.1}\text{O}_{3-\delta}\text{-Ce}_{0.8}\text{Sm}_{0.2}\text{O}_{1.9}$ (LSCSb-20SDC), $\text{Ba}_2\text{Co}_{1.5}\text{Mo}_{0.25}\text{Nb}_{0.25}\text{O}_{6-\delta}$ (BC1.5MN), $\text{NdBaCo}_2\text{O}_{5+\delta}\text{-Ce}_{0.9}\text{Gd}_{0.1}\text{O}_{1.95}$ (NBCO-8CGO), $\text{BaCe}_{0.16}\text{Y}_{0.04}\text{Fe}_{0.8}\text{O}_{3-\delta}$ (BCYF2), $\text{Ba}_{0.5}\text{Sr}_{0.5}\text{Co}_{0.6}\text{Fe}_{0.4}\text{O}_{3-\delta}\text{-Ce}_{0.8}\text{Sm}_{0.2}\text{O}_{1.9}$ (BSCF4-SDC), $\text{BaZr}_{0.1}\text{Fe}_{0.9}\text{Ni}_{0.1}\text{O}_{3-\delta}$ (BCFY10), $\text{Gd}_{0.1}\text{Ca}_{0.1}\text{Ba}_{1.8}\text{Co}_9\text{O}_{14}$ (NCBCO), $\text{Gd}_{0.8}\text{Sr}_{0.2}\text{CoO}_{3-\delta}\text{-Ce}_{0.8}\text{Gd}_{0.2}\text{O}_{1.95}$ (GSC-SDC)]. **d** Rietveld refinement profiles of XRD pattern of C/H-BSCF hybrid at 700 °C.

Related data (**Supplementary Fig. 22**) and discussions (yellow-highlighted in **Line 413-438, Page 22-23**) have been added in the revised manuscript as follows:

*“We investigated the thermal expansion coefficients (TEC) of various air electrodes in order to confirm that C/H-BSCF, a self-assembled composite electrode with two phases having the same elemental composition, can significantly reduce the thermal stress to improve the match with the electrolyte. **Supplementary Fig. 22a** shows the linear expansion (dL/L_0) of the C-BSCF, C/H-BSCF, C/H-BSCF-PM and H-BSCF samples with temperature. the TEC of C-BSCF and H-BSCF were $23.8 \times 10^{-6} \text{ K}^{-1}$ and $19.2 \times 10^{-6} \text{ K}^{-1}$, respectively, where the TEC of cubic C-BSCF is consistent with literature reports¹⁸, while hexagonal H-BSCF exhibits a lower TEC than C-BSCF. As expected, the TEC of the C/H-BSCF and C/H-BSCF-PM hybrids, which were $19.7 \times 10^{-6} \text{ K}^{-1}$ and $20.4 \times 10^{-6} \text{ K}^{-1}$, respectively, were intermediate between those of C-BSCF and H-BSCF. The hybrid electrode shows better thermal matching with the electrolyte compared to the majority of conventional Co/Fe-based electrodes (**Supplementary Fig. 22b, c**)^{9,59-66}. It is worth noting that numerous composite electrodes prepared by physical mixing and self-assembly methods exhibit significant differences in terms of their two-phase structures and elemental compositions. These differences hinder the composite electrodes from effectively regulating thermal stresses among the phases at high temperatures. Therefore, the TEC of conventional composite electrode is directly dependent on the*

*TEC of each phase and the proportion of each phase within the composite^{61,65,67}. Interestingly, the TEC of C/H-BSCF is found to be closer to the TEC of H-BSCF. This suggests that a C/H-BSCF hybrid electrode, which possesses the same elemental composition as the two phases, can effectively modulate thermal stress at high temperatures, consequently reducing the overall TEC of the hybrid electrode. To better understand this behavior, we conducted Rietveld refinement on the XRD analysis of the C/H-BSCF hybrid at 700 °C. The results reveal that at 700 °C, the weight ratios of the cubic and hexagonal phases of C/H-BSCF were found to be 48.82 wt.% and 51.18 wt.%, respectively, as shown in **Supplementary Fig. 22d**. The C/H-BSCF hybrid electrode achieves favorable thermal matching at high temperatures by autonomously transitioning from the high-TEC cubic phase to the lower-TEC hexagonal phase.”*

Comment 2:

The stoichiometries of H-BSCF and C-BSCF were not determined in this work. Without knowing their stoichiometries, the XRD refinement results are not reliable, or probably wrong. These stoichiometries should be the input when XRD refinement was performed. I do have a concern that these XRD refinement results in this work should be recalculated.

Response to C2:

We would like to express our gratitude to the reviewer for raising this question. As pointed out by the reviewer, the stoichiometries of H-BSCF and C-BSCF were not determined, which might have led to unreliable XRD refinement results. In order to ensure a more accurate analysis of the content of the two phases, H-BSCF and C-BSCF, in the hybrid electrode, we determined their stoichiometries using the scanning transmission electron microscopy (STEM). These stoichiometries were then used in the XRD refinement process, enabling us to obtain revised XRD refinement results. **Fig. R2a** presents the STEM image of the C/H-BSCF powder. To further investigate the phase structure and composition of the hybrid C/H-BSCF, we conducted high-resolution transmission electron microscopy (HR-TEM). At points 1 and 2, diffraction patterns corresponding to (110) and (100) planes of C-BSCF and H-BSCF phases were detected, revealing lattice spacings of 0.282 nm and 1.01 nm, respectively (**Fig. R2b, c**) [*Liu et al., Chem. Eng. J. 450 137787 (2022)*; *Zhu et al., Adv. Mater. 32 e1905025 (2020)*]. EDX scanning at points 1 and 2

indicated the presence of Ba, Sr, Co, Fe, and O in both phases, with the main difference lying in the ratios of the A-site elements Ba and Sr to the B-site elements Co and Fe (**Fig. R2d, e**). Point EDX scanning results show that the elemental compositions of the cubic and hexagonal phases in C/H-BSCF are in general agreement with the expected design composition. With the stoichiometries of H-BSCF and C-BSCF phases determined, we proceeded to re-evaluate the XRD refinement of the C/H-BSCF hybrid.

Fig. R2 (also **Supplementary Fig. 2**) Phase composition and crystal structure analysis of the hybrid C/H-BSCF. **a** STEM image of C/H-BSCF particle. HR-TEM images of the **(b)** point 1 and **(c)** point 2 regions in fig. R2a, respectively. Point EDX scanning results at **(d)** point 1 and **(e)** point 2.

As depicted in **Fig. R3**, the XRD refinement showcased that the crystal structural composition of the C/H-BSCF sample surface consists of 57.26 wt.% cubic phase and 42.74 wt.% hexagonal phase. The cubic and hexagonal phases correspond respectively to the space group of C-BSCF and H-BSCF. The refined results were confirmed to be accurate based on reliability factors of $R_{\text{exp}} = 5.44\%$, $R_{\text{wp}} = 5.42\%$, and $\text{GOF} = 1.00$.

Fig. R3 (also **Fig. 2c**) Refined XRD profiles of the prepared C/H-BSCF hybrid.

Related data (**Supplementary Fig. 2** and **Fig. 2c**) and discussions (yellow-highlighted in **Line 170-178**, **Page 10**) have been added in the revised manuscript as follows:

“In order to improve the accuracy of the XRD refinement results for the C/H-BSCF hybrid, transmission electron microscopy and energy dispersive X-ray (TEM-EDX) techniques were employed to determine the stoichiometries of the two phases (Supplementary Fig. 2). These determined values were then utilized as input parameters for the XRD refinement process. The crystal structural composition of C/H-BSCF sample surface was revealed to be 57.26 wt.% cubic and 42.74 wt.% hexagonal through XRD refinement, where the cubic and hexagonal phases are consistent with the space group of C-BSCF and H-BSCF, respectively (Fig. 2c). The refined results were confirmed to be accurate based on reliability factors of $R_{exp} = 5.44\%$, $R_{wp} = 5.42\%$, and $GOF = 1.00^{28}$.”

Comment 3:

Figure 3c, the DRT analysis results should be discussed in details.

Response to C3:

We sincerely appreciate the reviewer for providing this valuable comment. In response, we have enhanced the revised manuscript by providing a detailed discussion of the DRT analysis results. **Figure R4** (Fig. 3c) illustrates the DRT plots obtained from measurements conducted at 600 °C in wet air for the C-BSCF, C/H-BSCF and H-BSCF electrodes.

The DRT plots exhibit distinct peaks in the low frequency (LF, <10 Hz), intermediate frequency (IF, 10-2000 Hz), and high frequency (HF, >2000 Hz) regions, which are conventionally associated with the gas diffusion process, ion migration and surface exchange, and charge transfer process, respectively [Niu et al., *Adv. Energy Mater.* 12 2103783 (2022); He et al., *Adv. Mater.* 35 e2209469 (2023)]. Compared to the LF and HF regions, the resistance in the IF region primarily governs the total resistance. This indicates that the electrocatalytic process of the air electrode is notably influenced by the surface exchange and bulk diffusion rates, specifically involving surface adsorption and dissociation of oxygen and water, along with the bulk diffusion rates of oxygen ions and protons. Interestingly, the peak area of the hybrid electrode C/H-BSCF in the IF region is situated below those of the single-phase C-BSCF and H-BSCF electrodes. This observation not only demonstrates the excellent surface exchange and bulk diffusion rates of the C/H-BSCF electrode but also highlights the advantages of the synergistic effects of the dual phases in both oxygen reduction reaction (ORR) and hydration reactions on the air electrode surface.

Fig. R4 (also **Fig. 3c**) DRT analysis of EIS measured at C-BSCF, C/H-BSCF and H-BSCF electrodes at 600 °C.

Related discussions (yellow-highlighted in **Line 254-268, Page 14**) have been added in the revised manuscript as follows:

“As depicted in Fig. 3c, compared to the LF and HF regions, the resistance in the IF region primarily governs the total resistance, indicating that the surface exchange and bulk diffusion rates of the air

electrode, specifically the surface adsorption and dissociation of oxygen and water, as well as the bulk diffusion rates of oxygen ions and protons, significantly limit the electrocatalytic process. Interestingly, the peak area of the hybrid electrode C/H-BSCF in the IF region is not between, but lower than that of the single-phase C-BSCF and H-BSCF electrodes. This not only demonstrates the excellent surface exchange and bulk diffusion rates of the C/H-BSCF electrode but also highlights the advantages of the synergistic effect from the dual phase in the ORR and hydration reaction on the air electrode surface. Meanwhile, the temperature dependence of ASR for C- BSCF, C/H-BSCF, and H-BSCF electrodes in the HF, IF, and LF regions is illustrated in **Supplementary Fig. 9**. The activation energy of the C/H-BSCF electrode in the IF region is 1.24 eV, which is in the middle of the activation energies of C-BSCF and H-BSCF. This is mainly due to the different ionic conduction promotion rates of the cubic and hexagonal phases in the C/H-BSCF hybrid electrode³⁹.”

Comment 4:

Figure 3d, the ASR decreases with increasing P(H₂O) and it increases with further increasing P(H₂O). Additional discussions should be provided.

Response to C4:

We sincerely appreciate the reviewer for their valuable comment. In response, we have addressed these concerns in the revised version. Particularly, we have provided a specific explanation to help clarify the issues regarding **Fig. R5**.

To investigate the electrochemical activity of the C/H-BSCF air electrode under various water vapor pressures, symmetric cell with C/H-BSCF electrode was evaluated under 0-20% H₂O-air conditions. As depicted in **Fig. R5**, firstly, as the water vapor pressure changed from 0% to 3% H₂O-air, the ASR of the C/H-BSCF cell decreased from 0.62 to 0.26 $\Omega \text{ cm}^2$. This significant reduction in ASR indicates that the C/H-BSCF electrode has a fast hydration reaction rate and excellent proton diffusion capability. Furthermore, the activation energy of the hybrid electrode C/H-BSCF decreased with increasing water vapor pressure, which aligns with the main characteristics of the air electrode in proton-conducting electrochemical cells. However, beyond 3% H₂O-air, it is worth noting that the ASR at 700 °C increased

more prominently compared to 500 °C. This can be attributed to the increased water vapor pressure leads to competition between oxygen and water molecules for adsorption on the electrode surface, thus lowering the ORR rate of the electrode. At 500 °C, the intrinsic ORR rate of the air electrode is relatively slow, so the increase in water vapor pressure results in a relatively weaker increase in ASR. Therefore, it is crucial to control the water vapor pressure for optimizing the electrochemical activity of the air electrode.

Fig. R5 (also **Fig. 3d**) Arrhenius plots of the ASRs of symmetric cell with C/H-BSCF electrode at different water pressures (0, 3, 5, 10, 20% H_2O -air) at 500-700 °C.

Related discussions (yellow-highlighted in **Line 286-291, Page 15**) have been added in the revised manuscript as follows:

“This significant reduction in ASR indicates that the C/H-BSCF electrode has a fast hydration reaction rate and excellent proton diffusion capability. However, beyond 3% H_2O -air, it is worth noting that the ASR at 700 °C increased more prominently compared to 500 °C. This can be attributed to the increased water vapor pressure leads to competition between oxygen and water molecules for adsorption on the electrode surface, thus lowering the ORR rate of the electrode.”

Comment 5:

It is good to have hydrogen permeation results. However, the stability of BSCF or the hybrid electrode is

not good. The permeation results might not be able to represent the properties of BSCF or hybrid electrode.

Response to C5:

Thank you to the reviewer for acknowledging the hydrogen permeation results. As the reviewer mentioned, the stability of C-BSCF or hybrid electrode C/H-BSCF in a hydrogen atmosphere is poor, primarily due to the susceptibility of a large amount of transition metal ions, such as Co and Fe to reduction under hydrogen gas, resulting in the breakdown of the original perovskite structure. Therefore, to ensure reliable hydrogen permeation results, both sides of the membrane were coated with a 500 nm-thick Pd film using magnetron sputtering as reported [Zhou *et al.*, *J. Mater. Chem. A* 7 13265 (2019)]. After sputtering, the membranes were further calcined in an argon atmosphere at 800 °C for 2 h (**Fig. R6a, b**). This is done to prevent the reduction of the perovskite hydrogen permeation membrane and simultaneously enhance the surface hydrogen catalytic rate. Finally, we have also included additional detailed information and evidence in the revised manuscript.

Fig. R6 (also **Supplementary Fig. 20**) SEM images of (a) the surface and (b) cross section of the sputtered palladium film.

Related data (**Supplementary Fig. 20**) and discussions (yellow-highlighted in **Line 391-393, Page 21**) have been added in the revised manuscript as follows:

*“To ensure reliable hydrogen permeation results, both sides of the membrane were coated with a 500 nm-thick Pd film using magnetron sputtering (**Supplementary Fig. 20**)⁵⁸.”*

Comment 6:

Figure 4i. Electronic or electrical?

Response to C6:

We are apologizing for any confusion caused by our unclear presentation. In Fig. 4i, we illustrate the total electrical conductivity of C-BSCF, C/H-BSCF and H-BSCF at 300-800 °C under dry air conditions. Related figure (yellow-highlighted in **Line 336, Page 18**) has been changed in the revised manuscript.

Comment 7:

Some details regarding the cells should be provided. For example, what's the anode area, cathode area, what's the symmetric cell area. Some additional information was missed. Please add them as much as possible.

Response to C7:

Thanks for the reviewer's comment. We apologize for the oversight in the manuscript regarding critical information on cell preparation. In response, we have included as much detailed information as possible in the revised manuscript (yellow-highlighted in **Line 628-631, Page 33**).

The diameters of the calcined half-cells and electrolyte sheets were approximately 11.8 and 12.4 mm, respectively. The active area of the air electrode of the anode-supported single cell is 0.28 cm². Following the grinding process, the thickness of the symmetric cell was reduced to 0.4 mm, and the area on a single side was approximately 1.08 cm².

Reviewer #2 (Remarks to the Author):

In the field of reversible proton ceramic electrochemical cells, current investigations pertaining to cathodes are predominantly oriented toward the advancement of double perovskite materials. Nevertheless, the primary objective of this article is to substantiate the manifold advantages engendered by employing the same material while manipulating distinct crystal structures for the cathode and the entire cell, thereby manifesting its ingenuity. Within the domain of materials research, it is widely acknowledged that

structural configuration exerts a profound influence on performance; however, expounding the precise rationale behind how structural modifications instigate alterations in material properties present a challenge in practical applications. Thus, this article possesses the capacity to engender heightened contemplation among its readership and elicit an augmented interest in the exploration and comprehension of material structures. Collectively, the paper demonstrates exceptional quality, facilitating multiple methodologies for experimental validation, which in turn augments the credibility of the outcomes. Nonetheless, certain gaps or ambiguities persist within the text, warranting the author's elucidation and explication.

Response:

We express our sincere gratitude to the reviewer for providing us with detailed and insightful comments. These comments have greatly contributed to enhancing the quality of this work. We have made every effort to address these comments in the revised version of our manuscript. Please find our responses to the comments point by point as follows.

Comment 1:

1. The C/H-BSCF-PM electrode was used for comparison with the C/H-BSCF electrode to demonstrate that mechanical mixing results in particle enlargement and non-uniform distribution of the two phases, which adversely affects ion transport and surface diffusion, consequently inhibiting the electrode's oxygen activation rate. However, this statement contradicts the observation in Fig. 3f. Therefore, a detailed explanation is warranted to resolve this contradictory result.

Response to C1:

Thanks for the reviewer's comment. We sincerely apologize for any confusion resulting from the mishandling of our data. The DRT curves depicted in **Fig. R7b** (Fig. 3f) were fitted based on the EIS results presented in **Fig. R7a** (Fig. 3e). As the symmetric cell of the C/H-BSCF electrode exhibits a lower polarization resistance (R_p), the corresponding DRT profile should display a smaller total peak area in **Fig. R7b**. Regrettably, due to an incorrectly labeled legend in **Fig. R7b**, the DRT spectra for C/H-BSCF and C/H-BSCF-PM electrodes were unintentionally interchanged, leading to conflicting outcomes.

In the revised manuscript, we have rectified **Fig. R7b** accordingly. The revised figure clearly demonstrates that the C/H-BSCF electrode displays a lower R_p compared to C/H-BSCF-PM, thus illustrating the superior electrochemical activity of the self-assembled composite electrode. Additionally, the DRT analysis clearly reveals smaller peak areas for the C/H-BSCF electrode in both the intermediate-frequency (IF, 10-2000 Hz) and low-frequency (LF, <10 Hz) regions, further indicating faster ion migration and surface mass transfer in comparison to the conventionally physically mixed C/H-BSCF-PM electrode [Xu *et al.*, *Adv. Funct. Mater.* 32 2110998 (2022)]. It is worth noting that the enhancement in ion migration and surface exchange rates with the self-assembled composite electrode over its physically mixed counterpart has been widely confirmed [Kim *et al.*, *J. Mater. Chem. A* 10 2496 (2022); Liang *et al.*, *Adv. Mater.* 34 2106379 (2021)]. For example, Kim *et al.* present $\text{Ba}_{0.5}\text{Sr}_{0.5}\text{Co}_{0.6}\text{Fe}_{0.2}\text{Zr}_{0.1}\text{Y}_{0.1}\text{O}_{3-\delta}$ (BSCFZY) as a superior biphasic nano-composite cathode, which self-assembles into two discrete cubic perovskites: Co-rich ($\text{Ba}_{0.5}\text{Sr}_{0.5}\text{Co}_{0.7}\text{Fe}_{0.2}\text{Zr}_{0.07}\text{Y}_{0.03}\text{O}_{3-\delta}$) and Zr-rich ($\text{Ba}_{0.6}\text{Sr}_{0.4}\text{Co}_{0.3}\text{Fe}_{0.2}\text{Zr}_{0.4}\text{Y}_{0.1}\text{O}_{3-\delta}$) phases. Symmetric cell testing confirmed the superiority of the self-assembled composite electrode over the mixed electrode of BSCF and BCFZY prepared via ball milling. Additionally, SEM images and DRT fitting results of the electrode powders validated the same conclusions as presented in our manuscript.

Fig. R7 (also **Fig. 3e, f**) **a** EIS and **(b)** DRT analysis of self-assembled and physically mixed C/H-BSCF electrodes measured at 600 °C under 3% H_2O -air conditions.

Comment 2:

More data on higher current densities should be provided in Fig. 7e. This would not only more convincingly verify that higher oxygen partial pressure leads to elevated electron losses, but also provide a better understanding of the Faradaic efficiency at high current densities. If the Faradaic efficiency remains within a reasonable range as the current density increases, the reported exciting current density of 3.73 A cm^{-2} @ 1.3 V in electrolysis mode holds significance. Please provide the requested information.

Response to C2:

We thank the reviewer for this question. As the reviewers suggested, providing more information about the Faraday efficiency (FE) at higher current densities is necessary. In response to this, we have made revisions and additions to the manuscript and Supplementary Information. Specifically, we have included extra data for a single cell with C/H-BSCF electrode within the current density range of -1.8 to -3.2 A cm^{-2} . As shown in the **Fig. R8**, we evaluated the hydrogen production rate of the cell in electrolysis mode at $600 \text{ }^\circ\text{C}$ under 40% H_2O -air condition. It can be observed that as the current density increased from -0.8 to -3.2 A cm^{-2} , the hydrogen production rate also increased from 4.25 to $18.99 \text{ mL cm}^{-2} \text{ min}^{-1}$. Furthermore, the corresponding FE was calculated based on the hydrogen production rate at different current densities. As the current density increased, the FE gradually reached its maximum value of 96% at -1.4 A cm^{-2} . However, when the current density further increased to -2.2 A cm^{-2} , the FE dropped to 85%. This drop can be attributed to the positive correlation between the electronic conductivity of the electrolyte and oxygen partial pressure, leading to increased electronic losses [Liu *et al.*, *Nano-Micro Lett.* 14 217 (2022)]. Interestingly, the final FE remained within a reasonable range of approximately 85%. This emphasizes the significance of the cell's exciting performance achieving -2.72 A cm^{-2} @ 1.3 V under the electrolysis mode at $600 \text{ }^\circ\text{C}$.

In addition, we compared the FE with that of the reported advanced R-PCECs. Saqib *et al.* reported a high-performance and durable bifunctional oxygen electrode material, $\text{Gd}_{0.3}\text{Ca}_{2.7}\text{Co}_{3.82}\text{Cu}_{0.18}\text{O}_{9.8}$ (GCCCO), applied in R-PCECs, which achieved similar FE at $600 \text{ }^\circ\text{C}$ [Saqib *et al.*, *Energy Environ. Sci.* 14 2472 (2021)]. Duan *et al.* We also observed a similar trend of FE variation with increasing current density in R-PCECs, as demonstrated by Duan *et al.* Their research employed $\text{BaCo}_{0.4}\text{Fe}_{0.4}\text{Zr}_{0.1}\text{Y}_{0.1}\text{O}_{3.8}$

(BCFZY) as the air electrode and BZCYYb as the electrolyte [Duan *et al.*, *Nat. Energy* 4 230 (2019)].

Fig. R8 (also **Fig 7e** and **Supplementary Fig. 33**) a,b FE of the single cell with C/H-BSCF electrode measured at different current densities at 600 °C under 40% H₂O-air conditions.

Related data (**Supplementary Fig. 33**) and discussions (yellow-highlighted in **Line 565-581, Page 29-31**) have been added in the revised manuscript as follows:

“As shown in **Fig. 7d**, the FE gradually increases with the increasing water pressure and reaches up to 91% at 40% H₂O-air atmosphere. Furthermore, As the current density increases, the FE gradually increases to a maximum value of 96% at -1.4 A cm⁻². However, when the current density further increased to -2.2 A cm⁻², the FE dropped to 85%. This drop can be attributed to the positive correlation between the electronic conductivity of the electrolyte and oxygen partial pressure, leading to increased electronic losses^{13,75}. Interestingly, the final FE remained within a reasonable range of approximately 85% (**Fig. 7e**, **Supplementary Fig. 33**). However, acceptable FE indicates that C/H-BSCF oxide as air electrode has a good potential for commercialization.”

Comment 3:

Multiple comparisons of particle sizes are made in the text, such as mechanical mixing generating larger particles of C/H-BSCF-PM than C/H-BSCF, and line-scan distance may need to be compared with particle size to make the distribution of two phases more apparent, and thus strengthen the persuasiveness from this comparison.

Response to C3:

We appreciate the reviewer's thoughtful comments. We conducted additional studies of the C/H-BSCF-PM composite to increase the persuasiveness of this comparison. The particle size of the C/H-BSCF-PM powder is in the micrometer range, as shown in **Fig. R9a**. We employed energy dispersive x-ray (EDX) spectral line scan detection of the A-site Ba element and the B-site Co element and used the difference in the atomic ratio of the A/B site to identify the two phases because the elemental compositions of the C-BSCF and H-BSCF phases are the same. As seen in **Fig. R9b**, along the direction of the line sweep, the CO content is first higher and then lower, while the Ba content is first lower and then higher, indicating that particles A and B are cubic C-BSCF and hexagonal H-BSCF, respectively. The two phases of the physically mixed C/H-BSCF-PM electrode may be seen to be spread at the micrometer scale. In contrast, the two phases of the hybrid electrode C/H-BSCF prepared by the one-step method are uniformly distributed on the nanoscale (**Fig. 2d**). As a result, the self-assembled produced hybrid electrode has a smaller per-phase size than the physically mixed prepared electrode with greater particle size of the phases, which promotes the accelerated surface mass transfer and ion exchange of the air electrode. Meanwhile, in contrast to the C/H-BSCF-PM electrode, which acts as an ion transport channel through mutual contact between particles, the two phases in C/H-BSCF are distributed in a single particle with a strongly interacting interface between the phases, allowing the air electrode to improve its electrochemical performance by increasing the kinetic rate of the reaction.

Furthermore, Song *et al.* created highly active nanocomposite electrodes $\text{BaCo}_{0.7}(\text{Ce}_{0.8}\text{Y}_{0.2})_{0.3}\text{O}_{3-\delta}$ (BCCY) and confirmed that the composite BCCY has significantly inferior two-phase distribution and electrochemical activity than self-assembled BCCY [Song *et al.*, *Joule* 3 2842 (2019)]. By comparing the BSCF-BZCYYb composite electrodes prepared by the one-pot method and the physical mixing method, Liu *et al.* discovered that the self-assembled prepared electrode had better two-phase distribution and particle size, which significantly improved the kinetic rate of the reaction on the surface of the air electrode and ionic diffusion [Liu *et al.*, *Appl. Catal. B: Environ.* 319 121929 (2022)]. Finally, we have updated parts of the description and added the EDX line sweep profiles of the C/H-BSCF-PM composite in the revised manuscript.

Fig. R9 (also **Supplementary Fig. 12**) a SEM image and (b) EDX spectroscopy line-scan profiles of C/H-BSCF-PM.

Related data (**Supplementary Fig. 12**) and discussions (yellow-highlighted in **Line 299-303, Page 16**) have been added in the revised manuscript as follows:

“This is primarily attributed to the fact that mechanical mixing results in a poorer distribution and interface of the two phases, which can negatively impact ion transport and surface diffusion, thereby constraining the oxygen activation rate of the electrode (Fig. 3e, Supplementary Fig. 12)^{17,21}.”

Comment 4:

The author mentions that R_p exhibits a lower activation energy of 0.73 eV in Fig. 6b but does not provide specific comparative data. It is recommended to either indicate the comparative values mentioned in the

text or provide a reasonable explanation for the absence of such data.

Response to C4:

We sincerely thank the reviewer for the thoughtful comment. In the subsequent paragraph of the manuscript, we provided a comprehensive comparison and description of R_p in single cells with C-BSCF, H-BSCF and C/H-BSCF air electrodes. As shown in **Fig. R10 (Supplementary Fig. 28)**, the activation energies of C-BSCF, H-BSCF composite electrode exhibits a lower activation energy, which is attributed to the favorable synergistic effect of the hybrid electrode, resulting in better catalytic activity compared to the single-phase air electrodes within the investigated temperature range. These results undoubtedly suggest promising prospects for the application of the C/H-BSCF composite electrode, particularly at lower temperatures. Finally, we removed the excessive discussion of **Fig. 6b** in the revised manuscript.

Fig. R10 (also **Supplementary Fig. 28**) Arrhenius plots of the R_p of the C-BSCF, C/H-BSCF and H-BSCF air electrodes in FC mode.

Related discussions (yellow-highlighted in **Line 506-511, Page 27**) have been modified in the revised manuscript as follows:

*“The total resistance (R_t), R_p , and ohmic resistance (R_o) dependency with temperature corresponding to the EIS of this cell at open circuit voltage (OCV) were summarized in **Fig. 6b**. At 650 $^{\circ}\text{C}$, R_p is only 0.09 $\Omega \text{ cm}^2$, and the major component of its R_i is bigger R_o (0.28 $\Omega \text{ cm}^2$), with the decrease of temperature, at*

500 °C, R_p and R_o are 0.56 and 0.49 $\Omega \text{ cm}^2$, respectively. It should be noted that the resistance of the cell in actual operation should be even lower than that obtained under OCV conditions.”

Reviewer #3 (Remarks to the Author):

In this work, Liu et al. developed a dual-phase air electrode for R-PCECs, consisting with self-assembled H-BSCF and C-BSCF, which shows a good potential as air electrode. They found that BSCF-1.5 has the best performance and attribute it to the synergistic effect. This work is interesting and well-organized. However, it lacks of novelty since there are some similar works reported self-assembled air electrodes, and there are also some questions need to be solved. I suggest a major revision of this work.

Response:

We extend our sincere appreciation to the reviewer for positive feedback and recognition of the value of our research. We are also deeply grateful for the numerous insightful and critically important comments raised by the reviewer, as they have significantly contributed to improving the quality and rigor of our manuscript. Preparation of air electrodes using self-assembly strategy has outstanding advantages over conventional physical mixing method, which enables excellent phases distribution and strong interactions between multiphases, thus accelerating the kinetic rate of air electrode reaction. It is worth noting that numerous studies have reported the synthesis of air electrodes using self-assembly method, effectively demonstrating the efficacy and efficiency of this strategy. In our study, we have innovatively constructed hybrid air electrodes by precisely modulating the molar ratios of ions at the A and B sites of perovskite materials. This novel method can achieve the tuning of the content of each phase in the hybrid electrode, which is different from the conventional self-assembly method that is not controllable for the composition and content of the phases in the composite electrode. By finely tuning the A and B site ion ratios, we have successfully designed a hybrid electrode, denoted as C/H-BSCF (BSCF-1.5), which exhibits optimal catalytic activity and showcases exceptional electrochemical performance in both fuel cell and electrolysis cell modes on R-PCECs. Additionally, we have thoroughly investigated the synergistic effects of C-BSCF and H-BSCF in the hybrid electrode C/H-BSCF. Lastly, we have taken every comment into careful

consideration and made revisions to our manuscript accordingly. To address all the comments in a comprehensive manner, we have prepared point-to-point responses, which will be provided below.

Comment 1:

It is proposed that C/H-BSCF benefits from the reduced mismatched thermal stress between multiple phases, but the TEC data of C-BSCF and H-BSCF are lacking.

Response to C1:

We appreciate the reviewer's detailed comment, as it has been immensely helpful in refining our work. In response to the reviewer's feedback, we have made revisions to the manuscript, including the addition of thermal expansion coefficients (TEC) data for various air electrode materials. First, we have focused on investigating the TEC of different air electrodes, which play a critical role in determining the thermal compatibility between air electrodes and electrolytes. To illustrate this, **Figure R11** has been included to showcase the linear expansion (dL/L_0) of C-BSCF, C/H-BSCF, C/H-BSCF-PM, and H-BSCF samples with temperature. The TEC values of C-BSCF and H-BSCF were found to be $23.8 \times 10^{-6} \text{ K}^{-1}$ and $19.2 \times 10^{-6} \text{ K}^{-1}$, respectively. The TEC of cubic C-BSCF aligns with previous literature reports [*Song et al., Joule 3 2842 (2019)*], while hexagonal H-BSCF exhibits a lower TEC than C-BSCF. Moreover, the TEC values of the C/H-BSCF and C/H-BSCF-PM hybrids were measured to be $19.7 \times 10^{-6} \text{ K}^{-1}$ and $20.4 \times 10^{-6} \text{ K}^{-1}$, respectively. As expected, these hybrid electrodes demonstrated TEC values that lie between those of C-BSCF and H-BSCF. This finding suggests that the hybrid electrode possesses better thermal compatibility with the electrolyte compared to most traditional Co-based electrodes. Notably, its TEC is lower than that of multiple advanced Co/Fe-based perovskite air electrodes. These additions to our study further enhance our understanding of the thermal characteristics and compatibility of the different air electrode materials examined. (**Fig. R1b, c**).

Fig. R11 (also **Supplementary Fig. 22a**) The results of dL/L_0 for C-BSCF, C/H-BSCF, C/H-BSCF-PM and H-BSCF from 40 to 1000 °C.

Related data (**Supplementary Fig. 22a**) and discussions (yellow-highlighted in **Line 413-422, Page 22**) have been added in the revised manuscript as follows:

*“We investigated the thermal expansion coefficients (TEC) of various air electrodes in order to confirm that C/H-BSCF, a self-assembled composite electrode with two phases having the same elemental composition, can significantly reduce the thermal stress to improve the match with the electrolyte. **Supplementary Fig. 22a** shows the linear expansion (dL/L_0) of the C-BSCF, C/H-BSCF, C/H-BSCF-PM and H-BSCF samples with temperature. the TEC of C-BSCF and H-BSCF were $23.8 \times 10^{-6} \text{ K}^{-1}$ and $19.2 \times 10^{-6} \text{ K}^{-1}$, respectively, where the TEC of cubic C-BSCF is consistent with literature reports¹⁸, while hexagonal H-BSCF exhibits a lower TEC than C-BSCF. As expected, the TEC of the C/H-BSCF and C/H-BSCF-PM hybrids, which were $19.7 \times 10^{-6} \text{ K}^{-1}$ and $20.4 \times 10^{-6} \text{ K}^{-1}$, respectively, were intermediate between those of C-BSCF and H-BSCF.”*

Comment 2:

From TEM results, we can see that the interface of C-BSCF and H-BSCF consists of C-BSCF (110) and H-BSCF (002). The atomic arrangement of two phases in this interface is so different that there should be an obvious strain in the surface region. Maybe the improved catalysis activity is aroused by the strain

effect and should be discussed.

Response to C2:

We appreciate the reviewer's comment. The interface between the C-BSCF (110) and H-BSCF (002) phases in the C/H-BSCF hybrid electrode exhibits a distinct atomic arrangement. As the reviewer pointed out, strain effect due to different atomic arrangements is advantageous for enhancing catalytic activity. Strain effects have become an important strategy for regulating surface chemistry and optimizing nano catalysts' catalytic performance. For example, Wang *et al.* enabled the intrinsic activity of the catalysts by using lattice strain to change the coordination environment of cobalt atoms could regulate the d-band center, thus balancing the adsorption and intrinsic catalytic effects of polysulfides [Wang *et al.*, *Adv. Mater.* 34, 2204403 (2022)]. Furthermore, Meng *et al.* achieved atomic-layer IrO_x on an adjustable Ir-O bond length IrCo nano-branch by utilizing compressive strain effects. The precisely controlled compressive strain effects balance the interactions between adsorbates and substrates and promote the rate-determining step for HOO* formation [Meng *et al.*, *Adv. Mater.* 31, 1903616 (2019)]. However, accurate assessment of strain is challenging due to the thermal expansion of multiphases under high-temperature testing conditions, as well as the fluctuation of thermal strain with temperature oscillation. The interface effects caused by thermal strain also exhibit enhanced influence on the catalytic activity of air electrodes.

Previous investigations on interfacial effects in multi-phase catalysts have demonstrated their potential to significantly improve catalytic activity. For example, Zhang *et al.* reported a dual-phase cathode material, double perovskite structure PrBaCo₂O_{5+δ} (PBC) and fluorite structure Gd_{0.1}Ce_{0.9}O_{2-δ} (GDC), successfully synthesized using a one-pot method, with remarkable oxygen reduction reaction activity [Zhang *et al.*, *J. Mater. Chem. A* 10 3495 (2022)]. Meanwhile, it was verified that the formation of a coherent interfacial structure between PBC and GDC particles not only facilitates the slowing down of lattice thermal expansion but also directs the oxygen transport among PBC particles with different orientations. Unfortunately, evaluating the k_{chem} of air electrodes composited with various ratios of GDC makes it impossible to demonstrate definitively how the interfacial structure between PBC and GDC facilitates oxygen ion conduction. A productive ORR catalyst was also created by Zhao *et al.* using mullite SmMn₂O₅, O-deficient perovskite BaMnO_{3-δ}, and MnO_x [Zhao *et al.*, *Nano Energy* 51 91 (2018)]. The charge transfer

at the interface between $\text{BaMnO}_{3-\delta}$ and SmMn_2O_5 was investigated to demonstrate that this interface is more favorable for the enhancement of ORR activity compared with the pure phase SmMn_2O_5 . Lian *et al.* used first-principles calculations to examine the impact of the heterostructure of the perovskite $(\text{Nd,Sr})\text{CoO}_3$ /Ruddlesden-Popper (R-P) oxide $(\text{Nd,Sr})_2\text{CoO}_4$ on ORR activity [Lian *et al.*, *J. Phys. Chem. Lett.* 14 2869 (2023)]. It is confirmed that the more distant rock-salt layers on the heterointerfaces can facilitate the insertion of interstitial oxygen and form a high-speed transport channel of interstitial oxygen. Even though the work mentioned above has demonstrated that such heterogeneous interfaces are advantageous for enhancing ORR activity from different aspects, there hasn't been a thorough examination of phase interfaces in many of the advanced composite air electrodes that have been published thus far. such as $\text{Sr}_x(\text{Y}_y(\text{Nb}_{0.1}\text{Co}_{0.9})_{1-y})\text{O}_{3-\delta}-\text{Y}_2\text{W}_3\text{O}_{12}$ [Zhang *et al.*, *Nature* 591 246 (2021)], $(\text{La}_{0.6}\text{Sr}_{0.4})_{0.95}\text{Co}_{0.2}\text{Fe}_{0.8}\text{O}_{3-\delta}-\text{SrMoO}_4$ [Zhuang *et al.*, *Nat. Catal.* 5 300 (2022)], $\text{Ba}_{0.5}\text{Sr}_{0.5}(\text{Co}_{0.7}\text{Fe}_{0.3})_{0.6875}\text{W}_{0.3125}\text{O}_{3-\delta}$ [Shin *et al.*, *Nat. Energy* 2 1 (2017)], $\text{Ba}_{0.9}\text{Co}_{0.7}\text{Fe}_{0.2}\text{Nb}_{0.1}\text{O}_{3-\delta}$ [Pei *et al.*, *Nat. Commun.* 13 1 (2022)], $\text{PrBa}_{0.8}\text{Ca}_{0.2}\text{Co}_2\text{O}_{5+\delta}$ [Chen *et al.*, *Joule* 2 938 (2018)] and $\text{BaCo}_{0.7}(\text{Ce}_{0.8}\text{Y}_{0.2})_{0.3}\text{O}_{3-\delta}$ [Song *et al.*, *Joule* 3 2842 (2019)]. The complexity of the ORR/OER process on the composite air electrode surface makes it difficult to experimentally discern the specific roles played by the different heterogeneous interfaces in the reaction. Moreover, there is currently a lack of suitable high-temperature in-situ characterization techniques to thoroughly study the electrochemical properties of these interfaces. Additionally, enhancing the ORR and OER rates of air electrodes primarily relies on achieving a synergistic effect between the phases, which influences both the surface catalytic processes and bulk phase diffusion dynamics in composite air electrodes [Hu *et al.*, *Prog. Mater. Sci.* 133 101050 (2023)].

Nonetheless, however, we can also observe the impact of interfacial effects on catalytic activity by comparing the electrochemical performance of physically mixed air electrode C/H-BSCF-PM and hybrid air electrode C/H-BSCF in this study. Unfortunately, due to the structural complexity and high atomic number of H-BSCF ($\text{Ba}_4\text{Sr}_4(\text{Co}_{0.8}\text{Fe}_{0.2})_4\text{O}_{16-\delta}$), constructing effective heterogeneous interfacial models of H-BSCF and C-BSCF, as well as conducting DFT calculations, are challenging and extensive tasks. We would like to express our gratitude for the valuable comment provided by the reviewer, and we have made

the necessary revisions to the manuscript based on this feedback.

Related discussions (yellow-highlighted in **Line 195-198, Page 11**) have been added in the revised manuscript as follows:

“In the C/H-BSCF hybrid, the strain and interfacial effects yielded by the different atomic arrangements at the interface of the C-BSCF (110) and H-BSCF (002) phases have positively contributed to the enhancement of the catalytic activity³⁰⁻³².”

Comment 3:

When described XRD refinement results of H-BSCF (line 169 in page 9), it should be $a=b= 11.7115(7) \text{ \AA}$ but not $a=b=c= 11.7115(7) \text{ \AA}$

Response to C3:

We sincerely thank the reviewer for the thoughtful comments. We're sorry for this error. In the revised manuscript, it has been modified to $a=b= 11.7115(7) \text{ \AA}$.

Comment 4:

EIS and DRT results in Figure 3 should be discussed further. Activation energies of different processes (HF, IF and LF) should be calculated respectively. It is encouraged to test EIS at various P_{O_2} and to perform DRT analysis to probe the rate-determine step of C-BSCF, H-BSCF and C/H-BSCF, which may be different among three electrodes.

Response to C4:

We very much appreciate the reviewer very much for this comment. We have supplemented the revised manuscript with a more detailed discussion of the DRT analysis results. **Figure R12** shows the DRT plots for the C-BSCF, C/H-BSCF and H-BSCF electrodes measured at 600 °C in wet air. The DRT plots can be divided into the peaks distributed in low frequency (LF, <10 Hz), intermediate frequency (IF, 10-2000 Hz), and high frequency (HF, >2000 Hz), which are normally associated with the gas diffusion process, ion migration and surface exchange, and charge transfer process, respectively [Niu *et al.*, *Adv. Energy Mater.* 12 2103783 (2022); He *et al.*, *Adv. Mater.* 35 e2209469 (2023)]. Compared to the LF and HF regions, the

resistance in the IF region constitutes the major part of the total resistance, indicating that the surface exchange and bulk diffusion rates of the air electrode, specifically the surface adsorption and dissociation of oxygen and water as well as the bulk diffusion rates of oxygen ions and protons, are the major factors limiting the electrocatalytic process. The peak area of the hybrid electrode C/H-BSCF in the IF region is not between but smaller than that of the single-phase C-BSCF and H-BSCF electrodes. This not only demonstrates the excellent surface exchange and bulk diffusion rates of the C/H-BSCF electrode but also highlights the advantages of the synergistic effect of the dual phase in the ORR and hydration reactions on the air electrode surface.

Fig. R12 (also **Fig. 3c**) DRT analysis of EIS measured at C-BSCF, C/H-BSCF and H-BSCF electrodes at 600 °C.

Figure R13 shows the temperature dependence of the ASR of C-BSCF, C/H-BSCF and H-BSCF electrodes in HF, IF and LF regions. For the three different electrodes the ASR in the IF region remains predominant in the tested temperature range of 700-500 °C. The activation energy of the C/H-BSCF electrode is 1.24 eV, which is in the middle of the activation energies of C-BSCF and H-BSCF. This is mostly due to the different rates of ionic conduction promotion between the cubic and hexagonal phases in the C/H-BSCF hybrid electrode. The impedance increase is most pronounced in the LF region with decreasing temperature, and the activation energies of the C-BSCF, C/H-BSCF and H-BSCF electrodes

in the LF region are all around 1.73 eV. In the HF region, the C/H-BSCF electrode has the lowest activation energy of 0.45 eV, indicating a fast charge transfer and oxygen activation rate at the three-phase boundary (TPB) compared to the single-phase C-BSCF and H-BSCF.

Fig. R13 (also **Supplementary Fig. 9**) Temperature dependence of ASRs of the (a) C-BSCF, (b) C/H-BSCF and (c) H-BSCF electrodes under wet air (3% H₂O).

Thanks to the reviewer for such insightful comment. To understand and compare the surface reactions of C-BSCF, C/H-BSCF and H-BSCF air electrodes under practical conditions, here we supplemented the testing of the EIS of symmetric cells with different air electrodes at different oxygen partial pressures (P_{O2}) and used DRT analyses for a detailed discussion of the different processes. As shown in **Fig. R14a,b,c**, the ORR kinetic of the C-BSCF, C/H-BSCF and H-BSCF air electrodes were investigated by measuring the EIS at 600 °C under different P_{O2}. When the air electrode was exposed to pure oxygen, as expected, the symmetric cell demonstrated a relatively low R_p. As the P_{O2} decreased, the R_p showed a significant increase. Shown in **Fig. R14d,e,f**, are the DRT plots of the electrochemical processes for C-BSCF, C/H-BSCF and H-BSCF at different P_{O2} at 600 °C. The integral area of each process corresponds to the polarization resistance of each process, and the general relationship between R_p and P_{O2} follows the equation $R_p = k(P_{O_2})^{-n}$. The C/H-BSCF hybrid electrode exhibits the lowest R_p at different oxygen partial pressures at 600 °C. However, in the HF region, the C-BSCF, C/H-BSCF and H-BSCF electrodes show similar R_p and n values of about 0.2, indicating that the charge transfer processes of different electrodes do not significantly discrepancy. The lower R_p of C/H-BSCF electrode is mainly attributed to the decrease in resistance of the IF and LF regions compared to C-BSCF and H-BSCF electrodes. As shown in **Fig.**

R14g,h,i, C-BSCF and H-BSCF electrodes exhibit lower R_p in the IF and LF regions, respectively, indicating that the rapid ORR rate of the C/H-BSCF hybrid electrode is mainly attributed to the promotion of oxygen ion diffusion rate and surface mass transfer by the cubic and hexagonal phases, respectively.

Fig. R14 (also **Supplementary Fig. 10**) **a,b,c** EIS and (**d,e,f**) DRT of C-BSCF, C/H-BSCF and H-BSCF electrodes as a function of oxygen partial pressure. **g,h,i** Dependence of R_p on oxygen partial pressure for C-BSCF, C/H-BSCF and H-BSCF electrodes in the HF, IF and LF regions.

Related data (**Supplementary Fig. 9,10**) and discussions (yellow-highlighted in **Line 254-282, Page 14-15**) have been added in the revised manuscript as follows:

“As depicted in Fig. 3c, compared to the LF and HF regions, the resistance in the IF region primarily governs the total resistance, indicating that the surface exchange and bulk diffusion rates of the air electrode, specifically the surface adsorption and dissociation of oxygen and water, as well as the bulk diffusion rates of oxygen ions and protons, significantly limit the electrocatalytic process. Interestingly, the peak area of the hybrid electrode C/H-BSCF in the IF region is not between, but lower than that of the

single-phase C-BSCF and H-BSCF electrodes. This not only demonstrates the excellent surface exchange and bulk diffusion rates of the C/H-BSCF electrode but also highlights the advantages of the synergistic effect from the dual phase in the ORR and hydration reaction on the air electrode surface. Meanwhile, the temperature dependence of ASR for C- BSCF, C/H-BSCF, and H-BSCF electrodes in the HF, IF, and LF regions is illustrated in **Supplementary Fig. 9**. The activation energy of the C/H-BSCF electrode in the IF region is 1.24 eV, which is in the middle of the activation energies of C-BSCF and H-BSCF. This is mainly due to the different ionic conduction promotion rates of the cubic and hexagonal phases in the C/H-BSCF hybrid electrode³⁹.

To understand and compare the surface reactions of C-BSCF, C/H-BSCF and H-BSCF air electrodes under practical conditions, the EIS of symmetric cells with different air electrodes at different partial pressures of oxygen (P_{O_2}) were tested and analyzed in conjunction with DRT for different processes (**Supplementary Fig. 10**). When the air electrode was exposed to pure oxygen at 600 °C, as expected, the symmetric cell demonstrated a relatively low R_p . As the P_{O_2} decreased, the R_p showed a significant increase. The C/H-BSCF hybrid electrode exhibits the lowest R_p at different oxygen partial pressures at 600 °C. However, in the HF region, the C-BSCF, C/H-BSCF and H-BSCF electrodes show similar R_p and n values of about 0.2, indicating that the charge transfer processes of different electrodes do not significantly discrepancy^{9,10}. The lower R_p of C/H-BSCF electrode is mainly attributed to the decrease in resistance of the IF and LF regions compared to C-BSCF and H-BSCF electrodes, which further suggests that the fast ORR rate of the C/H-BSCF hybrid electrodes is mainly attributed to the facilitated diffusion rate of the oxygen ions and the surface mass transfer by the cubic and hexagonal phases, respectively.”

Comment 5:

Activation energy of electrical conductivity should be calculated and discussed.

Response to C5:

We are grateful to the reviewer for this comment. We have calculated the activation energy of electrical conductivity and discussed it in the revised manuscript. Temperature dependence of the electrical conductivity (σ) for C-BSCF, C/H-BSCF and H-BSCF is presented as an Arrhenius-type plot as shown in

Fig. R15. The hexagonal H-BSCF ($0.3\text{-}3.6\text{ S cm}^{-1}$) had weak conductivity at $300\text{-}800\text{ }^{\circ}\text{C}$, while the hybrid C/H-BSCF ($5.9\text{-}30.9\text{ S cm}^{-1}$) is second only to the cubic C-BSCF ($23.2\text{-}65.3\text{ S cm}^{-1}$). In addition, all samples exhibit thermally activated semiconductor behavior in the range of $300\text{-}800\text{ }^{\circ}\text{C}$. The activation energies of C-BSCF, C/H-BSCF and H-BSCF are 0.276 , 0.305 , and 0.351 eV in the range of $300\text{-}500\text{ }^{\circ}\text{C}$, respectively. It is noteworthy that a turning point occurs around $500\text{ }^{\circ}\text{C}$, suggesting different conductive mechanisms in the $300\text{-}500\text{ }^{\circ}\text{C}$ and $500\text{-}800\text{ }^{\circ}\text{C}$ ranges. This is mainly attributed to the fact that the conductive mechanism is closely related to small polaron hopping behavior, which is affected by the combination of the oxygen release and the carrier migration rate [Zhong *et al.*, *Chem. Eng. J.* 425 131822 (2021); Huan *et al.*, *ChemSusChem* 13 4994 (2020)]. Due to its larger concentration of oxygen vacancies and lower B-site average cation valence at high temperatures, H-BSCF exhibits the highest activation energy in both the $300\text{-}500\text{ }^{\circ}\text{C}$ and $500\text{-}800\text{ }^{\circ}\text{C}$ ranges [Jo *et al.*, *Chem. Eng. J.* 451 138954 (2023)].

Fig. R15 (also **Fig. 4i**) Arrhenius plots of electrical conductivity of C-BSCF, C/H-BSCF and H-BSCF samples in air at $300\text{-}800\text{ }^{\circ}\text{C}$.

Related data (**Fig. 4i**) and discussions (yellow-highlighted in **Line 401-409, Page 21**) have been added in the revised manuscript as follows:

“The hexagonal H-BSCF ($0.3\text{-}3.6\text{ S cm}^{-1}$) had weak conductivity at 300-800 °C, while the hybrid C/H-BSCF ($5.9\text{-}30.9\text{ S cm}^{-1}$) is second only to the cubic C-BSCF ($23.2\text{-}65.3\text{ S cm}^{-1}$) (Fig. 4i)⁵³. In addition, all samples exhibit thermally activated semiconductor behavior in the range of 300-800 °C. The activation energies of C-BSCF, C/H-BSCF and H-BSCF are 0.276, 0.305, and 0.351 eV in the range of 300-500 °C, respectively. It is noteworthy that a turning point occurs around 500 °C, suggesting different conductive mechanisms in the 300-500 °C and 500-800 °C ranges. This is mainly attributed to the fact that the conductive mechanism is closely related to small polaron hopping behavior, which is affected by the combination of the oxygen release and the carrier migration rate.”

Comment 6:

Proton conductivity is essential for air electrodes of R-PCECs, which should be explicitly measured but not discussed roughly by H₂O-TPD (line 344 in page 18).

Response to C6:

We thank the reviewer for his suggestion to correct our misrepresentation. We have removed inappropriate discussion from the revised manuscript. As the reviewer pointed out, proton conductivity is of paramount importance for air electrodes in R-PCECs. Although H₂O-TPD serves as a significant approach to investigate the hydration capacity of air electrodes, it does not directly reflect the proton conductivity of the material since the hydration capacity solely indicates the ability of air electrodes to uptake protons. In this study, we performed tests on the proton diffusion capability of the hybrid C/H-BSCF and cubic C-BSCF by utilizing dense hydrogen-permeable membrane. to ensure reliable hydrogen permeation results, both the membrane opposing surfaces were coated with 500 nm-thick Pd film by magnetron sputtering as reported [Zhou et al., *J. Mater. Chem. A* 7 13265 (2019)]. After the sputtering, the membranes were further calcined in an argon atmosphere at 800 °C for 2 h. This is done to prevent the reduction of the perovskite hydrogen permeation membrane and simultaneously enhance the surface hydrogen catalytic rate. As illustrated in **Fig. R16a**, the C/H-BSCF hydrogen-permeable membrane has a large H₂ permeation flux at 450-650 °C, indicating that the hybrid C/H-BSCF has superior proton diffusion performance. For example, the hydrogen permeation flux of C/H-BSCF was 0.039 mL cm⁻² min⁻¹ at 600 °C, whereas the flux of C-

BSCF exhibited only 0.031 mL cm⁻² min⁻¹. Additionally, the proton conductivity of cubic C-BSCF and hybrid C/H-BSCF was computed for quantitative analysis using Eq. 1 and is depicted in **Fig. R16b**.

$$\sigma_{H^+} = J_{H_2} \frac{4F^2L}{RT} \left/ \ln \frac{P_{H_2, \text{supp.}}}{P_{H_2, \text{perm.}}} \right. \quad (1)$$

Where J_{H_2} is H₂ permeation flux (mol cm⁻² s⁻¹), F is the Faraday constant (96485.3326 C mol⁻¹), L is the membrane thickness, R is the ideal gas constant (8.314 J K⁻¹ mol⁻¹), T is the temperature, $P_{H_2, \text{supp.}}$ is H₂ partial pressure at the feed side, and $P_{H_2, \text{perm.}}$ is H₂ partial pressure at the permeate side. In comparison to cubic C-BSCF, hybrid C/H-BSCF had significantly higher proton conductivities, measuring 0.033, 0.029, 0.026, 0.020, and 0.015 S cm⁻¹ at 650, 600, 550, 500, and 450 °C, respectively.

Figure R16 (also **Fig.4h** and **Supplementary Fig. 21**) **a** H₂ permeation fluxes of C-BSCF and C/H-BSCF hydrogen permeable membranes at 450-650 °C. **b** Proton conductivities of C-BSCF and C/H-BSCF at 450-650 °C

Related data (**Supplementary Fig. 21**) and discussions (yellow-highlighted in **Line 391-398, Page 21**) have been added in the revised manuscript as follows:

*“To ensure reliable hydrogen permeation results, both sides of the membrane were coated with a 500 nm-thick Pd film using magnetron sputtering (**Supplementary Fig. 21**)⁵⁸. The dual-phase C/H-BSCF hydrogen permeation membrane has a larger H₂ permeation flux at 450-650 °C, indicating that the hybrid C/H-BSCF has superior proton diffusion performance. For example, the hydrogen permeation flux of C/H-BSCF was 0.039 mL cm⁻² min⁻¹ at 600 °C, whereas the flux of C-BSCF exhibited only 0.031 mL cm⁻² min⁻¹”*

[†]. Meanwhile, the corresponding proton conductivities of C/H-BSCF and C-BSCF were also calculated as shown in Supplementary Fig. 21.”

Comment 7:

The migration ability of oxygen is discussed in line 324. However, O₂-TPD is not the direct evidence of it, ECR result is more accurate.

Response to C7:

Thanks for the reviewer’s comment. We fully agree with the reviewer that O₂-TPD cannot be considered as direct evidence for assessing the oxygen migration ability of materials, because the temperature and amount of oxygen desorption do not form a direct correlation with the rate of oxygen migration under different temperature conditions. Therefore, according to the reviewer's suggestion, we changed the description in the revised manuscript. In fact, it has been frequently documented that O₂-TPD is utilized to characterize indirectly the ability of air electrodes to migrate oxygen. The rationale is that a weaker metal-oxygen connection, which is more conducive to oxygen migration in the bulk phase, is suggested by the lower initial temperature of oxygen desorption. Again, similar defenses have been cited in several recent studies. [Song et al., *Joule* 3 2842 (2019); Ding et al., *Nat. Commun.* 11 1970 (2020); Liang et al., *Adv. Mater.* 34, 2106379 (2021); Niu et al., *Appl. Catal. B Environ.* 270 118842 (2020); Tang et al., *Small* 18 e2201953 (2022)].

Related discussions (yellow-highlighted in **Line 357-361, Page 19**) have been added in the revised manuscript as follows:

“The initial desorption temperature and amount of oxygen desorption are critical indications of oxygen activation and oxygen vacancy content of the air electrodes under operating conditions²². The hybrid C/H-BSCF has a lower initial desorption temperature and higher resolved oxygen content compared to the cubic C-BSCF (Fig. 4e).”

Comment 8:

About ECR:

- a) ECR results of H-BSCF should be provided to verify the synergistic effect of C/H-BSCF by comparing it with C-BSCF and H-BSCF. It is encouraged to perform ECR tests of PM-C/H-BSCF.
- b) The raw data curves and fitting curves in Figure S12 should be provided to verify the accuracy of fitting.
- c) Sintering temperature of bar samples should be given in the experimental section, as well as SEM pictures of the surface and cross-section of bars.
- d) When fabricating bar samples for ECR, the densification procedure requires a high sintering temperature, which may raise the change in composition or crystal structure. XRD and refinement results of sintered bars should be provided and discussed.

Response to C8:

We thank the reviewer for these important concerns.

- a) ECR results of H-BSCF should be provided to verify the synergistic effect of C/H-BSCF by comparing it with C-BSCF and H-BSCF. It is encouraged to perform ECR tests of PM-C/H-BSCF.

Response to C8a:

We fully agree with the reviewer's comment and to further validate the synergistic effect of C/H-BSCF we performed ECR tests on H-BSCF and C/H-BSCF samples as well. The D_{chem} and k_{chem} values were estimated via ECRTOOLS from the ECR curves. **Figure R17** displays D_{chem} and k_{chem} values of C-BSCF, C/H-BSCF, C/H-BSCF-PM and H-BSCF from 500 to 700 °C, obtained based on the conductivity relaxation method. As expected, at each temperature, both D_{chem} and k_{chem} of C/H-BSCF are much higher than that of C-BSCF, C/H-BSCF-PM and H-BSCF. For instance, at 700 °C, the D_{chem} and k_{chem} of the C/H-BSCF oxides were $4.88 \times 10^{-4} \text{ cm}^2 \text{ s}^{-1}$ and $4.69 \times 10^{-3} \text{ cm s}^{-1}$, respectively, whereas they were $2.27 \times 10^{-4} \text{ cm}^2 \text{ s}^{-1}$ and $2.51 \times 10^{-3} \text{ cm s}^{-1}$ for C-BSCF. The lowest D_{chem} and k_{chem} values were found for H-BSCF, which were $4.25 \times 10^{-5} \text{ cm}^2 \text{ s}^{-1}$ and $4.15 \times 10^{-4} \text{ cm}^2 \text{ s}^{-1}$, respectively. The oxygen surface exchange and bulk phase diffusion rates of H-BSCF are noticeably better than those of C-BSCF when the temperature decreases at 550 and 500 °C, despite the fact that the D_{chem} and k_{chem} values of C-BSCF are higher than those of H-BSCF at 700 °C. The D_{chem} and k_{chem} of the C/H-BSCF-PM composite oxide were also investigated under the same conditions as those depicted in **Fig. R17a, b**. In the tested temperature range, the D_{chem} and k_{chem} values of C/H-BSCF-PM were lower than those of self-assembled synthesized C/H-BSCF, which may

have been attributed to the poorer phase distribution of the physically mixed samples, which impeded the synergistic effect of the two phases thus unfavorable to ion migration and surface exchange. Finally, we added and discussed relevant data in the revised manuscript as well.

Fig. R17 (also **Fig. 4f** and **Supplementary Fig. 19**) Arrhenius plots of the (a) D_{chem} and (b) k_{chem} for C-BSCF, C/H-BSCF, C/H-BSCF-PM and H-BSCF oxides from 500 to 700 °C.

Related data (**Supplementary Fig. 19**) and discussions (yellow-highlighted in **Line 362-381, Page 19-20**) have been added in the revised manuscript as follows:

*“The D_{chem} and k_{chem} values of C-BSCF, C/H-BSCF, C/H-BSCF-PM and H-BSCF in the temperature range of 500 to 700 °C, determined using the conductivity relaxation method (**Supplementary Fig. 16**), are depicted in **Fig. 4f** and **Supplementary Fig. 19**⁵³. All samples were calcined to dense bars and the phase structure and composition remained stable (**Supplementary Fig. 17,18**). As anticipated, at each temperature, both D_{chem} and k_{chem} of C/H-BSCF exhibit significantly higher values compared to C-BSCF, C/H-BSCF-PM, and H-BSCF. For instance, at 700 °C, the D_{chem} and k_{chem} values of the C/H-BSCF oxides were measured to be $4.88 \times 10^{-4} \text{ cm}^2 \text{ s}^{-1}$ and $4.69 \times 10^{-3} \text{ cm s}^{-1}$, respectively, whereas corresponding values for C-BSCF were $2.27 \times 10^{-4} \text{ cm}^2 \text{ s}^{-1}$ and $2.51 \times 10^{-3} \text{ cm s}^{-1}$. On the other hand, H-BSCF exhibited the lowest D_{chem} and k_{chem} values, which were $4.25 \times 10^{-5} \text{ cm}^2 \text{ s}^{-1}$ and $4.15 \times 10^{-4} \text{ cm}^2 \text{ s}^{-1}$, respectively. Even though the D_{chem} and k_{chem} values of C-BSCF were higher than those of H-BSCF at 700 °C, the oxygen surface exchange and bulk phase diffusion rates of H-BSCF were noticeably superior to C-BSCF when*

the temperature dropped to 550 and 500 °C. Additionally, the D_{chem} and k_{chem} values of the C/H-BSCF-PM composite oxide were also explored under the same conditions as depicted in Fig. 4f and Supplementary Fig. 19. Throughout the tested temperature range, the D_{chem} and k_{chem} values of C/H-BSCF-PM were lower than those of self-assembled synthesized C/H-BSCF (Supplementary Table 3). This discrepancy can be attributed to the inferior phase distribution of the physically mixed samples in C/H-BSCF-PM, which hindered the synergistic effect of the two phases and subsequently hindered ion migration and surface exchange.”

b) The raw data curves and fitting curves in Figure S12 should be provided to verify the accuracy of fitting.

Response to C8b:

First of all, we thank the reviewer for the valuable comment. It is worth mentioning that the previous Figure S12 was a line graph plotted from the raw data, not a fitted curve. In order to avoid making readers think that we provided a fitted curve plot, we changed the line graph to a point graph plotted from the raw data (Fig. R18). In addition, we also added the ECR curves of C/H-BSCF-PM and H-BSCF samples in the revised manuscript.

Fig. R18 (also **Supplementary Fig. 16**) ECR curves for (a) C-BSCF, (b) C/H-BSCF, (c) C/H-BSCF-PM and (d) H-BSCF samples at 500-700 °C with pO_2 changes from 21% to 10%.

c) Sintering temperature of bar samples should be given in the experimental section, as well as SEM pictures of the surface and cross-section of bars.

Response to C8c:

We thank the reviewer for meticulous comment. We described the sintering temperatures of bar samples of different materials in the last sentence of the Materials synthesis and cell fabrication subsection in the experimental section. Electrical conductivity, ECR and coefficient of thermal expansion tests were performed using dense bar samples prepared from the corresponding air electrode powders by die casting method. C-BSCF, C/H-BSCF, C/H-BSCF-PM and H-BSCF samples were calcinated for 10 hours at 1130, 1200, 1200, and 1230 °C to obtain dense bar samples, respectively. **Figure R19** illustrates the SEM images

of the surface and cross-section of the C-BSCF, C/H-BSCF, C/H-BSCF-PM and H-BSCF bar samples, and the results indicate that each samples reached the required densities after calcination at the corresponding temperatures. The reliability of the experimental results is further confirmed.

Fig. R19 (also **Supplementary Fig. 17**) SEM images of the surface and cross-section of C-BSCF, C/H-

BSCF, C/H-BSCF-PM and H-BSCF bars.

d) When fabricating bar samples for ECR, the densification procedure requires a high sintering temperature, which may raise the change in composition or crystal structure. XRD and refinement results of sintered bars should be provided and discussed.

Response to C8d:

We appreciate the meticulous suggestion from the reviewer, and based on the comment we performed XRD testing and refinement analysis on C-BSCF, C/H-BSCF, C/H-BSCF-PM and H-BSCF bars samples, which were supplemented in the revised manuscript. Here, we calcined the cubic C-BSCF and hexagonal H-BSCF samples at 1130 and 1230 °C for 10 h, respectively, while the C/H-BSCF and C/H-BSCF-PM samples were obtained to be calcined at 1200 °C for 10 h. Subsequently, the four dense bar samples were subjected to XRD testing and refinement analysis as shown in **Fig. R20**. As expected, all the samples had good thermal stability and maintained their original phase structure after high temperature calcination and no obvious phase transition or impurity phase generation occurred. In addition, XRD refinement showed that the crystal structure of the surface of the C/H-BSCF sample after densification consisted of 58.45 wt.% of cubic phase and 41.55 wt.% of hexagonal phase. The revised C/H-BSCF powder had 57.26 wt.% of cubic phase and 42.74 wt.% of hexagonal phase, respectively. Although the two-phase contents of the densification-treated sample were slightly different from the initial powder, they were still within the acceptable range. The physically mixed C/H-BSCF-PM samples also exhibited similar mass ratios of the two phases, further suggesting that the hybrid air electrodes are thermally stable.

Fig. R20 (also **Supplementary Fig. 18**) Refined XRD profiles of the (a) C-BSCF, (b) C/H-BSCF, (c) C/H-BSCF-PM and (d) H-BSCF bars.

Related data (**Supplementary Fig. 18**) and discussions (yellow-highlighted in **Line 365-366, Page 19**) have been added in the revised manuscript as follows:

All samples were calcined to dense bars and the phase structure and composition remained stable (Supplementary Fig. 17,18).

Comment 9:

The stability of powders at wet atmosphere (40% H₂O) should be discussed according to XRD and SEM results.

Response to C9:

We thank the reviewer for these important concerns. In compliance with the reviewer's comment, we investigated the stability of C-BSCF, C/H-BSCF and H-BSCF powders under high water pressure through XRD and SEM analysis of the powders treated in a 40% H₂O-air environment. The revised manuscript has been supplemented with relevant discussions and data. **Figure R21a** illustrates the XRD patterns of C-BSCF, C/H-BSCF, and H-BSCF powders after treatment at 40% H₂O-air treatment at 600 °C for 10 hours. The C-BSCF sample maintains the diffraction peaks characteristic of cubic perovskite and shows no signs of impurity phases, indicating its excellent stability under high water pressure. On the other hand, the C/H-BSCF and H-BSCF samples retain their original crystal structures after the 40% H₂O-air treatment, but additional diffraction peaks corresponding to BaO_x were observed. This suggests that the high-water pressure enhances the segregation of Ba in the H-BSCF samples. Furthermore, based on the intensity of the diffraction peaks, it is evident that the H-BSCF sample possesses a higher content of BaO_x compared to the C/H-BSCF sample, which may be attributed to the lower content of Ba element or the hexagonal phase in the C/H-BSCF sample. **Figure R21b** presents the surface morphologies of C-BSCF, C/H-BSCF, and H-BSCF powders treated at 600 °C for 10 hours in a 40% H₂O-air environment. These images further demonstrate that the hexagonal H-BSCF experiences significant segregation of BaO_x under high water pressure. Zhu *et al.* have previously shown that BaO_x plays a vital role in activating oxygen by introducing BaO_x into conventional perovskite and inert Au electrodes, resulting in a manifold increase in catalytic activity [Zhu *et al.*, *Sci. Adv.* 8 eabn4072 (2022)]. It has also been proven that BaO_x can effectively enhance the ORR and OER activities of Ba-containing perovskite oxides. Hence, appropriate segregation of Ba in the H-BSCF phase under high water pressure offers greater potential for enhancing electrochemical activity.

Fig. R21 (also **Supplementary Fig. 34**) **a** XRD profiles (**b**) and SEM images of the C-BSCF, C/H-BSCF and H-BSCF powders after treatment at 40% H₂O-air treatment at 600 °C for 10 hours.

Related data (**Supplementary Fig. 34**) and discussions (yellow-highlighted in **Line 581-589, Page 31**) have been added in the revised manuscript as follows:

*“Additionally, **Supplementary Fig. 34a** displays the XRD patterns of C-BSCF, C/H-BSCF and H-BSCF powders following treatment at 600 °C under 40% H₂O-air. It was discovered that C-BSCF still maintains a cubic phase structure, but that C/H-BSCF and H-BSCF exhibit varying degrees of secondary phase BaO_x diffraction peaks (**Supplementary Fig. 34b**). This suggests that the high-water pressure induces the segregation of Ba in the hexagonal phase. It has been demonstrated that BaO_x can effectively enhance the ORR and OER activities of Ba-containing perovskite oxides. Hence, appropriate segregation of Ba in the H-BSCF phase under high water pressure offers greater potential for enhancing electrochemical activity⁷⁷.*

Comment 10:

About DFT calculations

a) All of the structure files should be provided.

b) When constructing structures of C-BSCF and H-BSCF, there will be many different configurations with

different atomic arrangements due to doping. Did authors use the configuration with the lowest energy? The relative energy of different configurations should be provided and discussed.

c) More details about DFT calculations should be provided.

d) How many layers of slab models?

e) How many layers were fixed during the calculation?

f) Thermodynamic correction of adsorbates at operating condition should be performed, frequencies, ZPE and $T\Delta S$ of every structure should be provided.

g) Does dipole correction is taken into consideration for slab models?

h) Does the Hubbard correction is taken into consideration?

i) When calculating reaction pathways, is there any oxygen vacancy on the surface?

j) For calculations of surface, there are many different adsorption sites (line, Co-top, Fe-top, VO-top, bridge, and so on), which should be considered sufficiently to ensure the reaction occurs on the sites with the most negative value of adsorption energy.

k) The synergistic effect should be calculated by constructing a heterostructure, it is encouraged to perform calculations at the C/H-BSCF interface.

Response to C10:

We would like to express our gratitude to the reviewer for providing such comprehensive feedback. In light of these comments, we have re-performed the DFT calculations for the process energy bases of ORR and OER for the C-BSCF and H-BSCF air electrodes. Moreover, we deeply apologize for the oversight in our original manuscript, where some specific details regarding the DFT calculations were omitted. We have rectified this error by including the necessary information in the revised version. The following is a point-by-point response to the comments.

Related data (**Supplementary Fig. 23,24, Supplementary Table 5,6, and Fig. 5**) and discussions (yellow-highlighted in **Line 446-485, Page 24-25**) have been added in the revised manuscript as follows:

“The energy barriers of ORR and OER kinetic processes on cubic C-BSCF and hexagonal H-BSCF surfaces were studied using density functional theory (DFT). In this study, we selected stable low-index surfaces of the Co, Fe-terminated C-BSCF (100) and H-BSCF (0001) for DFT calculations,

respectively^{26,68}. As shown in **Supplementary Fig. 23**, the energies of different doped configurations of C-BSCF and H-BSCF were calculated and the most structurally stable models with the lowest total energies were selected for the energy calculations of the ORR and OER pathways (**Supplementary Table 4**). Moreover, to more accurately model the OER process on the surfaces of C-BSCF and H-BSCF, the Fe-top site on the electrode surface with lower H₂O adsorption energy was chosen for as the active site for both models (**Supplementary Fig. 24, Supplementary Table 5**). **Figure 5a, b** shows the energy profiles of a reasonable reaction pathway in order to fully investigate the OER process on C-BSCF and H-BSCF, respectively. The OER process on the air electrode surface is divided into several steps: H₂O adsorption (*H₂O, * indicates the species' adsorbed state), oxygen evolution (*H₂O → *OH + *H → *O + *H → *O → *O₂), and *O₂ desorption, whereas the ORR process is the inverse of the OER process^{69,70}. Notably, the maximum energy barrier to be crossed by C-BSCF in the OER pathway is 3.25 eV, and the reaction process is *O + *H → *O, indicating that proton conduction is the main factor restricting the OER rate of the C-BSCF electrode. In contrast, H-BSCF only requires crossing 1.15 eV in this process, which further validates the superior proton migration capability of hexagonal H-BSCF. Similarly the favorable proton migration of H-BSCF is also demonstrated in the *OH + *H → *OH process. For H-BSCF, O desorption becomes the major hindrance for the OER process, needing to overcome an energy barrier of 2.03 eV, whereas C-BSCF exhibits more advantages in terms of oxygen desorption and water adsorption compared to H-BSCF. Hence, it is evident that the combination of cubic and hexagonal phases in C/H-BSCF holds more advantages in the OER process than single-phase C-BSCF and H-BSCF. Based on the OER energy profiles of cubic C-BSCF and hexagonal H-BSCF, it is speculated that for the OER process on the surface of hybrid electrode C/H-BSCF, H₂O will be preferentially adsorbed on the surface of cubic phase and undergo initial decomposition (*H₂O → *OH + *H), followed by further decomposition of *OH on the surface of hexagonal phase and final completion of oxygen desorption (**Fig. 5e**). Furthermore, as the inverse process of OER, the energy profiles of the ORR processes of the cubic C-BSCF and hexagonal H-BSCF electrodes are illustrated in **Fig. 5c, d**, respectively. C-BSCF and H-BSCF have oxygen adsorption energies of -0.43 and -2.03 eV, indicating that the oxygen adsorption processes are both spontaneous. However, the largest energy barrier to overcome for the ORR process in the case of non-spontaneous

*water desorption is 1.98 and 1.03 eV, respectively, for the C-BSCF and H-BSCF electrodes. The energy barrier crossed by the cubic C-BSCF electrode from *O₂ to *H₂O is -6.48 eV, but the energy consumed by the H-BSCF electrode for the same reaction process is -3.93 eV, indicating that the cubic C-BSCF electrode has a significant advantage in the reduction of oxygen. As a consequence, based on the ORR process of cubic C-BSCF and hexagonal H-BSCF, a schematic representation of the ORR process occurring on the surface of the hybrid air electrode C/H-BSCF is depicted in Fig. 5f.”*

a) All of the structure files should be provided.

Response to C10a:

We appreciate the reviewer's comment. In our response, we have submitted the complete set of structural files regarding the DFT calculations. These files consist of the diverse doping structures of both C-BSCF and H-BSCF, as well as the structural files for the various adsorption sites of H₂O on C-BSCF and H-BSCF. Furthermore, we have included the structural files of the OER (ORR) processes for both C-BSCF and H-BSCF in our submission.

b) When constructing structures of C-BSCF and H-BSCF, there will be many different configurations with different atomic arrangements due to doping. Did authors use the configuration with the lowest energy? The relative energy of different configurations should be provided and discussed.

Response to C10b:

We gratitude to the reviewer for the valuable comment. In our study, we have taken into consideration three distinct doping configurations for the cubic C-BSCF and four diverse doping configurations for the hexagonal H-BSCF (**Fig. R22**). The energy levels of the various doped configurations of C-BSCF and H-BSCF were calculated and documented in **Table R1**. Notably, among these configurations, C-BSCF-3 and H-BSCF-4 exhibit the lowest energy values of -243.068 and -330.257 eV, respectively. Therefore, for subsequent modeling purposes, we have opted to utilize the C-BSCF-3 and H-BSCF-4 structures due to their minimal energy levels and enhanced stability.

Fig. R22 (also **Supplementary Fig. 23**) Different doping configurations of (a) C-BSCF and (b) H-BSCF.

Table R1 (also **Supplementary Table 4**) Energies of different doped configurations of C-BSCF and H-BSCF

Configuration names	Total energies (eV)
C-BSCF-1	-242.236
C-BSCF-2	-242.955
C-BSCF-3	-243.068
H-BSCF-1	-329.998
H-BSCF-2	-329.745
H-BSCF-3	-329.729
H-BSCF-4	-330.257

Related data (**Supplementary Fig. 23** and **Table 4**) and discussions (yellow-highlighted in **Line 446-452**,

Page 24) have been added in the revised manuscript as follows:

*“The energy barriers of ORR and OER kinetic processes on cubic C-BSCF and hexagonal H-BSCF surfaces were studied using density functional theory (DFT). In this study, we selected stable low-index surfaces of the Co, Fe-terminated C-BSCF (100) and H-BSCF (0001) for DFT calculations, respectively^{26,68}. As shown in **Supplementary Fig. 23**, the energies of different doped configurations of C-BSCF and H-BSCF were calculated and the most structurally stable models with the lowest total energies were selected for the energy calculations of the ORR and OER pathways (**Supplementary Table 4**).”*

c) More details about DFT calculations should be provided.

Response to C10c:

We appreciate the reviewer's comment. Based on this comment, we provide a more detailed description of the DFT calculations in the revised manuscript (yellow-highlighted in **Line 675-690, Page 35**).

“All the calculations are performed in the framework of the density functional theory with the projector augmented plane-wave method, as implemented in the Vienna ab initio simulation package [Kresse et al., Phys. Rev. B 59 1758 (1999)]. The generalized gradient approximation proposed by Perdew, Burke, and Ernzerhof is selected for the exchange-correlation potential [Perdew et al., Phys. Rev. Lett. 77 3865 (1996)]. The long-range van der Waals interaction is described by the DFT-D3 approach [Grimme et al., J. Chem. Phys. 132 154101 (2010)]. The cut-off energy for plane wave is set to 500 eV. The energy criterion is set to 10^{-6} eV in iterative solution of the Kohn-Sham equation. In order to consider the strong electronic correlations in the localized Fe-3d and Co-3d orbitals, we add on-site Hubbard U of 3.70 eV and 3.52 eV respectively [Lodziana et al., Phys. Rev. Lett. 99 206402 (2007); García-Mota et al., J. Phys. Chem. C 116 21077 (2012)]. A vacuum layer of 15 Å is added perpendicular to the sheet to avoid artificial interaction between periodic images, and dipol correction is taken into consideration. The Brillouin zone integration is performed using a $3\times 3\times 1$ k-mesh for C-BSCF and a $2\times 2\times 1$ k-mesh for H-BSCF. All the structures are relaxed until the residual forces on the atoms have declined to less than 0.03 eV/Å. After the structure optimizations converged, the stabilities of the adsorbed structures were checked by the frequency's calculations, and the thermodynamic correction of adsorbates at operating condition were

also calculated. Before the calculations, the most stable doping sites of Fe were determined by calculating the total energy of different doped structures. And the preferred adsorption sites for H₂O is also taken into consideration by calculating the total energy of corresponding adsorption structures.”

d) How many layers of slab models?

Response to C10d:

We appreciate the reviewer's comment. For C-BSCF, the model contains five layers of atoms, where the lower three layers are fixed to simulate the structure of the bulk phase. For H-BSCF, the number of atomic layers could not be counted because its structure is irregular in the 0001 direction. The modeling was done by taking 1.5 times the length of the c-edge along the 0001 direction, which is about 10 Å, where a periodic structure below was fixed to simulate the bulk phase structure.

e) How many layers were fixed during the calculation?

Response to C10e:

We appreciate the reviewer's comment. For the c-BSCF and H-BSCF models, we fixed the three atomic layers underneath and one cycle used to model the structure of the bulk phase, respectively.

f) Thermodynamic correction of adsorbates at operating condition should be performed, frequencies, ZPE and TAS of every structure should be provided.

Response to C10f:

We gratitude to the reviewer for the insightful comment. In our DFT calculations, we conducted frequency calculations for each adsorbed structure as well as the transition state structure. During these calculations, we maintained the matrix model's fixity, enabling the adsorbed small molecules to vibrate in 2N degrees of freedom, where N represents the number of atoms in the small molecule. The frequency of each vibrational mode of the adsorbed molecules was determined based on our calculation results.

For the stable adsorption structure, which lies outside the transition state structure, all small molecules exhibited vibrations without any imaginary frequencies. This observation confirms the stability of the

structure at that particular moment. However, with regards to the transition state structures, we noticed the presence of exactly one imaginary frequency. This signifies that the respective transition state structures are located at the saddle point of the potential energy surface, validating their role as reasonable transition state structures.

To further refine our analysis, we employed vaspkit to thermodynamically correct the adsorbed molecules at a temperature of 298 K. This correction allowed us to obtain the frequency, ZPE, and TΔS values for each structure (**Table R2, R3**). In our revised manuscript, we will incorporate the pertinent data to reinforce the rigor and scientific validity of our DFT calculations

Table R2 (also **Supplementary Table 6**) Frequencies of the structures of the C-BSCF and H-BSCF reaction processes.

		Frequencies (cm ⁻¹)						
		*H ₂ O	*OH+*H	*OH	*OH(TS)	*O+*H	*O	*O ₂
C-BSCF	1f	3712.524	3715.298	3698.838	1534.093	3431.05	539.747	1536.215
	2f	3306.291	3370.782	691.5385	893.0461	875.599	143.2482	112.6365
	3f	1554.014	946.7179	516.0957	630.6273	700.9091	121.4698	66.12432
	4f	725.7877	699.0307	151.0977	511.1154	368.7323		50.41883
	5f	579.6506	534.6541	132.7959	170.809	188.3148		24.84868
	6f	309.7386	483.7285	19.55591	588.3934	179.2097		13.80394
	7f	280.0206	228.1486					
	8f	137.8007	172.1399					
	9f	82.68794	121.7279					
H-BSCF	1f	3768.161	3773.422	3762.472	1825.158	3142.869	601.7816	879.6629
	2f	3644.396	2969.247	522.7825	1188.135	978.4043	235.1142	443.9459
	3f	1545.859	1029.351	440.1786	668.2542	698.8283	165.1762	428.4271
	4f	452.0684	815.571	251.2754	365.8682	535.0296		185.6172
	5f	377.3766	580.0342	106.7067	154.737	267.771		165.6534
	6f	307.6515	459.9621	70.92847	790.9388	213.8166		70.56255
	7f	173.7961	246.218					
	8f	70.90124	138.1902					
	9f	41.73783	80.61461					

Table R3 (also **Supplementary Table 7**) ZPE and TΔS of the structures of the C-BSCF and H-BSCF reaction processes.

	Reaction steps	ZPE	TAS
C-BSCF	*H ₂ O	0.662606	0.133428
	*OH+*H	0.636799	0.117254
	*OH	0.322974	0.083943
	*OH(TS)	0.231832	0.046634
	*O+*H	0.356072	0.079106
	*O	0.049871	0.082707
	*O ₂	0.111837	0.221142
H-BSCF	*H ₂ O	0.643598	0.072555
	*OH+*H	0.625664	0.12906
	*OH	0.31953	0.13701
	*OH(TS)	0.260501	0.052986
	*O+*H	0.361832	0.059495
	*O	0.062121	0.061685
	*O ₂	0.134763	0.085047

g) Does dipole correction is taken into consideration for slab models?

Response to C10g:

Thanks to the reviewer for the comment. We have considered dipole corrections in the DFT calculations for the slab models of C-BSCF and H-BSCF (**Fig. R23**).

Fig. R23 Work functions calculated by DFT for C-BSCF and H-BSCF electrodes.

h) Does the Hubbard correction is taken into consideration?

Response to C10h:

Thanks to the reviewer for the comment. We have considered the Hubbard correction in our calculations. In order to consider the strong electronic correlations in the localized Fe-3d and Co-3d orbitals, we add on-site Hubbard U of 3.70 eV and 3.52 eV respectively [Lodziana *et al.*, *Phys. Rev. Lett.* 99 206402 (2007); García-Mota *et al.*, *J. Phys. Chem. C* 116 21077 (2012)]. Finally, we have added these detailed parameters in the revised manuscript.

i) When calculating reaction pathways, is there any oxygen vacancy on the surface?

Response to C10i:

We thank the reviewer for this comment. There are no oxygen vacancies on the model's surface, which we used for calculating the reaction pathways. This approach is commonly employed to calculate the ORR and OER reaction pathway for fuel cell air electrodes [Zhai *et al.*, *Nat. Energy* 7 866 (2022); Wang *et al.*, *Adv. Mater.* Pei *et al.*, *Nat. Commun.* 13 2207 (2022); Song *et al.*, 11 2101899 (2021)]. For instance, Zhou *et al.* also computed the Gibbs free energy change for the OER process of BaCoO_{3-δ} and PrBa_{0.8}Ca_{0.2}Co₂O_{5+δ} air electrodes using the oxygen-full theoretical model [Zhou *et al.*, *ACS Energy Lett.* 6 1511 (2021)]. Additionally, Ding *et al.* and Niu *et al.* did not account for the impact of oxygen vacancies on their calculations when calculating the proton mobility energies of PrNi_{0.5}Co_{0.5}O_{3-δ} or when investigating the boost of ORR activity of La_{0.6}Sr_{0.4}Co_{0.2}Fe_{0.8}O_{3-δ} by various catalytic coatings by DFT calculations [Ding *et al.*, *Nat. Commun.* 11 1970 (2020); Niu *et al.*, *Adv. Funct. Mater.* 31 2100034 (2021)]. This is primarily because it can be challenging to predict the oxygen vacancy content of various air electrodes when temperature and humidity fluctuate, potentially resulting in significant discrepancies between the built model and the real. At the same time, the introduction of oxygen vacancies also affects the structural changes, while inevitably having an impact on the energy band structure of the system, the density of electronic states, and the charge distribution of the charged system. Furthermore, the different positions of the oxygen vacancies impact the adsorption and desorption of H₂O and O₂, as well as ionic and electronic conduction throughout the model. Consequently, this ultimately affects the calculation results for each material's reaction pathway. Hence, in order to conduct a more accurate comparison of

intrinsic catalytic activities among different air electrodes, we deliberately chose to exclude oxygen vacancies on the model's surface. However, this does not imply that the catalytic promotion by oxygen vacancies should be disregarded. In the revised manuscript, we will duly acknowledge this aspect.

j) For calculations of surface, there are many different adsorption sites (line, Co-top, Fe-top, VO-top, bridge, and so on), which should be considered sufficiently to ensure the reaction occurs on the sites with the most negative value of adsorption energy.

Response to C10j:

We thank the reviewer for this comment. We initially examined the adsorption of H₂O at various sites before calculating the OER reaction pathways for different air electrodes. For the C-BSCF model, we considered the possible adsorption of H₂O at the Fe-top site, two Co-top sites, the O-top site, and the Hollow site (**Fig. R24**). Among the above five adsorption sites, the lowest adsorption energy is -327.766 eV for the Fe-top site (**Table R4**). However, for the H-BSCF model we calculated the adsorption energies for the Ba-top site, the Co-top site, the Fe-top site, the Co-Co bridge site, and the O-top site to be -491.811, -491.253, -492.132, -492.035, and -492.206, respectively (**Fig. R24, Table R4**). Only the lowest adsorption energy of O-top site in the H-BSCF model was observed, but the catalytic activity of non-metallic sites is usually poor [*Pei et al., Nat. Commun. 13 2207 (2022)*]. Meanwhile, in order to facilitate the comparison with C-BSCF, here we choose the Fe-top site as the active site for both models. Finally, we add the adsorption energy data of different sites in the revised manuscript and describe them.

Fig. R24 (also **Supplementary Fig. 24**) Different adsorption sites on the surface of C-BSCF and H-BSCF models.

Table R4 (also **Supplementary Table 5**) Adsorption energies of different sites on the surfaces of C-BSCF and H-BSCF models.

	Adsorption sites	Total energy (eV)
C-BSCF	Co1-top	-327.561
	Co2-top	-327.605
	Fe-top	-327.766
	Hollow	-327.072
	O-top	-327.098
H-BSCF	Ba-top	-491.811
	Co-top	-491.253
	Fe-top	-492.132
	Line-Co	-492.035
	O-top	-492.206

Related data (**Supplementary Fig. 24** and **Table 4**) and discussions (yellow-highlighted in **Line 452-455**, **Page 24**) have been added in the revised manuscript as follows:

“Moreover, to more accurately model the OER process on the surfaces of C-BSCF and H-BSCF, the Fe-top site on the electrode surface with lower H₂O adsorption energy was chosen for as the active site for both models (Supplementary Fig. 24, Supplementary Table 5).”

k) The synergistic effect should be calculated by constructing a heterostructure, it is encouraged to perform calculations at the C/H-BSCF interface.

Response to C10k:

We thank the reviewer for this comment. According to the feedback from the reviewer, we attempted to create the heterostructure of C/H-BSCF, and then analyzed the synergistic effect of the two phases at the heterogeneous interface by DFT calculation. As shown in **Fig. R25**, we presented heterogeneous interface models comprising the (110) crystal plane of C-BSCF and the (001) crystal plane of H-BSCF, as well as the (001) crystal plane of C-BSCF and the (001) crystal plane of H-BSCF. The total number of atoms in these two heterostructure models are 336 and 748, respectively. The significant divergence in crystal structures between the two phases and the intricate structural features of the H-BSCF phase necessitated the construction of such extensive heterostructure models. However, even employing a heterostructure model of 336 atoms consisting of C-BSCF (110) and H-BSCF (001) to explore synergistic effects via DFT calculations is an unattainable task for any academic institution and unit, mainly due to the computational expenses and the challenge of scaling up a large number of kernels given the parallel efficiency constraints of most quantization software packages. Nevertheless, we fully agree with the reviewer's notion that heterogeneous interfaces in multiphase hybrid electrodes could play a crucial role in facilitating ionic conduction and enhancing catalytic activity. This aspect has been thoroughly discussed in response to comment 2. Furthermore, in our manuscript, we have also demonstrated, through electrochemical testing and characterization of the hybrid electrode C/H-BSCF, as well as the single-phase C-BSCF and H-BSCF, that the remarkable electrochemical performance of C/H-BSCF can be directly attributed to the synergistic effect of the two phases.

Fig. R25 The heterogeneous interface models comprising (a) the (110) crystal plane of C-BSCF and the (001) crystal plane of H-BSCF, (b) as well as the (001) crystal plane of C-BSCF and the (001) crystal plane of H-BSCF.

REVIEWER COMMENTS

Reviewer #1 (Remarks to the Author):

The authors have made great efforts to revising the manuscript. All my concerns have been addressed.

Reviewer #2 (Remarks to the Author):

The revised manuscript has addressed all the comments in my previous review. I recommend the paper to be accepted for publication.

Reviewer #3 (Remarks to the Author):

In the revised manuscripts, authors addressed questions about experimental results.

Unfortunately, for the results about DFT calculation, there are still some problems:

1. For the calculation models:

a) For the response of comment 10(b), authors only considered different atomic arrangements of Co\Fe, yet ignoring the arrangements of Ba\Sr. Nevertheless, arrangement of A sites is also important, and should be considered to get the most reasonable calculation models.

b) In this work, C-BSCF models with five layers were used. However, in the calculation of cubic perovskite oxides, 4, 8, or 12 layers should be used. When the number of layer is an integer multiple of 4, one can calculate surface energy by $E_{\text{surface}} = (E_{\text{slab}} - N \cdot E_{\text{bulk}}) / 2S$, which requires the numbers of atoms in slab model should be integer multiple of bulk. So the 5-layer model is too thin and can't be used to calculate surface energy.

c) Before using Co- Fe-terminated slab models, surface energies with different termination condition should be calculated and the termination surfaces with lowest surface energies should be used for further calculations. For example, in C-BSCF, there will be many termination conditions: Co-O, Co-Fe-O, Ba/Sr-O (related with A site arrangement). For H-BSCF, termination condition is more complex.

2. For calculation methods:

a) How did authors get transition states? more details should be provided.

b) When calculating surface reaction pathways, surface oxygen vacancy is critical since oxygen vacancy is regarded as active sites for ORR and OER. It is encouraged to take surface oxygen vacancy into consideration. Moreover, the position of surface oxygen vacancy should be also considered.

Point-to-Point Responses to Reviewers' Comments and Suggestions

Firstly, we would like to express our gratitude for the time and effort the reviewers dedicated to reviewing our manuscript (NCOMMS-23-23007A). We also highly appreciate the thoughtful comments and suggestions provided by the reviewers, which have substantially enhanced the quality and clarity of our manuscript. We have carefully revised the manuscript to thoroughly address all the comments and concerns raised. The modifications have been marked **in yellow** in the revised manuscript. Our point-to-point responses are presented below.

Reviewer #1 (Remarks to the Author):

The authors have made great efforts to revising the manuscript. All my concerns have been addressed.

Response:

We sincerely appreciate the reviewer #1's positive comments and kind recommendation of this manuscript for publication.

Reviewer #2 (Remarks to the Author):

The revised manuscript has addressed all the comments in my previous review. I recommend the paper to be accepted for publication.

Response:

We highly appreciate the reviewers #2' feedback and recommend the journal for acceptance of this paper.

Reviewer #3 (Remarks to the Author):

In the revised manuscripts, authors addressed questions about experimental results. Unfortunately, for the results about DFT calculation, there are still some problems:

Response:

We very much appreciate the Reviewer #3's valuable comments to improve our manuscript. According to the raised suggestions, our responses to his/her detailed comments per point are as follows.

Comment 1:

For the calculation models:

a) For the response of comment 10(b), authors only considered different atomic arrangements of Co\Fe, yet ignoring the arrangements of Ba\Sr. Nevertheless, arrangement of A sites is also important, and should be considered to get the most reasonable calculation models.

b) In this work, C-BSCF models with five layers were used. However, in the calculation of cubic perovskite oxides, 4, 8, or 12 layers should be used. When the number of layer is an integer multiple of 4, one can calculate surface energy by $E_{\text{surface}}=(E_{\text{slab}}-N \cdot E_{\text{bulk}})/2S$, which requires the numbers of atoms in slab model should be integer multiple of bulk. So the 5-layer model is too thin and can't be used to calculate surface energy.

c) Before using Co- Fe-terminated slab models, surface energies with different termination condition should be calculated and the termination surfaces with lowest surface energies should be used for further calculations. For example, in C-BSCF, there will be many termination conditions: Co-O, Co-Fe-O, Ba/Sr-O (related with A site arrangement). For H-BSCF, termination condition is more complex.

a) For the response of comment 10(b), authors only considered different atomic arrangements of Co\Fe, yet ignoring the arrangements of Ba\Sr. Nevertheless, arrangement of A sites is also important, and should be considered to get the most reasonable calculation models.

Response to C1a:

We thank the reviewer for this valuable comment and suggestion. We agree with the reviewer that arrangement of A sites is also important for the structural stability besides the active sites of B-site metal ions. As suggested by the reviewer, in the revised manuscript, we have also taken into consideration of the arrangements of Ba/Sr in the model construction to obtain the most reasonable computational model

besides the arrangements of Co\Fe. As shown in **Fig. R1**, we considered two and four common arrangements of Ba/Sr for cubic C-BSCF and hexagonal H-BSCF, respectively, in view of structural differences. The energy levels of various Ba/Sr arrangements for C-BSCF and H-BSCF were calculated and recorded, as presented in **Table R1**. It is worth noting that the lowest energy values were found for C-BSCF-1 and H-BSCF-1, which also have lowest energy values among three distinct B-site doping configurations for cubic C-BSCF and four diverse B-site doping configurations for hexagonal H-BSCF as calculated before. Therefore, in our work, we adopted the **C-BSCF-1** and **H-BSCF-1** structures with lowest Ba/Sr and Co/Fe arrangement total energies for the subsequent model construction.

Table R1 (also **Supplementary Table 4**) Total energies of different models of Ba/Sr arrangements for C-BSCF and H-BSCF.

Configuration names	Total energies (eV)
C-BSCF-1	-242.423
C-BSCF-2	-242.233
H-BSCF-1	-330.000
H-BSCF-2	-329.907
H-BSCF-3	-329.871
H-BSCF-4	-329.706

Fig. R1 (also **Supplementary Fig. 23**) Different models of Ba/Sr arrangements for (a) cubic C-BSCF and (b) hexagonal H-BSCF.

Related data (**Supplementary Fig. 23 and Supplementary Table 4**) and discussions (yellow-highlighted in **Line 447-450, Page 24**) have been added in the revised manuscript as follows:

“First, the energies of C-BSCF and H-BSCF with different A- and B-site arrangement configurations were calculated to choose the most reasonable models with the lowest total energy for subsequent calculations of the ORR and OER pathways (Supplementary Fig. 23 & 24, and Supplementary Table 4 & 5).”

b) In this work, C-BSCF models with five layers were used. However, in the calculation of cubic perovskite oxides, 4, 8, or 12 layers should be used. When the number of layer is an integer multiple of 4,

one can calculate surface energy by $E_{\text{surface}}=(E_{\text{slab}}-N \cdot E_{\text{bulk}})/2S$, which requires the numbers of atoms in slab model should be integer multiple of bulk. So, the 5-layer model is too thin and can't be used to calculate surface energy.

Response to C1b:

We thank the reviewer for this valuable comment and suggestion. We agree with the reviewer that surface energy (E_{surface}) is one common parameter for assessing model surface stability. In our previous manuscript, we mainly evaluated the model surface stability by calculating the total energies of the systems of different models during the calculation process using the 5-layer for C-BSCF. Total energy is determined by calculating the interaction energy of all electrons in the system, which encompasses electron exchange and Coulomb interactions between atoms, which has been also widely calculated for assessing model structure stability (*Int. J. Quantum Chem.* 1995, 55, 339; *Int. J. Quantum Chem.* 2014, 114, 14; *WIREs Energy Environ.* 2023, 5, e476; *Energy & Fuels* 2023, 37 6078; *Electrochim. Acta* 2023, 452, 142325; *J. Solid State Chem.* 2023, 317, 123650; *J. Mol. Liq.* 2023, 390, 122950; *Eur. Phys. J. Plus* 2023, 138, 307).

Additionally, **as suggested by the reviewer, we also calculated the E_{surface}** to evaluate the stability of the model employed in this manuscript **using an integer multiple of atomic layers of the bulk phase** (please see the **Table R2** in response to Comment 1c). As can be seen, the energy values obtained from the calculation of and E_{surface} and total energy have **consistent trends** for various C-BSCF and H-BSCF terminations, suggesting that total energy and E_{surface} can be both used to reflect the model surface stability in our work.

Furthermore, the number of model layers is appropriately streamlined in order to reduce the amount of computation in subsequent calculations.

c) Before using Co- Fe-terminated slab models, surface energies with different termination condition should be calculated and the termination surfaces with lowest surface energies should be used for further calculations. For example, in C-BSCF, there will be many termination conditions: Co-O, Co-Fe-O, Ba/Sr-O (related with A site arrangement). For H-BSCF, termination condition is more complex.

Response to C1c:

We thank the reviewer for this valuable comment and suggestion. Before using Co-Fe-terminated slab models, we calculated the surface energies of the different atomic surface terminations to obtain the most stable surface atomic configurations. As shown in **Fig. R2**, we calculated total energies and E_{surface} for the three common termination models (O, Fe/Co-O, and Ba/Sr-O) for C-BSCF. However, for H-BSCF, we selected five representative terminations for total energies and E_{surface} calculations due to the numerous types of atoms exposed on the same surface (**Fig. R3**). As can be seen from **Table R2**, the lowest E_{surface} and total energy values were both observed for C-BSCF-slab1 and H-BSCF-slab1 terminations, which have been used to calculate the energy barrier for the ORR/OER process.

Table R2 (also **Supplementary Table 6**) Total energies and E_{surface} values of different surface terminations for C-BSCF and H-BSCF.

Configuration names	Total energies (eV)	E_{surface} (eV)
C-BSCF-slab1	-491.558	-0.0549
C-BSCF-slab2	-491.533	-0.0547
C-BSCF-slab3	-474.561	0.0841
H-BSCF-slab1	-667.552	-0.0314
H-BSCF-slab2	-666.077	-0.0253
H-BSCF-slab3	-658.005	0.0083
H-BSCF-slab4	-654.677	0.0222
H-BSCF-slab5	-659.700	0.0012

Fig. R2 (also Supplementary Fig. 25a) Different surface terminations for C-BSCF.

Fig. R3 (also Supplementary Fig. 25b) Different surface terminations for H-BSCF.

Related data (**Supplementary Fig. 25 and Supplementary Table 6**) and discussions (yellow-highlighted in **Line 451-455, Page 24**) have been added in the revised manuscript as follows:

“In addition, we selected stable low-index surfaces of the Co, Fe-terminated C-BSCF (100) and H-BSCF (0001) for DFT calculations, respectively^{26,68}. Before using Co, Fe-terminated slab models, we calculated the surface energies of the different surface terminations to obtain the most stable surface atomic configurations (Supplementary Fig. 25, and Supplementary Table 6).”

Comment 2:

For calculation methods:

- a) How did authors get transition states? more details should be provided.
- b) When calculating surface reaction pathways, surface oxygen vacancy is critical since oxygen vacancy is regarded as active sites for ORR and OER. It is encouraged to take surface oxygen vacancy into consideration. Moreover, the position of surface oxygen vacancy should be also considered.

Response to C2a:

We thank the reviewer for raising this question and valuable suggestions. As suggested by the reviewer, we provide a more detailed description of the transition states in the revised manuscript. The transition states were calculated by CI-NEB method. 6 intermediate points (including the initial state and final state) were taken into consideration during solving the energy minimum path. After convergence of the calculations, the transition states were verified by frequency calculations.

Related detailed description (yellow-highlighted in **Line 693-696, Page 35**) have been added in the revised manuscript as follows:

“The transition states were calculated by CI-NEB method. 6 intermediate points (including the initial state and final state) were taken into consideration during solving the energy minimum path. After convergence of the calculations, the transition states were verified by frequency calculations.”

- b) When calculating surface reaction pathways, surface oxygen vacancy is critical since oxygen vacancy

is regarded as active sites for ORR and OER. It is encouraged to take surface oxygen vacancy into consideration. Moreover, the position of surface oxygen vacancy should be also considered.

Response to C2b:

We thank the reviewer for this valuable comment and suggestion. Firstly, we agree with that the reviewer that surface oxygen vacancy may be critical since oxygen vacancy could be possible active sites for ORR and OER. However, we apologize that, at present, we cannot take the oxygen vacancy into account for DFT calculations mainly due to the **huge complexity of our material system**. Through the communications with theoretical experts, it will be a **very difficult task** and **require a very long time** for calculating our work if considering oxygen vacancy in view of **the structural complexity of materials and the uncertainty of oxygen vacancy**. The H-BSCF oxide among the H-BSCF/C-BSCF hybrid, i.e., $\text{Ba}_4\text{Sr}_4(\text{Co}_{0.8}\text{Fe}_{0.2})_4\text{O}_{16}$, is one complex oxide with complex crystal structure, which has **many** possible oxygen-site positions for generating oxygen vacancy. Besides, by calculations, it's **challenging to accurately predict the oxygen vacancy concentrations** of H-BSCF and C-BSCF oxide in the hybrid air electrodes when temperature and humidity fluctuate, potentially resulting in significant discrepancies between the built models and the real. Thus, future in-depth theoretical investigations could shed more light on this when computational conditions are available.

Secondly, due to above mentioned limits, **currently many research groups** concerning studies on perovskite oxides for air electrodes in electrochemical cells **do not take into account the existence of oxygen vacancies in the DFT calculations** (*Nat. Energy* 2022, 7 866; *Nat. Energy* 2023, 8, 1145; *Nat. Commun.* 2022, 13 2207; *Nat. Commun.* 2020, 11 1970; *Joule* 2021, 11, 2101899; *ACS Catal.* 2019, 9, 5074; *ACS Energy Lett.* 2021, 6 1511; *Adv. Funct. Mater.* 2021, 31, 2100034; *Chem. Mater.* 2022, 34 5938; *Adv. Energy Mater.* 2021, 11, 2101899). For example, Ciucci *et al.* prepared a nanocomposite electrode consisting of tetragonal ($\text{SrFeO}_{3-\delta}$ -based perovskite, $\text{Sr}_a\text{Ce}_b\text{Fe}_c\text{Ni}_d\text{O}_{3-\delta}$) and RP perovskite ($\text{Sr}_4\text{Fe}_3\text{O}_{10-\delta}$ -based perovskite, $\text{Sr}_x\text{Ce}_y\text{Fe}_m\text{Ni}_n\text{O}_{3-\delta}$) phases enriched with CeO_2 and NiO nanoparticles on the surface. Experiments and calculations identify that the RP phase promotes hydration and proton transfer, while NiO and CeO_2 nanoparticles facilitate O_2 surface exchange and O^{2-} transfer from the surface to the major perovskite. Due to the complexity of the structure, **a model without oxygen vacancies** was used in the

calculation of the proton diffusion of both the tetragonal and RP perovskite phases by DFT. Furthermore, Ding's group synthesized nominal A-site excess $(\text{La}_{0.8}\text{Sr}_{0.2})_{1+x}\text{MnO}_3$ perovskite oxides for bifunctional oxygen electrocatalysis using the polymer-assisted chemical solution (PACS) method. The OER on the $\text{La}_{0.75}\text{Sr}_{0.25}\text{MnO}_3$ and $\text{La}_{0.75}\text{Sr}_{0.25}\text{Mn}_{0.92}\text{O}_3$ surfaces were also analyzed based on **two models without oxygen vacancies**. On the whole, the models without oxygen vacancies in complicated systems can be also used to calculate the physicochemical properties and reaction processes of air electrodes.

Thirdly, in this work, **DFT calculations based on the models without considering oxygen vacancy are consistent with the experimental results**. Series of control experiments and various characterizations (including XPS, TG, O_2 -TPD, ECR, etc.) indicate that hexagonal H-BSCF has superior proton migration capability and cubic C-BSCF is beneficial for oxygen activation, which has been also demonstrated by DFT results as shown in **Fig. 5**. Therefore, the **agreement** between the theoretical calculations and experimental results, as well as the ability to explain existing experimental phenomena, suggest **the effectiveness** of the current computational models.

REVIEWERS' COMMENTS

Reviewer #3 (Remarks to the Author):

All my concerns have been addressed. I recommend the paper to be accepted for publication.